# Aversive memory formation in humans involves an amygdala-hippocampus phase code

Manuela Costa [1,13] ✉, Diego Lozano-Soldevilla[1,13], Antonio Gil-Nagel [2,3], Rafael Toledano[2,4], Carina R. Oehrn [5], Lukas Kunz[6], Mar Yebra [1,11], Costantino Mendez-Bertolo[1,12], Lennart Stieglitz [7], Johannes Sarnthein[7], Nikolai Axmacher [8], Stephan Moratti[1,9] & Bryan A. Strange [1,10] ✉

Memory for aversive events is central to survival but can become maladaptive in psychiatric disorders. Memory enhancement for emotional events is thought to depend on amygdala modulation of hippocampal activity. However, the neural dynamics of amygdala-hippocampal communication during emotional memory encoding remain unknown. Using simultaneous intracranial recordings from both structures in human patients, here we show that successful emotional memory encoding depends on the amygdala theta phase to which hippocampal gamma activity and neuronal firing couple. The phase difference between subsequently remembered vs. not-remembered emotional stimuli translates to a time period that enables lagged coherence between amygdala and downstream hippocampal gamma. These results reveal a mechanism whereby amygdala theta phase coordinates transient amygdala-hippocampal gamma coherence to facilitate aversive memory encoding. Pacing of lagged gamma coherence via amygdala theta phase may represent a general mechanism through which the amygdala relays emotional content to distant brain regions to modulate other aspects of cognition, such as attention and decision-making.

We tend to remember emotional events better than neutral ones. Although adaptive to survival, emotional memory enhancement for traumatic experience can contribute to anxiety[1] and post-traumatic stress disorders[2]. Research in animal models and humans implicate the amygdala[3,4], and hippocampus[5,6] in emotional memory. Patients with selective amygdala lesions show reduced episodic memory for emotional items[7–9] and functional MRI (fMRI) studies have reported increased amygdala responses to emotional, relative to neutral, stimuli during memory encoding[10–12]. However, animal[13] and humans studies[10–12,14] suggest the amygdala is not a site of long-term episodic memory storage, but rather that it influences memory storage processes in the hippocampus. Despite this long-standing modulation hypothesis[13], the circuit-level neurophysiological mechanism

underlying amygdalo-hippocampal interactions is not yet understood. One proposal is that the interplay between the two structures occurs via coordinated oscillatory activity[15,16].

In humans, limitations of current non-invasive neuroimaging techniques dictate that the circuit dynamics between amygdala and hippocampus during emotional memory formation be determined using direct electrophysiological recordings from both structures. We therefore recorded intracranially from two cohorts of pharmaco-resistant epilepsy patients (Cohort 1: $n = 13$, Supplementary Table 1; Cohort 2: $n = 6$, Supplementary Table 2) while they performed an emotional memory task (Fig. 1a–c). In Cohort 1, electrodes had been implanted in the amygdala and in 8 patients also in the ipsilateral hippocampus (Fig. 1c), allowing simultaneous recordings of oscillatory

responses associated with emotional stimulus encoding from both structures (Supplementary Fig. 1a), and, critically, the assessment of emotional memory formation-dependent amygdala-hippocampal connectivity. Cohort 2 comprised patients with microelectrode recordings implanted in the amygdala and in the ipsilateral hippocampus, permitting analysis of single neuron activity.

Patients viewed neutral and emotional complex scenes and 24 h later performed a recognition memory test, during which these scenes were presented again, intermixed with novel emotional and neutral foils. Patients made "remember", "know", and "new" (R, K, N) responses to distinguish memories accompanied by a sense of recollection (R) rather than familiarity (K)[17]. Emotion has been shown to selectively enhance memories accompanied by a sense of recollection[18], and previous fMRI data demonstrate that amygdala encoding-related responses are larger to subsequently recollected vs. familiar or forgotten emotional stimuli[11]. By drawing on data from our two cohorts, we found a recollection enhancement for emotional stimuli that was associated with phase-dependent cross-frequency coupling between oscillatory activity in the amygdala and hippocampus. The phase of human amygdala theta to which hippocampal broadband gamma activity and neuronal firing couples was found to determine subsequent remembering of aversive stimuli. The importance of this theta phase difference appears to be in enabling transient lagged coherence between amygdala and hippocampal broadband gamma activity.

## Results

### Recollection is enhanced for emotionally aversive pictures

Across both patient cohorts, recollection performance for emotional stimuli (eR) was higher than all other response categories (Fig. 1b). As is commonly observed[19], the remember false alarm (RFA) rate was higher for emotional (21.94% ± 3.98; mean ± s.e.m.) compared to neutral stimuli (9.51% ± 1.56; $t_{17}$ = 3.76, $P$ = 0.002, d = 0.878). This difference may partially be linked to higher perceived relatedness for emotional items compared to neutral ones, which might increase a false sense of recollection due to generalization in memory[20,21]. However, it has been

shown that relatedness alone is not sufficient to account for all emotion-dependent differences in false recognition[19,22]. Performance for familiar (K) responses was equivalent for emotional (22.77% ± 3.48) and neutral items (20.55% ± 2.65; $t_{17}$ = 1.01, $P$ = 0.325, d = 0.170, Supplementary Table 3). Thus, memory recollection performance was evaluated by comparing the rate of R with K responses for emotional (eR, eK) vs. neutral items (nR, nK), as described previously[11]. This confirmed a significant emotion by memory interaction (two-way ANOVA within-subjects $F_{(1,17)}$ = 5.47, $P$ = 0.032, η² = 0.244) (Fig. 1b and Supplementary Table 4-5; Supplementary Fig. 3).

### Amygdala and hippocampal gamma responses during aversive memory formation

All data reported here pertain to encoding-related responses. To determine the oscillatory responses associated with enhanced emotional memory encoding, we employed a subsequent memory approach, categorizing trials according to whether patients later remembered (R) the stimulus presented at encoding or not (known/forgotten items, KF) to operationalize successful encoding. In the amygdala (Fig. 2a–c), successful encoding of aversive, but not of neutral scenes, was associated with fast gamma activity (97-125 Hz) starting 0.31 s after stimulus onset (emotion by subsequent memory interaction, summed $t$-value = 1012.36, $P$ = 0.01, Fig. 2b). Subsequently remembered aversive scenes induced higher broadband gamma power changes than subsequently known/forgotten aversive items (eR vs. eKF $t_{15}$ = 2.12, $P$ = 0.050, d = 0.53, *post-hoc t*-tests on mean power changes across significant time-frequency clusters, Fig. 2c, Supplementary note 1). Memory-related responses to neutral stimuli did not show a significant difference (nR vs. nKF, $t_{15}$ = −1.90, $P$ = 0.075, d = −0.47). Similar amygdala fast gamma responses (time window 0.41-1.1 s) were observed if the Cohort 1 patient group was restricted to those with electrodes in both amygdala and hippocampus (Supplementary Fig. 4).

By contrast to the amygdala spectral power changes, hippocampal-induced responses (50−75 Hz from 0.54 s) were

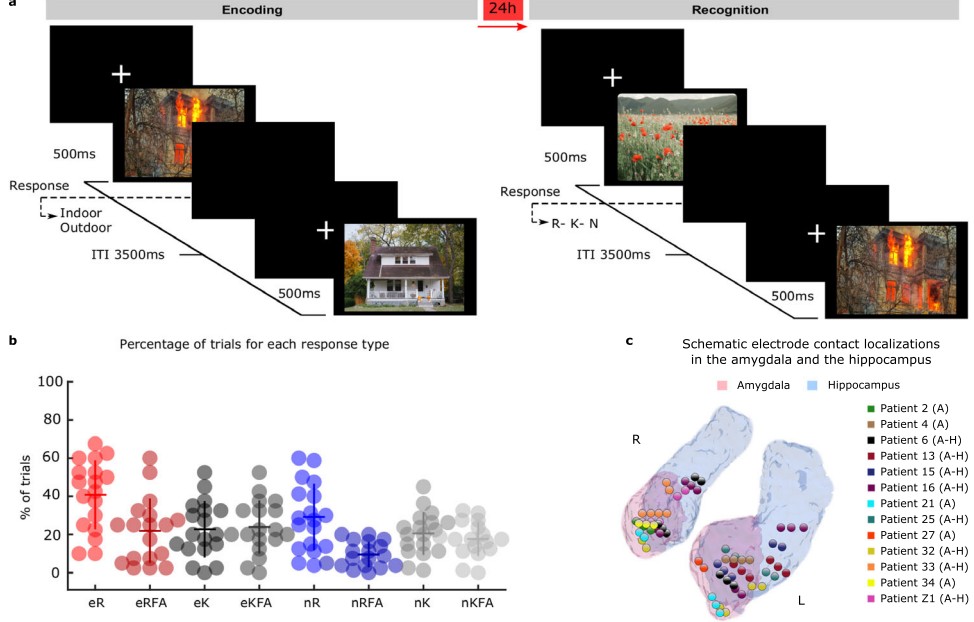

**Fig. 1 | Task design, behavior and electrode contact localization. a** Examples of aversive and neutral scene presented during the experiment. **b** Recognition performance for emotionally aversive (e) and neutral (n) correctly remembered, remembered false alarms, correctly known and known false alarms (R, RFA, K, KFA, respectively) responses for all patients (*n* = 18). Individual patient performance is plotted (horizontal/vertical lines indicate the mean/ ± s.e.m). For eR items, 100%

indicates correct remember responses to all previously presented emotional pictures and zero false alarms to new emotional pictures. **c** Summary of electrode contact localization for Cohort 1 in the left and right amygdala (pink) and hippocampus (light blue) for all patients included in the analysis (*n* = 13 patients; 17 amygdala (A) electrodes; *n* = 8 patients; 9 electrodes in both structures (A–H)).

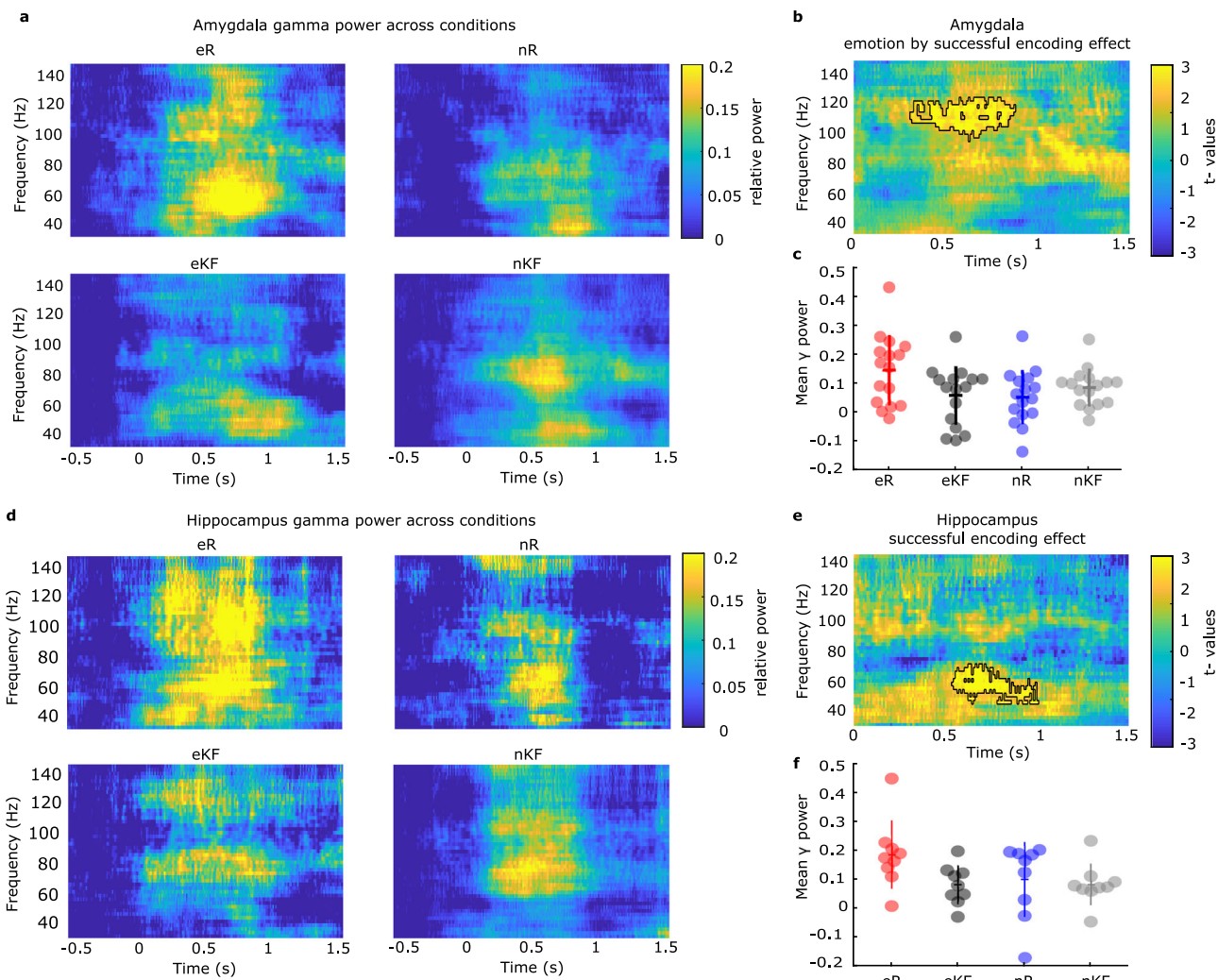

**Fig. 2 | Fast amygdala gamma activity tracks enhanced emotional memory formation, whereas hippocampal gamma activity increased during emotional and neutral memory formation. a** Time-frequency plots of amygdala broadband gamma power change for aversive and neutral subsequently remembered, and not remembered (known/forgotten) trials. **b** Time frequency resolved test statistics for the comparison between emotion and successful encoding effect, two-sided paired $t$ test (cluster-based permutation test). Black outline indicates the significant cluster. **c** Mean amygdala gamma power, relative to baseline, in the significant cluster for each amygdala electrode ($n = 12$ patients, 16 electrodes). **d** Time-frequency plots of hippocampus power change in the gamma range for aversive and neutral subsequently remembered KF trials. **e** Time frequency-resolved test statistics showing the main effect of successful encoding (R vs. KF), two-sided paired $t$ test (cluster-based permutation test). **f** Mean hippocampus gamma power, relative to baseline, in the significant cluster ($n = 8$ patients, 9 electrodes). Source Data are provided as a Source Data file.

associated with subsequent recollection (R vs. KF) of both aversive and neutral pictures (Fig. 2d–f; main effect of successful encoding, summed $t$-value = 776.82, $P = 0.0035$, Supplementary note 1). Regarding lower oscillatory frequencies (1–34 Hz), significant condition differences were not evident in either structure (Supplementary Fig. 5). Note that the power spectral analysis comparing subsequently remembered with either K or F responses showed analogous time frequency effects in both amygdala (Supplementary Fig. 6) and hippocampus (Supplementary Fig. 7), with no significant power differences for K vs. F in either structure. This supports previous observations showing a selective role of these structures in encoding that leads to subsequent recollection as opposed to familiarity responses[11], and provides a rationale for collapsing K and F trials in these and subsequent analyses. Note also that medial temporal cortical recording from entorhinal and perirhinal cortex showed greater gamma responses (50–67 Hz) following successful encoding of neutral, but not aversive scenes (Supplementary Fig. 8, $n = 8$), suggesting a specific mechanism between amygdala and hippocampus for emotional memory formation.

## Amygdala - hippocampus coupling during aversive scene viewing

Having characterized amygdala and hippocampal broadband gamma power responses during memory formation, we next systematically examined their coupling during emotional memory formation. In rodents, emotional memory retrieval is associated with amygdalo-hippocampal theta coherence[15]. In non-human primates, long-range communication between amygdala and cortical regions during aversive learning is achieved via theta coherence, with directional flow of information from amygdala to cortex[23,24]. We therefore tested for coherence and directional influence (Granger causality) between the amygdala and hippocampus. Unexpectedly, we did not find emotion-dependent coherence differences across encoding conditions (Supplementary Fig. 9). Directionality analysis using frequency-resolved Granger causality did, however, reveal greater causal influence from amygdala to hippocampus (lag of 0.36 s) during viewing of aversive as compared to neutral scenes. The effect was observed within the theta/alpha band (3–17 Hz) and a time window (0.43-0.77 s) in which amygdala gamma activity to subsequently remembered emotional stimuli

becomes pronounced (Fig. 2b), Supplementary Fig. 4 and Fig. 3a, b; main effect of emotion, summed $t$-value=1103.47, $P = 0.0039$; main effect of emotion in the opposite direction as not significant (Fig. 3c and Supplementary Fig. 10).

The degree to which single neuron spiking phase-locks to human hippocampal theta oscillations is a predictor of memory strength[25,26]. Given that spiking has been shown to correlate with gamma power[27,28], we hypothesized that a stronger modulation of hippocampal gamma band activity by amygdala theta oscillations would occur during encoding of aversive scenes leading to remembering. We thus tested the phase-amplitude coupling (PAC) associated with subsequently remembered vs. non-remembered emotional items by calculating the modulation index (MI) (Supplementary Fig. 11–14)[29]. Amygdala theta phase coupling with hippocampal gamma amplitude was on average stronger for viewing emotional compared to neutral scenes ($F_{(1,8)} = 6.73$, $P = 0.031$, $\eta^2 = 0.457$, Fig. 3d–e), similar to previous observations during passive viewing of emotional (fearful) faces vs. neutral landscapes[16]. However, PAC was unrelated to memory formation (i.e., MI values did not show a significant emotion by memory interaction, Fig. 3f).

Using a complementary approach, we isolated hippocampal gamma bursts in each condition and used them to triggered averages in amygdala raw field potentials. The cross-correlogram (CC) obtained from the peak-triggered averages (PTA) resulting from emotional vs. neutral stimuli confirmed that hippocampal gamma peaks locked to ongoing amygdala theta oscillations and did not show phase difference between emotional and neutral stimuli (Fig. 3g, for one representative patient, and Supplementary Fig. 15 for remaining patients), in keeping with theta-gamma PAC results derived via the MI approach (Fig. 3d, e). The same relationship was observed between hippocampal theta and gamma (Fig. 3h, Supplementary Fig. 17). At zero lag, cross-correlogram values equal to 1 indicate the same amygdala phase for hippocampal gamma bursts across conditions. By contrast, negative values at zero lag indicate gamma bursts locking to opposing theta phase bins.

### The amygdala theta phase to which hippocampal gamma activity and neuronal firing couple determines subsequent remembering of aversive events

Notably, peak-triggered averages for successful (eR) vs. unsuccessful encoded (eKF) aversive scenes revealed a theta phase difference between the two conditions (Fig. 3i), an effect observed across all patients (Supplementary Fig. 17). The importance of this latter observation is that a potential function of phase-amplitude coupling in supporting memory formation is the facilitation of synaptic plasticity[30], which, in the case of theta-gamma PAC, is thought to depend on theta phase[31]. In the context of the current findings, a phase-dependent mechanism underlying emotional memory formation suggests that amygdala theta-hippocampal gamma phase-amplitude coupling is evoked by all emotional stimuli, but that the magnitude of the gamma amplitude is concentrated at different amygdala theta phase bins for subsequently remembered vs. not remembered emotional stimuli. We tested this explicitly, by calculating the Phase-Amplitude Coupling Opposition index (PACOi), which quantifies the strength of phase opposition between two different stimulus types (i.e., eR vs. eKF; Supplementary Fig. 18). In line with the peak-triggered average results, hippocampal gamma amplitude during successful vs. unsuccessful encoding of aversive scenes concentrated at different theta bins with a consistent phase difference of ~1.67 radians, corresponding to approximately 30-45 ms time difference (permutation test using Watson Williams test, $F_{WW} = 187.49$, $P = 0.00009$, Fig. 4a–c). This was observed for all amygdala-hippocampus electrode pairs (Supplementary Fig. 19). This result established a phase code for amygdala theta-to-hippocampus gamma coupling in determining aversive memory formation in humans. We found that theta rhythm provides a time reference that allows

assignment of specific emotional memory encoding contents indexed by the phase difference at which gamma activity locks to theta. These results are in line with a theta/gamma model as a general brain coding scheme[32].

In analyses of data from Cohort 1, we considered broad band gamma activity as an index of spiking activity[27,28]. For Cohort 2, we directly tested whether amygdala theta (3-12 Hz) organizes hippocampal neuronal firing by performing spike field coherence analyses (SFC) on simultaneous recordings of hippocampal single neurons and amygdala oscillatory activity (Fig. 4d–e, Supplementary Fig. 1b). Overall, we observed that hippocampal spikes (from $n = 31$ neurons) during successful vs. unsuccessful encoding of aversive scenes concentrated at different theta bins with a consistent phase difference of -1.22 radians (two sample Kolmogorov-Smirnov test, $P = 0.00072$), corresponding to ~26 ms time lag. At the single-unit level, the firing distribution of 6 neurons differed significantly (circular Kuiper test, Supplementary Fig. 21) between the two conditions, more than expected by chance (binomial test versus 5% chance, $P < 0.01$, Fig. 4f–h). Note that capturing single unit activity is often a challenge in the context of human recordings. Thus, results may be limited by the low number of units captured in the recording.

### Amygdala theta phase coordinates transient coherence between amygdala and hippocampal gamma activity

Given that amygdala gamma power (Fig. 2a–c) and phase-dependent amygdala theta-coupled hippocampal gamma activity (Fig. 4a–c) both tracked successful subsequent remember responses for emotional stimuli, an important question remained: are these two broadband gamma activities related? Although measures of coherence between amygdala and hippocampus did not show significant effects in the 50–150 Hz range, it remained possible that successful encoding of aversive scenes involved transient connectivity between broadband gamma activity bursts in both structures, to which standard measures of coherence would be insensitive. We therefore computed amygdala-hippocampus transient connectivity in Cohort 1, taking the time range in which we observed significant amygdala high gamma activity power changes for successful encoding of aversive scenes (0.41–1.1 s; Supplementary Fig. 4). The frequency ranges of gamma activity differed between effects observed in the amygdala (80-120 Hz) and in the hippocampus (50–75 Hz). This variability most likely reflects differences in signal to noise ratio (SNR) suggesting that we are measuring a broadband gamma activity rather than an oscillatory activity. Moreover, cluster-based permutation tests[33] do not provide statistical inference for the exact latency and frequency of the effects[34]. We selected time windows around the peaks of broadband gamma activity (60–120 Hz)[27,28] as frequency range overlapping between the two structures in the hippocampal recordings. In a second step, we cut epochs around identified peaks (±0.03 s) for both hippocampus and amygdala time-courses. Lastly, cross-correlations between these windows were computed for successful and unsuccessful encoding of aversive scenes (Fig. 5a). The shift in time is important because the observed phase opposition between PAC for two conditions (i.e., eR vs. eKF) implies a condition-dependent phase lead/lag between the high frequency activity in hippocampus relative to the theta phase cycle in amygdala (Fig. 5a, b). However, we do not know whether broadband gamma activity in amygdala shows a similar lead/lag time difference relative to hippocampal broadband gamma. Using the amplitude envelope as a measure of synchrony strength between the two structures, we found stronger amygdala to hippocampus transient gamma synchronization for subsequently remembered compared to not remembered aversive scenes (lag 0.044–0.056 s, summed $t$-value = 29.73, $P = 0.011$, Fig. 5c, Supplementary Fig. 20; controlling the number of trials between conditions, lag 0.044–0.06 s, summed $t$-value = 31.00, $P = 0.011$). The time period where the difference appears may be prone to edge artifacts, although we note that there is no significant

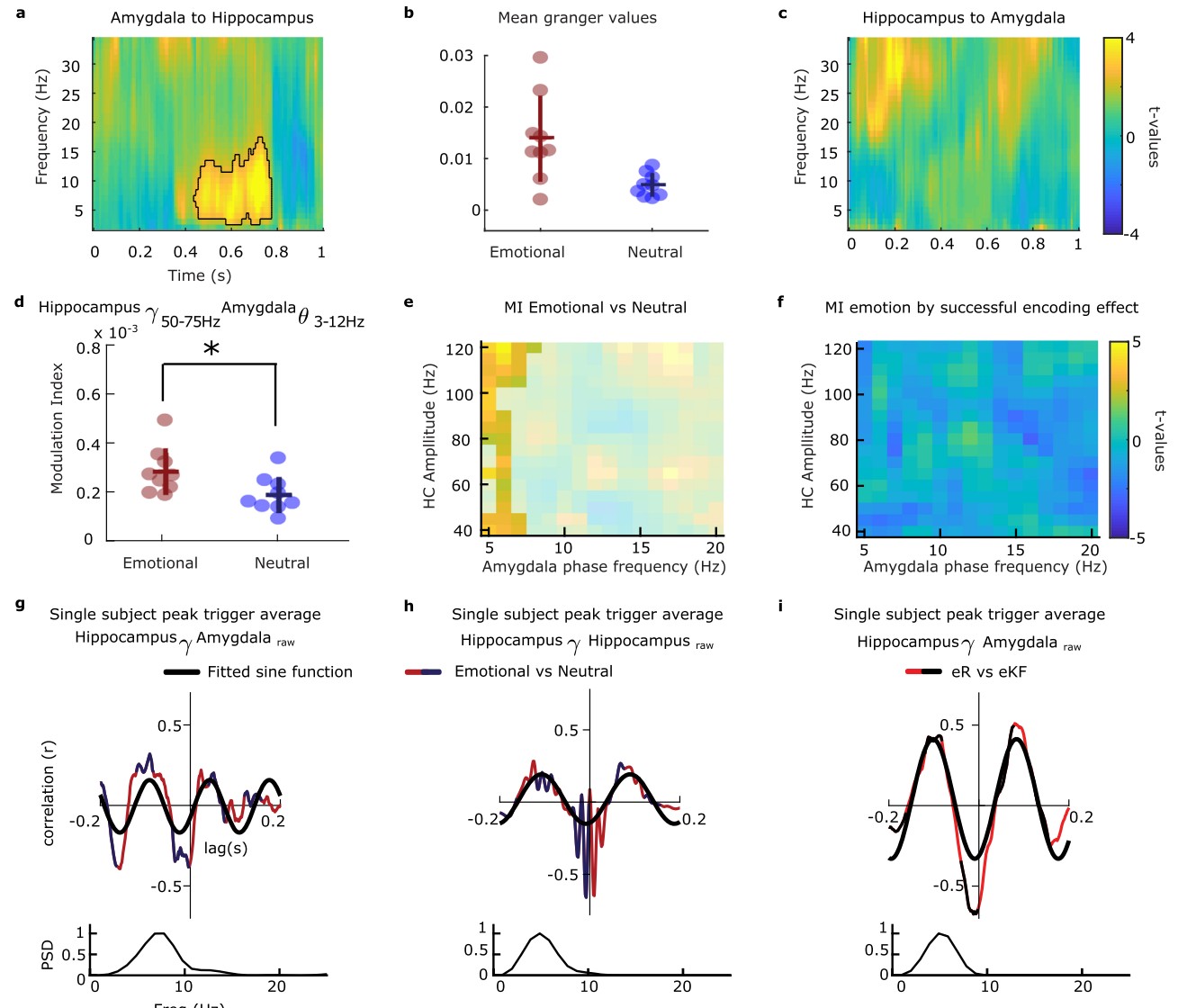

**Fig. 3 | Emotion-dependent amygdala-hippocampus theta-gamma coupling.**
**a** Granger causal influence of amygdala on hippocampal oscillations for aversive vs. neutral scenes (main effect of emotion), two-sided paired *t* test (cluster-based permutation test). **b** Mean Granger values for the significant cluster is plotted for each amygdala-hippocampus electrode pair (*n* = 8 patients; 9 electrode pairs). Horizontal/vertical bars represent mean/ ± s.e.m. **c** As for (**a**), but in the direction hippocampus to amygdala. **d** Amygdala phase of theta frequency (3–12 Hz) to hippocampus gamma amplitude (50–75 Hz) coupling is greater in response to viewing emotional vs. neutral stimuli, repeated measure ANOVA. Data points represent each pair of electrode contacts located in amygdala and hippocampus (*n* = 8 patients; 9 electrode pairs). *P = 0.031. **e** Inter-regional phase-to-amplitude coupling comodulograms (calculated using the Modulation Index method), with amygdala phase on the x-axis and hippocampus gamma amplitude on the y-axis,

showing group level statistics for the main effect of emotion (summed *t*-value = 104.95, *P* = 9.99 × 10⁻⁵), two-sided paired *t* test (cluster-based permutation test). **f** As for (**e**), but showing the absence of a significant emotion by successful encoding effect (eR-eKF vs. nR-nKF). **g** Inter-regional cross-correlation between emotional and neutral (dark blue/dark red) peak triggered average (PTA) and the corresponding power spectral density (PSD) of the cross-correlogram for Patient 6. A theta component is evident in the PSD and fitted sine wave (thick black line). **h** Same representative patient, within-hippocampus cross-correlation between emotional and neutral (blue/red) conditions. PTAs were computed as in **e** but averaging the raw local field potentials from the hippocampal recordings. **i** Same representative patient, PTA for aversive remember vs. know/forgotten (red/black) trials using the peaks of hippocampus gamma activity to average amygdala raw recordings. Source Data are provided as a Source Data file.

difference between conditions at the corresponding negative lag (indexing stronger hippocampus to amygdala transient gamma synchronization for subsequently remembered compared to not remembered aversive scenes). At shorter lags, there is more variability in amygdala to hippocampus transient gamma synchronization for subsequently remembered compared to not remembered aversive scenes (Supplementary Fig. 19), but this variability is, in turn, related to the patient-specific amygdala theta-hippocampal gamma phase lag between eR vs. eKF trials, as described next. Three individual patient amygdala-hippocampal gamma cross-correlations for eR vs. eKF trials are shown in Fig. 6b (inserts). Finally, to verify the role of ongoing

amygdala theta in the modulation of the transient gamma synchronization between amygdala and hippocampus, we tested whether the transient connectivity indexed by the amplitude envelope and PACOi results were related (Fig. 6a). PACOi phase differences were converted to time delays (Fig. 6b y-axis) using the following formula:

$$delay = \left( angle\left( z_{eR}^{f_x f_y} \right) - angle\left( z_{eKF}^{f_x f_y} \right) \right)/(2\pi) * (1/f_{x^c}) \qquad (1)$$

This formula measures, for each patient, the PACOi angle difference (as in Fig. 4a) between the eR and eKF conditions weighted by the

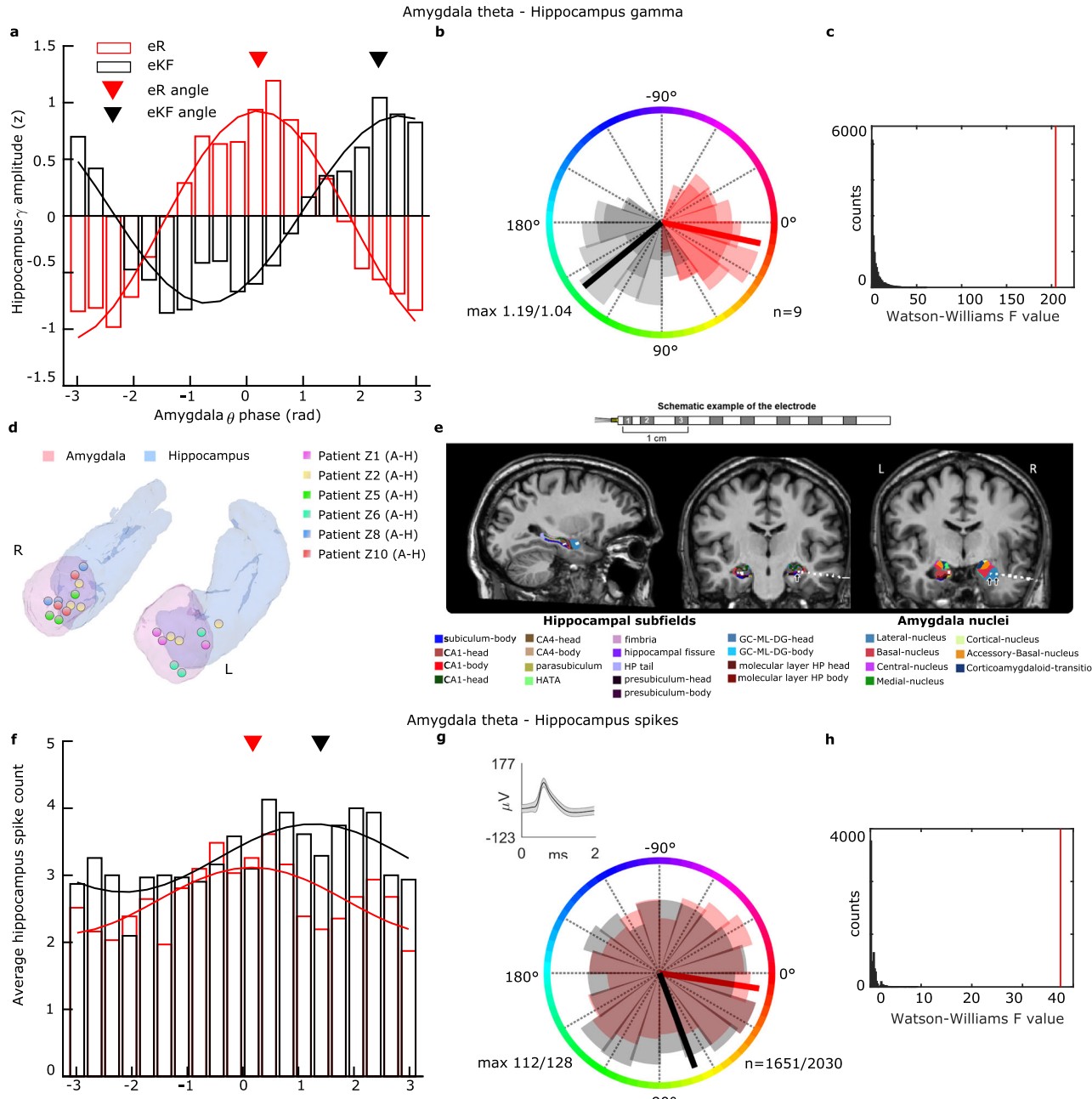

**Fig. 4 | Emotional memory-dependent amygdala phase opposition of hippocampal gamma and single neuron activity. a** Histogram of amygdala theta phases at which hippocampus broadband gamma amplitude occurs for eR (red) and eKF (black) trials. Inverted triangles represent the angle of preference in radians (eR:0.16 radians; eKF:1.83) after phase realignment. **b** Circular plot, shaded areas represent PACOi results (*n* = 8 patients, 9 electrodes) per phase bin (*n* = 20) for each condition; max is the maximum value of phase-amplitude coupling for eR:1.19 and eKF:1.04. **c** Histogram displays the permuted values for the Watson Williams test (one-way ANOVA for circular data), red line represents the empirical statistic (*FWW* = 187.49, *P* = 0.00009). **d** Summary of electrode contact localization in the left and right amygdala (pink) and hippocampus (light blue) for patients included in the SFC analysis (Cohort 2: 6 patients;7 electrodes). **e** Example for a single subject showing a post-operative CT image, thresholded to visualize electrode contacts, co-registered with the corresponding pre-operative MRI scan in native space and superimposed onto the amygdala nuclei and hippocampus subfields. The sagittal view shows that contacts are in the amygdala and anterior hippocampus. The first

coronal view shows hippocampal subfields and the white and black arrow point to the putative microelectrode location. The second coronal view shows amygdala nuclei and the two white and black arrows indicate the first two macroelectrodes locations. Subfields/nuclei are color coded. A schematic of the electrode implanted (Ad-Tech, Racine, WI) is provided showing microwires protruding from the tip and macro contacts (the first three contacts are numbered). **f** As for (**a**), but showing amygdala theta phases at which hippocampal spikes occur (average spike count over *n* = 31 neurons). Inverted triangles represent the angle of preference in radians (eR:0.18; eKF:1.40) after phase realignment. **g** Shaded areas in the circular plot represent spikes per the 20 phase bins for each condition. Maximum spike count for each condition (eR:112; eKF:128) and the total spike counts across neurons (eR:1651, eKF:2030) is indicated. Single-neuron waveform (mean ± std) for one example neuron is reported. **h** Histogram displays the permuted values for the Watson Williams test, red line represents the empirical statistic (*FWW* = 40.91, *P* = 0.00009).

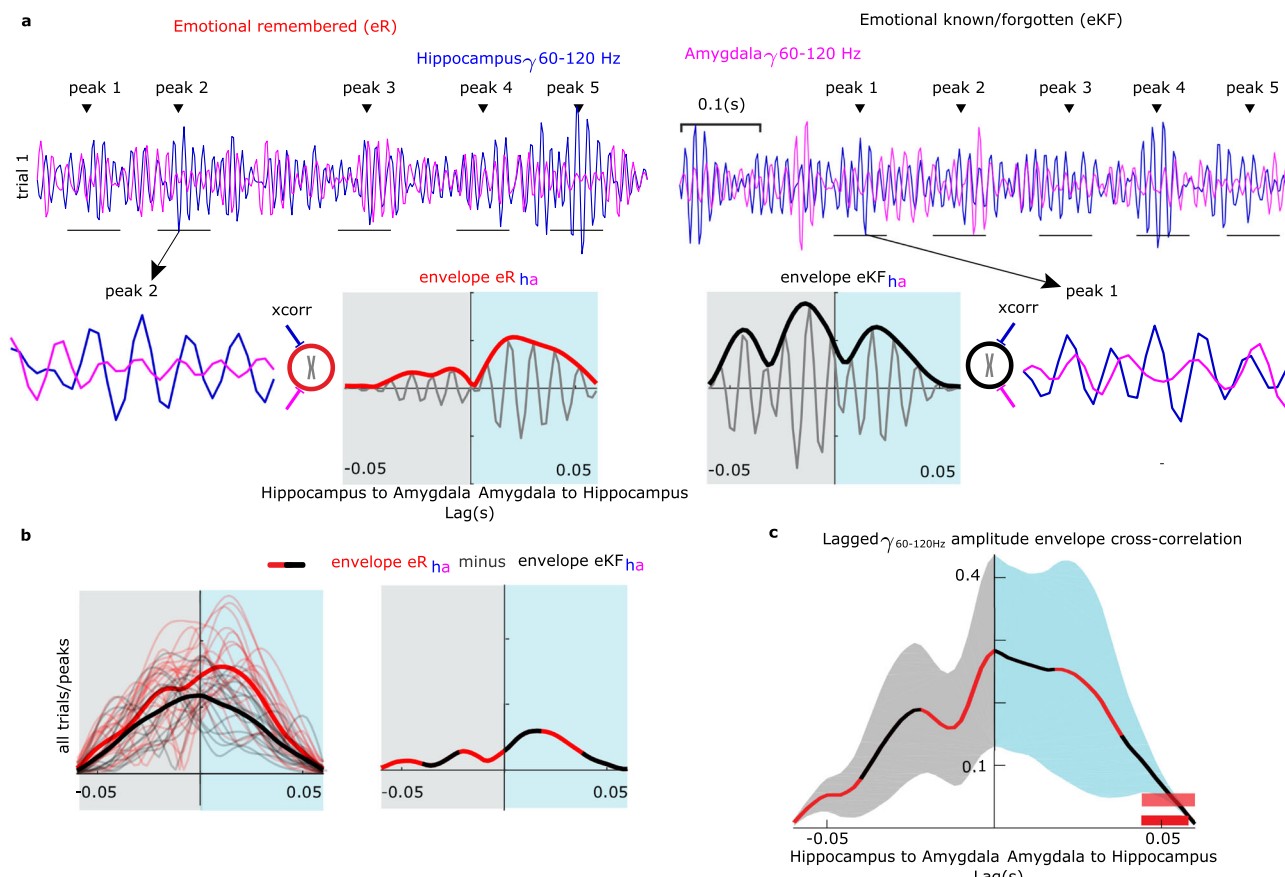

**Fig. 5 | Lagged coherence between amygdala and hippocampal broadband (60-120 Hz) gamma activity. a, b** Schematic depiction of the analysis. **a** Broadband gamma activity was bandpass filtered (60–120 Hz) and peaks in hippocampal recordings identified (top; two example trials from one patient). Short epochs around identified peaks were cut out (±0.03 s; thin horizontal lines under gamma traces). For each epoch, the amplitude envelope of the cross-correlation between hippocampus (blue) and amygdala (pink) epochs was computed for eR (below, red envelope) and eKF (black) trials. **b** The amplitude envelopes from all peaks and single trial cross-correlations (thin lines) were averaged (thick lines, left). The difference between the envelope eR_{ha} (emotional subsequent remember CC between hippocampal gamma peaks (h) and amygdala gamma peaks (a)) minus envelope

eKF_{ha} (emotional know/forgotten CC between hippocampal gamma peaks (h) and amygdala gamma peaks (a)) was computed (dashed red and black thick line, right). The negative and positive x-axis values of the envelope cross-correlogram indicate that hippocampus gamma leads amygdala gamma (light blue) and the reverse directionality (gray), respectively. **c** Following the analysis in **a** and **b** in all patients in Cohort 1, amplitude envelope cross-correlation shows amygdala leading hippocampus broadband gamma activity. The shaded contours represent ± s.e.m (color indicates directionality of transient coupling). Red bar depicts significant lag window for all trials; light red bar above after controlling the number of trials between conditions.

frequency centroid resulting from the significant PACOi low frequencies. These estimates were correlated with the peak of the envelope of the cross-correlation computed as the difference between subsequently remembered vs. not remembered aversive trials (Fig. 6b, x-axis). In line with the Granger causality results showing theta amygdala to hippocampus directionality effect, we found that amygdala gamma bursts lead hippocampal gamma bursts (Spearman rho = 0.78, $P = 0.013$; $P_{permuted} = 0.0078$, Fig. 6b), with a latency of 37.2 ± 5.9 ms (mean ± s.e.m.). The correlation suggests a monotonic relation between tri-partite activity (hippocampus gamma, amygdala theta, amygdala-hippocampus broadband 60–120 Hz): the PACOi theta-gamma time delay is linearly related to the lag obtained in the amplitude envelope cross-correlation in the 60–120 Hz range. This significant correlation indicates a role of amygdala theta in synchronizing amygdala and hippocampal high gamma activity.

## Discussion

Our data show that in response to an aversive visual stimulus, the amygdala influences ongoing hippocampal theta oscillations, which in turn organize the amplitude of local gamma activity and neuronal firing. High gamma activity in the amygdala was enhanced at 310 ms after stimulus presentation with a greater amplitude for emotional aversive

scenes that were later remembered (relative to those receiving known responses or forgotten ones). Subsequently, aversive scenes triggered unidirectional transmission of theta oscillations from the amygdala to the hippocampus compared to neutral ones (from 430 to 770 ms). At around 500 ms gamma power increased in the hippocampus for stimuli that were later recalled irrespective of their valence. Considering a time window from 400 to 1000 ms, the alignment of hippocampal broadband gamma activity (60–120 Hz) and neuronal firing to the amygdala's theta phase differed between subsequently remembered vs. not-remembered emotional stimuli. We observed a consistent phase difference between the two conditions of -1.67 radians, corresponding to approximately 30-45 ms. Crucially, this time difference also led to a transient lagged coherence (latency of around 37 ms) between gamma activity in the two structures that predicts subsequent memory for emotional stimuli (Fig. 7).

The BLA has strong connections to the hippocampus and electrical stimulation of the BLA improved performance in memory tasks[35,36]. In rodents, theta-modulated gamma stimulation applied to BLA is an efficient protocol to enhance hippocampal CA1 gamma responses[37,38], and directly stimulating the human amygdala at theta-modulated gamma frequency immediately following the presentation of emotionally neutral stimuli leads to memory enhancement[39]. Theta-

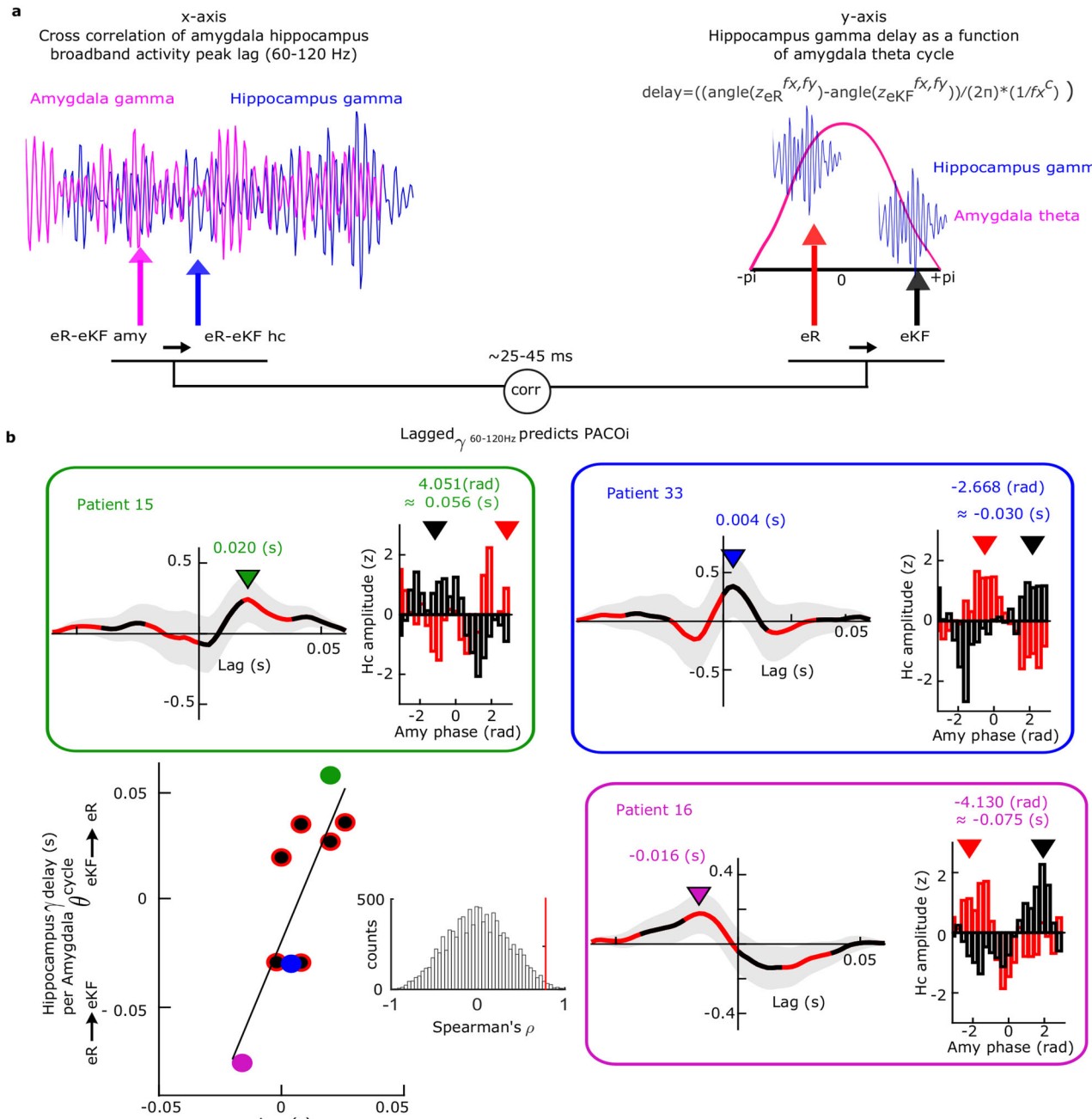

**Fig. 6 | Broadband gamma (60–120 Hz) transient connectivity between amygdala and hippocampus is paced by amygdala theta-hippocampal gamma phase opposition. a** Schematic depiction of the correlation analysis displayed in **b**. Left: example trial of broadband amygdala (pink) and hippocampal (blue) gamma activity. Cross correlation between hippocampal gamma peaks (blue arrow) and amygdala gamma peaks (pink arrow) for eR and eKF conditions was computed. Right: as example, we show hippocampal gamma activities (blue) occurring at two different amygdala theta phase (pink sine wave) for eR (red arrow) and eKF (black) conditions. Individual data for the cross correlation of amygdala hippocampus broadband activity peak lag were correlated with the PACOi frequency-specific phase opposition result translated into a time lag using the formula displayed in the figure. **b** Correlation between successful vs. unsuccessful emotional memory as measured as hippocampus-amygdala transient broadband gamma activity peak lag (x-axis) and hippocampus gamma delay as a function of amygdala theta cycle (y-axis). The colored dots (red, blue, purple) represent 3 different patients for whom individual data is shown above and right (with corresponding frame and font color). In these 3 plots, left: time corresponds to the hippocampus-amygdala transient broadband gamma activity cross-correlation peak lag for eR vs. eKF trials (colored inverted triangle). The shaded contours represent ± s.e.m.; right: hippocampus gamma delay as a function of amygdala theta cycle (the delay is the difference between black and red triangles). Note the different sign between Patient 15 (green) and Patient 16 (purple) in the two correlated measures. Source Data are provided as a Source Data file.

gamma phase-amplitude coupling in the human hippocampus has been shown to be a general mechanism for memory encoding[40,41]. Here, we showed that emotional memory enhancement does not simply depend on theta-gamma phase-amplitude coupling. Confirming theoretical positions that memory formation is phase-dependent in humans[23,24], we showed that the formation of memories for

emotional stimuli (later recollected emotional memories relative to familiar or forgotten ones) depends on the amygdala theta phase to which the hippocampal gamma and related neuronal firing couples.

We highlight the importance of theta modulation of gamma activity in encoding of emotional memories and lend support to direct stimulation protocols employing theta-burst stimulation aimed at

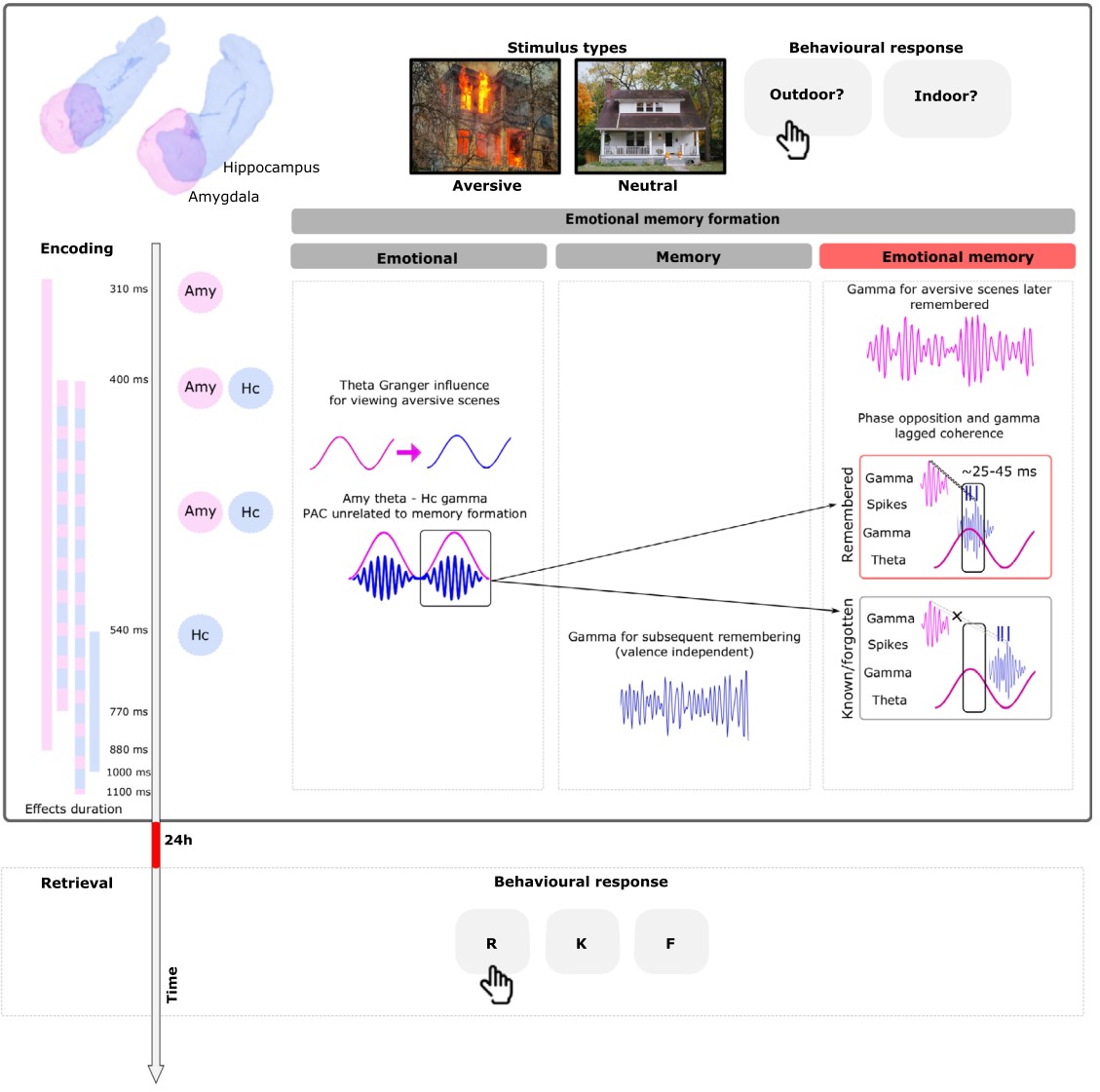

**Fig. 7 | Summary and time course of amygdala hippocampal activity and coupling during emotional memory formation.** All data reported pertain to encoding-related responses (gray box). Peri-stimulus time is indicated by the downward arrow (left), with process onsets and durations (solid color bar) observed in the amygdala (pink), in the hippocampus (blue) or implying a dynamic between the two structures (pink and blue). Effects at encoding are divided (from left to right) based on whether we observed a significant effect for aversive scenes (emotion column), for subsequently remembered scenes (memory column), or are significantly discriminative for emotional vs. neutral memory formation (emotional memory column). At earliest latency, amygdala gamma activity increases more for subsequently remembered vs. not remembered emotional vs. neutral pictures. Next, amygdala-hippocampal PAC and Granger causal effects are associated with viewing aversive scenes as compared to neutral ones. The pink horizontal arrow represents the directionality of Granger causal influence from the amygdala to the hippocampus. Gamma activity in the hippocampus increased for stimuli that are later recollected independently of their valence. Hippocampal gamma and single-unit responses aligned to different amygdala theta phases during subsequently remembered vs. not-remembered (known/forgotten) emotional stimuli. The vertical rectangle represents the amygdala theta phase to which hippocampal gamma activity and neuronal firing couple for aversive scenes later remembered. If gamma activity in the hippocampus occurred during a different period of the theta cycle (outside the rectangle), aversive scenes were not subsequently remembered. Thick dashed line represents the transient lagged coherence between gamma activity in the amygdala and the hippocampus that is observed only for aversive scenes later remembered. The latency of around 37 ms is consistent with the phase difference between the two conditions around 25–45 ms. These results together point to an amygdala-hippocampus phase code for aversive memory formation.

improving memory[39,42,43]. Theta-burst stimulation is the delivery of several stimulation pulses at high frequency (i.e., gamma frequency) that rhythmically alternate (in the theta range) with periods of no stimulation. By contrast, deep brain stimulation of the amygdala applied continuously at high frequency (e.g., 160 Hz), currently being trialed for PTSD[44], may not permit the amygdala-hippocampal phase code mechanism described here to take place in response to emotional events, thereby limiting the formation of novel emotional memories.

Hippocampal long term potentiation (LTP), a form of synaptic plasticity thought to be involved in learning and memory[30,45], is optimally induced in CA1 when electrical stimulation occurs at the peak of the theta oscillation[46]. This has led to a suggestion that inputs arriving during different specific theta phases will generate different synaptic modifications, which in turn will influence the likelihood that these inputs are encoded into long term memory[31,46,47]. Our findings support this suggestion, by showing that the precise theta phase at which hippocampal gamma peaks and spikes occurred determined whether aversive scenes were later remembered, or not. Critically, the phase difference between subsequently remembered vs. forgotten or familiar emotional stimuli translates to the time lag required for amygdala and hippocampal gamma bursts to reach transient time-lagged coherence (~25-45 ms). It is possible that this lag is related to the

time required for noradrenergic input, upon which emotional memory formation is critically dependent[12,13], to reach the medial temporal lobe, or that amygdala theta phase-dependent effects in the hippocampus are linked to the optimal conditions required for "emotion tagging" of memory[48] to occur. Future studies may explore whether a similar phase offset is present at retrieval or whether a different phase relationship may exist between amygdala and hippocampus. As encoding and retrieval has been shown to rely on complementary processes, the latter may be hippocampus-centered and thus may not rely on amygdala modulation[12,49]. Moreover, these findings could represent a general mechanism through which amygdala oscillations influence other brain areas to enable emotion-induced modulation of further aspects of cognition, including perception, attention and decision-making[14,50], and inform therapeutic approaches of amygdala stimulation to memory dysfunctions and psychiatric disorders.

## Methods

### Participants

Cohort 1: participants were 13 medication-resistant presurgical epilepsy patients with depth electrodes surgically implanted to aid seizure focus localization, age 18–59 years, 7 females (Supplementary Table 1, Supplementary Fig. 1a). Implantation sites were chosen solely on the basis of clinical criteria. Patients had normal or corrected-to-normal vision and had no history of head trauma or encephalitis. All patients had electrodes implanted in the amygdala and all amygdalae were radiologically normal on the pre-operative MRI.

We analyzed electrophysiological responses from 17 amygdalae from 13 patients (five had left, 4 right, and 4 bilateral medial temporal electrodes in the amygdala). Eleven patients also had electrodes in the ipsilateral anterior hippocampus. The hippocampal recording of one patient was excluded after hippocampal sclerosis was reported on the same side as the unilateral electrode implantation. For the second patient showing hippocampal sclerosis, only the non-pathological side was included. One further patient did not meet our criteria for spike-free trials (75%). We therefore analyzed electrophysiological responses from 9 hippocampi from 8 patients (1 patient had bilateral hippocampal electrodes). No statistical methods were used to pre-determine sample sizes but our sample sizes are larger or equal to those reported in previous publications[16,51]. Intracranial event-related potentials to the emotional and neutral pictures described, independent of subsequent memory, have been reported in 7 of the 13 patients presented here[51].

*Cohort 2:* This included 6 medication-resistant presurgical epilepsy patients implanted in the medial temporal lobe for diagnostic purposes, age 29-56 years, 3 females. We analyzed neuronal signals from $n = 7$ electrodes, simultaneously recording single neurons from the hippocampus and local field potentials (LFP) from the amygdala (Supplementary Table 2, Supplementary Fig. 1b). Note that patient Z1 is included in both cohorts; right medial temporal lobe in Cohort 1 and left in Cohort 2.

All patients signed informed consent and did not receive financial compensation. The study had full approval from the local ethics committees of the Hospital Ruber Internacional, Madrid, Spain and Kantonale Ethikkommission, Zurich, Switzerland (PB-2016-02055).

### Stereotactic electrode implantation

For the patients recorded at the Hospital Ruber Internacional (Cohort 1), a contrast enhanced MRI was performed pre-operatively under stereotactic conditions to map vascular structures prior to electrode implantation and to calculate stereotactic coordinates for trajectories using the Neuroplan system (Integra Radionics). DIXI Medical Microdeep depth electrodes (multi-contact, semi rigid, diameter of 0.8 mm, contact length of 2 mm, inter-contact isolator length of 1.5 mm) were implanted based on the stereotactic Leksell method. For the patients recorded in Zurich, Switzerland (Cohort 2 and patient Z1 in Cohort 1), the depth electrodes (1.3 mm diameter, 8 contacts of 1.6 mm length,

and spacing between contact centers 5 mm; Ad-Tech, Racine, WI, www. adtechmedical.com) were stereotactically implanted into the amygdala, hippocampus, and entorhinal cortex. Each macroelectrode had nine microelectrodes that protruded approximately 4 mm from its tip.

### Electrode contact localization

To localize electrodes, for each patient, the post-electrode placement CTs (post-CT) was co-registered to the pre-electrode placement T1-weighted magnetic resonance images (pre-MRI). To optimize co-registration, both brain images were first skull-stripped. For CTs this was done by filtering out all voxels with signal intensities between 100 and 1300 HU. Skull stripping of the pre-MRI proceeded by first spatially normalizing the image to MNI space employing the New Segment algorithm in SPM8 (http://www.fil.ion.ucl.ac.uk/spm). The resultant inverse normalization parameters were then applied to the brain mask supplied in SPM8 to transform the brain mask into the native space of the pre-MRI. All voxels in pre-MRI lying outside the brain mask and possessing a signal value in the highest 15th percentile were filtered out. The skull-stripped pre-MRI was then co-registered and re-sliced to the skull-stripped post-CT. Next, the pre-MRI was affine normalized to the post-CT, thus transforming the pre-MRI image into native post-CT space. The two images were then overlaid, with the post-CT thresholded such that only electrode contacts were visible. Electrode contacts for each patient are shown in Supplementary Fig. 1ab in native space.

### Electrode contact visualization

Skull-stripped pre-MRI and post-CT were normalized to MNI space, as described in[52]. The three-dimensional view of amygdala and hippocampus is an average of the segmented bilateral amygdala and hippocampus from each patient's pre-operative MRI (segmentations done using FreeSurfer v.6 software, https://surfer.nmr.mgh.harvard.edu/). Electrode contact locations for both cohorts patients are displayed using Paraview (www.paraview.org) in Figs. 1c and 4b. In the few cases where patients had electrodes implanted in both anterior and posterior hippocampus, only the anterior electrode contacts were included in our analyses, given the greater anterior vs. posterior hippocampal connectivity with amygdala[6].

### Stimuli

Patients were presented with 40 emotional and 80 neutral color pictures during the encoding session. These were drawn at random from a pool of 80 high-arousing aversive (mutilations and attack) scenes selected from the International Affective Picture System (IAPS)[53], and 160 low-arousing neutral pictures: 149 taken from the IAPS (household scenes and neutral persons) and eleven neutral landscape pictures taken from the world-wide web. Mean normative IAPS picture ratings (s.e.m.) on a nine-point scale for valence were 5.05 (±0.05) for neutral, and 2.04 (±0.05) for aversive pictures, and for arousal were 3.29 (±0.06), and 6.3 (±0.07) for neutral and aversive pictures, respectively. Emotional items are better remembered than neutral ones[8], thus the ratio of aversive to neutral stimuli was 1:2 to promote a balanced number of trials per condition (Supplementary Table 6).

### Procedure

Prior to signing informed consent, patients were shown one example of an aversive IAPS picture and instructed that they would see similar pictures both on that day and the next. Task instructions were provided both verbally and on-screen in Spanish (Cohort 1) and in German for patients recorded in Switzerland (Cohort 2). Encoding and recognition sessions were conducted during the third and fourth post-operative days, respectively, in Madrid, and second and third post-operative days in Zurich (Fig. 1a). During the encoding session, emotional and neutral pictures were presented pseudo-randomly (presentation time 0.5 s; interstimulus interval 3.5 s) with a constraint that

emotional pictures were separated by at least one neutral picture. Pictures were displayed on a 27 × 20.3 cm video monitor (1024 × 768 pixels) placed at a distance of 50 cm from the subject's eyes (30.2° x 22.9°). Patients were required to make an indoor-outdoor judgment to each picture via button-press. On average they made a button press to 99.07% ± 0.30% (mean ± s.e.m) of the trials. After 24 hours, patients were presented with all stimuli from the encoding session, randomly intermixed with 120 foils (40 emotional and 80 neutral pictures). Patients were required to make a "remember", "know", "new" decision (R-K-N)[54]. During both sessions, patients remained as still as possible attending the center of the screen while avoiding verbalisations and minimising eye-blinks.

#### Data acquisition
At the Ruber Hospital Internacional, Madrid (Cohort 1), ongoing intracranial EEG (iEEG) activity was acquired using an XLTEK EMU128FS amplifier (XLTEK, Oakville, Ontario, Canada). Intracranial EEG data were recorded at each electrode contact site at a 500 Hz sampling rate (online bandpass filter 0.1–150 Hz) and referenced to linked mastoid electrodes. Three patients were recorded at 2000 Hz sampling rate and data were down-sampled to 500 Hz. Intracranial data in Zurich (Cohort 2) were recorded against a common intracranial reference with a Neuralynx ATLAS system with sampling rate 32000 Hz for microelectrodes (online band-pass filter 5–8000 Hz) and 4000 Hz for macroelectrodes (online band-pass filter 0.5–1000 Hz, later down-sampled to 500 Hz).

#### Pre-processing analysis
Intracranial EEG data analysis was performed using the FieldTrip toolbox (https://www.fieldtriptoolbox.org)[55] running on Matlab version R2017b (the Mathworks, Natick, MA, USA). For all patients, recordings were transformed to a bipolar derivation by subtracting signals from adjacent electrode contacts within the hippocampus or amygdala. Previous studies demonstrate that bipolar referencing optimizes estimates of local activity[56–58] and connectivity patterns between brain regions[59]. For each amygdala and hippocampus bipolar channel, experimental condition, and patient, epochs from −7.5–7.5 s peri-stimulus time with respect to picture onset were extracted from continuous iEEG data. Epochs were then de-trended and no off-line filtering was applied. Trials containing signal artifacts caused by epileptic spikes or electrical noise were detected on visual inspection in the time domain and removed (percentage of trials without epileptic spikes or noise for each patient is reported in Supplementary Table 6). Note, that when patients had electrodes implanted in the amygdala and the hippocampus, trials were rejected if an artifact was detected in either or both structures (Supplementary Table 7).

#### Spectral analysis
Time-resolved spectral decomposition was computed for each trial using 7 Slepian multi-tapers for high frequencies (>35 Hz) and a single Hanning taper for low frequencies (≤35 Hz). The selected Slepian tapers for the analysis of high frequencies were based on windows of 0.4 s width and a 10 Hz frequency smoothing. The time-resolved spectral estimation was done in 2.5 Hz steps. In contrast to the high frequency analysis where we used a constant time-frequency smoothing, sliding windows were defined by 7 cycles per frequency step for the low frequency analysis. Trial-by-trial visual inspection and artifact rejection was then repeated in the time-frequency domain. Trials with interictal epileptiform activity (such as fast high frequency activity followed by lower frequency power) as well as excessive noise, including broadband noise from hospital equipment were removed[60]. Time-frequency estimates were then baseline corrected by calculating the relative percentage change with respect to baseline (−1 to −0.1 s pre-stimulus time). The spectral activity was then averaged over channels within each structure (amygdala and hippocampus) and trials for each patient.

#### Statistical analysis
We analyzed electrophysiological responses from 17 amygdalae from 13 patients (five had left, 4 right, and 4 bilateral medial temporal electrodes in the amygdala) and 9 hippocampi from 8 patients (1 patient had bilateral hippocampal electrodes). All analysis collapsed over contacts (bipolar channels) that were anatomically limited to the amygdala or the hippocampus. Thus, statistical inference was based on a fixed effect approach as previously adopted[51]. To test for subsequent memory effects, we specified six effects of interests. The event at encoding corresponding to aversive and neutral pictures separated according to whether they were later remembered, or either received a familiarity judgment or were forgotten (eR, eKF, nR, nKF). Given the low number of aversive forgotten trials compared to aversive remember ones, we merged know and forgotten trials for both emotional and neutral ones and performed the analysis on this set of data. Note, that repeating the spectral power analysis (described below) comparing remember vs. forgotten trials showed analogous frequency effect to the power analysis in which we compared remember vs. known trials (Supplementary Fig. 6–7). We focused our analyses on encoding responses to subsequently remembered vs. known/forgotten items. In both amygdala and hippocampus, we first tested for the interaction of emotion by successful encoding (aversive remember (eR) – aversive known/forgotten (eKF) vs. (neutral remember (nR) – neutral known/forgotten (nKF)). Power interactions were computed using 8 bipolar channels obtained from 7 patients. One patient was excluded (Patient Z1) because only 2 trials in the neutral remembered condition were obtained. In absence of a significant interaction, we tested for the main effect of emotion (aversive – neutral), and the main effect of successful encoding (remember – known/forgotten).

To correct the family wise error rate in the context of multiple comparisons across time and frequency dimensions, we applied a cluster-based permutation test[33] to the ensuing *t*-values obtained at each time-frequency bin to determine significant interactions or main effects in the time-frequency domain. We performed a 2-tailed *t* test using a maximum summation cluster statistic Montecarlo method. At each permutation step, selecting the a-priori time of interest in our data (from 0–1.5 s), clusters were formed by temporal and frequency adjacency with cluster threshold of alpha = 0.05 and a paired *t*-test was calculated at each time and each frequency bin for high (35–150 Hz) and low frequencies (0–34 Hz) separately using a threshold of alpha = 0.025. Permutation steps were repeated 10000 times.

For every dependent measure (time-frequency power, connectivity measures, etc) we only report the significant results unless explicitly stated.

To control for potential pre-stimulus power effects during the baseline period, we performed cluster-based permutation test during the baseline time window to test for main effects of memory, and emotion, and their interaction, for amygdala and hippocampus recordings (n = 8). None of the contrasts yielded significant condition differences in the baseline period.

#### Connectivity analyses
For all connectivity analyses, only the most lateral electrode contact pairs (bipolar channels) in the amygdala and the hippocampus were included. Given that the depth of electrode implantation (i.e., how medial the electrode tip is placed) varies across patients, taking the most lateral contact pairs promotes homogeneity of contact localization within the same lateral portion of the amygdala and hippocampus sampled across patients. By doing so, it is likely that electrode contacts included in our analyses are placed within (baso)lateral amygdala and CA1 of the hippocampus. Note that we repeated the time-frequency

power analysis using the most lateral channels and we found analogous frequency effects (Supplementary Fig. 6–7).

## Coherence

To investigate interactions between regions, we first computed spectral coherence using Fieldtrip. We subtracted the event-related potential from each single trial and computed a fast Fourier transform (FFT) using the single taper von Hann method with 7 cycles per time window (−0.2 to 1.5 s in 0.1 s steps) from 2–150 Hz. Entering the magnitude of coherence for the same frequency range, we tested for main effects of emotion, successful encoding and the emotion by memory interaction (eR-eKF vs. nR-nKF) followed by non-parametric cluster-based permutation correction for multiple comparisons (see above).

## Time and frequency resolved Granger causality

This analysis was computed using the Fieldtrip and BSMART (http://www.brain-smart.org/) toolboxes[61]. To ensure covariance stationarity[62], the mean-corrected time series was submitted to a Kwiatkowski-Phillips-Schmidt-Shin (KPSS) test[63] for each brain region (amygdala and hippocampus), subject, and condition and it was stationary. The time domain data was first low-pass filtered at 85 Hz and then down-sampled to 250 Hz. We computed time-dependent sets of multivariate autoregressive coefficients in overlapping windows of 0.4 s length (moving forward in 1-time point steps from −0.5 to 1.5 s)[61]. The optimal model order was estimated for each patient, most lateral electrode pair and condition by means of the Bayesian Information criterion (BIC). We selected a model order of 9 so to capture Granger causality for all subjects within a sufficiently large time window (corresponding to a time lag of 0.036 s). The order of Granger causality corresponded to a lag of 36 ms, which is in the range of latencies observed for human hippocampal responses to direct electrical stimulation of the amygdala (10-40 ms)[64]. Thereafter, Granger causality (2-34 Hz) was calculated based on the transfer matrices computed from the autoregressive coefficients. In order to assess statistically the directionality between the amygdala and the hippocampus, the Granger coefficients were compared using a cluster-based permutation test to quantify main effects of emotion, successful encoding and the interaction between them (eR-eKF vs. nR-nKF) in both directions (amygdala to hippocampus and hippocampus to amygdala), as described previously[16,65].

## Phase to amplitude coupling: Modulation index (MI) and PACOi

As unequal trial numbers can confound phase estimates, we controlled for this as follows. For each patient, contact and experimental condition, the lowest number of trials was determined ($N_{min}$). From the other condition, a subset of $N_{min}$ observations was randomly drawn from the observations constituting that particular experimental condition and the index of interest was computed. This procedure was re-computed 20 times by randomly selecting the subset of trials to the rest of the experimental conditions with higher number of trials. The controlled subsampled index was the average taken over the sub-sampled estimates.

## Phase to amplitude modulation index

We calculated phase-to-amplitude coupling (PAC) using the Modulation index (MI) as previously defined[29,66]. This was performed over the time period where we found a subsequent memory by emotion interaction within amygdala gamma broadband activity (0.41–1.1 s; Fig. 2b). First, for each patient and bipolar channel of interest, single trials were bandpass filtered around two sets of frequencies to obtain instantaneous phase ($f_p$; from 5–20 Hz in steps of 1 Hz) and amplitude ($f_a$; from 40 to 120 Hz in steps of 5 Hz). The analytic phase ($\varphi f_p$) was obtained by taking the angle of the Hilbert transform of the bandpass filtered data around $f_p$ with a bandwidth of ±2 Hz. For the same trial,

the analytic amplitude ($af_a$) was similarly obtained but by taking the magnitude of the Hilbert transform of the bandpass filtered data around $f_a$ with a bandwidth of ±15 Hz. We used FIR filters with the filter order being set to three cycles of the lower bandwidth bound. Before filtering, single trials were z-scored. Second, for a given frequency pair, we constructed the amplitude-phase histograms as follows. The analytic phase signal was divided in $n = 20$ equal bins ($\varphi_n = [-\pi\ \pi]$) and the mean analytic amplitude was taken over those specific bins. Third, for each frequency pair, the MI is computed as the Kullback-Leibler divergence[67] between the amplitude-phase histogram pooled from all corresponding trials and compared to the uniform distribution.

## Phase-amplitude coupling opposition index (PACOi)

The MI measures the modulation of the amplitude of high frequencies by the phase of low frequencies independently of the preferred phase angle at which the high frequency amplitude occurs. We derived a metric, the PACOi, to exploit potential differences in the preferred phase angle that two PAC distributions can produce. For example, the maximum at which the amplitude of an oscillation (i.e., gamma) concentrates in a given phase bin (i.e., theta peak) in each experimental condition can be different (or not) from another condition (i.e., theta trough). To formalize this quantification, we tested whether the pairwise phase consistency (PPC) of each trial type exceeded the overall PPC taken over the two groups of trials together[68]. Supplementary Fig. 18 provides, schematically, the rationale and quantification of the measure. The calculation of PACOi starts by first transforming the amplitude-phase histograms (see Modulation index above) to the complex domain as follows:

$$z_k = 1/N \sum_{b=1}^{N} (a_b e^{i\phi_b}), \tag{2}$$

with $k$ being a single trial of a particular experimental condition, $N$ the total number of phase bins, and $a_b$ and $\phi_b$ representing the amplitude and the angle of a specific bin $b$. Therefore, we obtain a complex number per single trial and experimental condition. Once in the complex domain, these indices are normalized to unit length ($z/abs(z)$) and the PPC is computed for each condition separately ($PPC_1$, $PPC_2$), and then by combining the trials of the two conditions ($PPC_{all}$; Supplementary Fig. 18). This procedure was carried out, with the rationale that if the gamma amplitude in eR and eKF concentrates systematically at different bins of theta, then the sum of the PPC obtained from each experimental condition separately should be larger than the PPC obtained from all trials pooled. Statistical significance was computed by randomly shuffling (n = 1000) the trial condition assignment (aversive remembered vs. aversive known/forgotten)[68]. It is a common observation that the angular bin to which the phase opposition points varies between test subjects[68,69]. Each patient's phase distributions were therefore realigned (Fig. 4a) such that, for each patient, the amygdala theta phase at which the hippocampus broadband gamma coupled was set to a phase angle of zero under the eR condition. The exact number of phase shifts was obtained by computing the angle at which the vector strength was pointing at (angle($\langle z_{eR} \rangle$), being $\langle \rangle$ the average over complex single trials). Once the angle was obtained (i.e., $\pi/2$) this angle was subtracted from both the eR and eKF conditions. Thus, eR theta-gamma PAC histogram is centered around zero. However, eKF angle preference can fall either [-π 0] or [0 π], which is a nontrivial property confirming our hypothesis that eR and eKF are associated with opposite phase angles. By convention, we mirror flipped the eKF histogram distribution when the vector strength angle falls with the [-π 0] interval.

Although PACOi measures phase difference indirectly (see VanRullen 2016 for an in-depth discussion about phase opposition measures in general), it provides substantial advantages for dealing with two important problems that need to be considered in the context of

phase estimation: (1) the magnitude of the phase concentration by itself is not informative. (2) Relative vs. absolute phase difference. PACOi measures whether the phase opposition (i.e., phase difference) is higher than a null model where trials are permuted[68]. Since it is based on differences in the sum of PPCs of two conditions relative to the sum of PPCs of these two conditions with the trials permuted (null model), the PACOi measure is insensitive to the angle where the phase difference occurs. In other words, the advantage of using PACOi method is that PACOi can measure phase difference independently of where it occurs along the 0-2pi interval. This approach is analogous to classical coherence analysis, where consistent phase differences between channel-pairs across trials are clipped between 0 and 1 independently of their location in the Angard diagram.

### Inter-region peak-triggered averages (PTA)

Cross-correlations between the averaged PTAs of two conditions served to illustrate the phase-offset between the PTA of condition A and the PTA of condition B (Fig. 3g–i). To compute PTAs, first, we band-pass filtered the hippocampal gamma band between 50 and 75 Hz using a two-pass finite impulse response (FIR) filter with a filter order of 3 cycles in the lowest frequency bound. Peaks were detected (0.1 s minimum inter-peak distance) in this filtered signal and time intervals of ±0.12 s around these peaks were used to average the raw traces (z-scored) over the same region (hippocampus gamma - hippocampus raw PTA, Supplementary Fig. 16) or between regions, taking amygdala raw trace (hippocampal gamma - amygdala raw PTA, Supplementary Fig. 15–17). Once averaged, the PTAs were de-trended. Power spectral density (PSD) of the cross-correlated PTAs was computed to find the main spectral component that dominates the PTA (PSD peak; $f_{peak}$). For example, a cross-correlation between the averaged PTAs of two conditions with a theta sine wave component with a peak offset at time lag zero would be in line with a phase-opposition resulting from gamma activity peaks aligned at different theta phase bins in condition A and B. Best sinusoidal fit was computed over the cross-correlated PTA with frequency, phase, amplitude and offset as free parameters. The frequency parameter was constrained selecting the frequency range of $\pm f_{peak}/4$.

### Broadband gamma transient connectivity analysis

This analysis measured transient connectivity (envelope cross-correlations) between amygdala and hippocampal broadband gamma activity. First, we band-pass filtered the time series from the amygdala and the hippocampus contacts between 60–120 Hz[27,28] using a two-pass finite impulse response (FIR) filter with a filter order of 3 cycles of the lower bound. This range was chosen because it encompasses both amygdala and hippocampal gamma ranges. Before bandpass filtering, single trials were demeaned in order to reduce ringing artifacts[70,71]. Hippocampal gamma peaks were detected (0.1 s minimum inter-peak distance) in this filtered signal and ±0.03 s epochs were selected. For each epoch, the amplitude envelope of the single-trial cross-correlation between hippocampus and amygdala was computed (see Fig. 5a, b). The single-trial amplitude envelope was computed by taking the magnitude of the Hilbert transform. Finally, single trial amplitude envelopes were averaged for each contact and experimental condition.

### Relationship between within vs. cross-frequency hippocampus-amygdala coupling

The correlation reported in Fig. 6b was computed as follows. For each patient, the peak of the amplitude envelope of the cross-correlation for eR minus eKF was taken. This comparison shows both the magnitude and direction of the transient broadband gamma activity between hippocampus and amygdala relative to each experimental condition. On the other hand, for each patient and bipolar channel, PACOi was computed under the eR vs. eKF contrast. The resulting significant phase differences found between the high frequency hippocampus

amplitude as a function of amygdala low frequency (theta) phase for eR vs. eKF were transformed to the time domain as follows:

$$delay = \left( angle\left(z_{eR}^{f_x, f_y}\right) - angle\left(z_{eKF}^{f_x, f_y}\right) \right)/(2\pi) * (1/f_{x^c}) \qquad (3)$$

with $f_{x^c}$ being the center of mass taken over the amygdala low frequencies, $z$ being the PACOi, $f_x$ and $f_y$ representing the x (gamma) and y (theta) significant PACOi pairs of frequencies (Supplementary Fig. 18, 19). Delay was the output time index used in the correlation, representing the time difference (i.e., lead/lag) between hippocampus high frequency amplitude in eR and eKF given a low frequency amygdala phase. To test the strength of the correlation we performed a surrogate analysis in which we permuted the patients delay and peak envelope association and computed the Spearman rho after each shuffling ($n = 10000$). The observed rho was then compared to the permutation distribution.

### Spike detection and sorting

In 6 patients with microelectrodes placed in the hippocampus (Cohort 2, $n = 7$ electrodes, one patient had bilateral hippocampal electrodes, Supplementary Fig. 1b), we recorded broadband neuronal activity (Neuralynx, sampling rate 32 kHz), which we separated into single- or multi neuron activity and LFP. Neuronal spikes were detected and sorted using Wave_Clus[72]. We refer here to a putative neuron by the term "neuron." The default settings were employed with the following exceptions: "template_sdnum" was set to 1.5 to assign unsorted spikes to clusters in a more conservative manner; "min_clus" was set to 60 and "max_clus" was set to 10 in order to avoid over-clustering; and "min-temp" was set to 0.05 to avoid under-clustering. All clusters were visually inspected and judged based on the spike shape and its variance, inter-spike interval (ISI) distribution, and the presence of a plausible refractory period[73]. If necessary, clusters were manually adjusted or excluded. For recording quality assessment (Supplementary Fig. 21), we calculated (1) the number of neurons recorded on each wire, (2) the ISI refractoriness for each neuron, (3) the mean firing rate for each neuron, and (4) the waveform peak signal-to-noise ratio (SNR) for each neuron. The ISI refractoriness was assessed as the percentage of ISIs with a duration of <3 ms. The waveform peak SNR was determined as: $SNR = A_{peak}/STD_{noise}$, where $A_{peak}$ is the absolute amplitude of the peak of the mean waveform, and $STD_{noise}$ is the standard deviation of the raw trace (filtered between 300 and 3000 Hz). We isolated zero, one or two distinct neurons from each microelectrode. In every trial, the number of spikes occurring between 0.41 and 1.1 seconds was calculated and spikes with ISIs <3 ms were removed. We included $n = 33$ neurons in the hippocampus for further analysis. Mean firing rate for eR and eKF condition separately is reported in Supplementary Fig. 22.

### Spike field coherence

We examined the relationship between the hippocampal neuronal action potentials and the amygdala LFP oscillations. The LFP recordings were taken from the macroelectrode recording from the ipsilateral amygdala, except for two patients in whom the LFP was extracted from an ipsilateral amygdala microelectrode due to a system data acquisition error. Macroelectrode data were recorded at 4 kHz. We down-sampled recordings to 2 kHZ as previously reported[74] and then applied a 40 Hz low pass filter. To calculate the amygdala theta phase, we filtered the data from 3 to 12 Hz. The analytic phase was obtained by taking the angle of the Hilbert transform of the bandpass filtered data for each trial in the condition eR and eKF. The analytic phase signal was divided in n = 20 equal bins ($\varphi_n = [-\pi \, \pi]$) and for each condition separately we extracted the amygdala theta phase at which each hippocampal spike occurred and calculated the number of spikes occurring at each bin in each condition. For each neuron and each condition, we averaged over bins (using circ_mean in the Circular

Statistics Toolbox[75]) and obtained the preferential phase in radians to depict spike theta coherence. Since the angular bin to which the SFC points varies between subjects we realigned each patient's phase distributions as explained above (see PACOi method). Thus, we obtained a spike-field coherence histogram for each neuron. For each neuron we performed a circular Kuiper test and compared eR vs eKF distributions (Supplementary Fig. 21). We further tested whether the number of significant neurons was above chance using a binomial test. For the group level analysis, a two-sample Kolmogorov-Smirnov test on the average results from all observed neurons ($n = 31$) was performed (Fig. 4b).

## Reporting summary
Further information on research design is available in the Nature Research Reporting Summary linked to this article.

## Data availability
All data needed to evaluate the conclusions in the paper are present in the paper and/or the supplementary materials. The dataset generated in this study are available on the following GitHub repository Costa-Lozanoetal. The repository contains all data required to reproduce the results. The raw data are available from the authors upon reasonable request. Source data are provided with this paper.

## Code availability
Codes are available in the CostaLozanoetal repository, https://github.com/TheStrangeLab/CostaLozanoetal.

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

## Acknowledgements

This project has received funding from the European Research Council (ERC) under the European Union's Horizon 2020 research and innovation

programme (ERC-2018-COG 819814) and by the Swiss National Science Foundation (funded by SNSF 204651 to J.S.). M.C. was supported by the Comunidad de Madrid, Ayudas para la contratación de investigadores predoctorales e investigadores postdoctorales cofinanciadas por Fondo Social Europeo a través del Programa Operativo de Empleo Juvenil y la Iniciativa de Empleo Juvenil (YEI) (PEJD-2017-POST/BMD-4763). L.K. was supported by the German Research Foundation (DFG; KU 4060/1-1). We thank the electroencephalography technicians at the Hospital Ruber Internacional and the Swiss Epilepsy Center, as well as Isabel Montón Quesada and Linda Zhang for technical assistance. We thank Dr. Rufin VanRullen for providing helpful comments about the rationale behind the phase opposition metrics.

## Author contributions

B.A.S. designed the experiment. M.C., C.M.-B., M.Y., J.S. and R.T. collected data. M.C., D.L.-S., S.M., C.O, L.K., N.A. and B.A.S. performed analyses. R.T. and A.G.-N. monitored patients and performed clinical evaluation. L.S. performed surgical electrode implantation. M.C., D.L.-S., and B.A.S. wrote the paper with input from all of the other authors.

## Competing interests

The authors declare no competing interests.

## Additional information

[1]Laboratory for Clinical Neuroscience, Center for Biomedical Technology, Universidad Politécnica de Madrid, IdISSC, Madrid, Spain. [2]Epilepsy Unit, Department of Neurology, Hospital Ruber Internacional, Madrid, Spain. [3]Fundación Iniciativa Para las Neurociencias (FINCE), Madrid, Spain. [4]Hospital Universitario Ramón y Cajal, Servicio de Neurología, Madrid, Spain. [5]Department of Neurological Surgery, University of California, San Francisco, CA, USA. [6]Department of Biomedical Engineering, Columbia University, New York, NY, USA. [7]Department of Neurosurgery, University Hospital Zurich, University of Zurich, Zurich, Switzerland. [8]Department of Neuropsychology, Institute of Cognitive Neuroscience, Faculty of Psychology, Ruhr University Bochum, Universitaetsstrasse 150, 44801 Bochum, Germany. [9]Department of Experimental Psychology, Complutense University of Madrid, Madrid, Spain. [10]Department of Neuroimaging, Reina Sofia Centre for Alzheimer's Research, Madrid, Spain. [11]Present address: Department of Neurosurgery, Cedars-Sinai Medical Center, Los Angeles, CA 90048, USA. [12]Present address: Departamento de Psicología. Facultad de Ciencias de la Educación, Universidad de Cádiz, and Instituto de Investigación Biomédica de Cádiz (INIBICA), Cádiz, Spain. [13]These authors contributed equally: Manuela Costa, Diego Lozano-Soldevilla. ✉e-mail: manuela.costa@ctb.upm.es; bryan.strange@upm.es

