## [Peer Review File · Nature Communications]

Aversive memory formation in humans involves an amygdala-hippocampus phase codeEditorial Note: Parts of this Peer Review File have been redacted as indicated to remove third-party material where no permission to publish could be obtained.

REVIEWER COMMENTS

Reviewer #1 (Remarks to the Author):

Costa et al. investigated electrophysiological responses and interactions between the amygdala and hippocampus during aversive and neutral stimuli, via human intracranial recordings. A larger literature, across species, has implicated these structures in memory formation and its emotional modulation. In their study, the authors employ a standard study/test recognition paradigm, using aversive (emotional) and neutral images. During these conditions, simultaneous invasive recordings from the amygdala and hippocampus were performed in two cohorts. In the second cohort, single unit activity was also recorded from the hippocampus. The authors report that both amygdala and hippocampus display increased gamma activity to successfully remembered aversive stimuli. In addition, spectral-granger analysis is reported to show directional influence of theta range activity between amygdala->hippocampus for aversive stimuli – but not theta coherence. Building on this influence, the authors finally report that hippocampal gamma and spiking activity is modulated by amygdala theta phase for aversive memory behavior – forming the basis of the title claim for evidence of phase coding. While several of the authors claims are well supported, the later observations of phase coding are more marginally supported by the data, particularly single unit findings. Further controls and caveats related to these observations should be provided.

Main comments:

-As a general note, the authors should ensure that results reporting is always clearly conveying if the data being analyzed is from the encoding or recognition phase of the task throughout the manuscript. The authors do this in many places, but it might often occur several sentences later. Correcting this will benefit the reader.

-Related to this point, analyses comparing encoding and retrieval findings is limited (or if performed is hard to track). As the paradigm involves recognition of previously presented and new images, this provides an important point of comparison for responses to the same image, before and after learning. Given the focus on memory formation, and claims of evidence for a specific coding mechanism, comparing key findings at encoding and recognition of the same stimuli is critical. Does the same phase encoding occur during recognition? In addition to the points on surrogate data noted below, the recognition data provide a useful control dataset for showing the robustness of encoding data.

-Trial numbers differ for the emotional (40) and neutral (80) stimuli. However, it's often not clear how this sample difference, or other data divisions (R vs. K trial#), are being accounted for when comparing electrophysiological data for these conditions. Spectral and certainly cross-frequency analysis will be influenced by such sample size differences (trial number).

-Related to this point, it's not always clear how multiple electrodes/units from individual subjects (nested data) are being handled. As this raise issues of independence in samples, multi-level model approaches or data collapsing must typically be employed.

-Gamma effects are reported for several results, but the frequency range greatly differs. There are five different gamma ranges reported (97-125 Hz); (50-75 Hz); (50 -67 Hz); (50 -75 Hz); (60 -120 Hz); (80 -120 Hz). The authors should provide some interpretation of this variability and note it to the reader. Given the often low sensitivity of macro-depth electrodes in hippocampus/amygdala, these differences are most likely reflecting recording SNR, rather than unique functional signatures. These variable findings need to be reconciled with the specific analysis of a single range for gamma correlation later in the manuscript.

-Appropriate surrogate data are critical for establishing chance levels with several of the analyses performed. It does appear the authors have employed such methods (i.e. through permutation testing), but the methods would benefit from more explicit details about specific permutation (what variable is being randomized) to ensure the inference of interest is being tested. In addition, the permuted/chance levels of metrics should be included in plots where relevant. As the authors know, small samples for many of the measured employed will always provide non-zero metrics. For example, the broadband gamma correlation analysis should include a randomization control, that is plotted with the observed data shown – providing better ground for the reliability of the effect shown.

-For analysis of cross-frequency modulation between regions, a critical factor is the reliability of the specific frequency couplings. It would therefore be more important for the main figures to show comodulograms (currently in the supplement) and/or theta phase triggered spectrograms (not just triggered averages, for a given subject). Alternatively, a spectrogram of the triggered averages, rather than PSD, would better speak to specific coupling (see comments below on PSD).

-PACOi. While these LFP results suggests some phase consistent offset – it does not support the authors statement that: “This result established a phase code for amygdala theta-to-hippocampus gamma coupling in determining aversive memory formation in humans”.

-As shown for LFP data, basic task event-related raster plots of spiking data should be shown. This provides basic context for interpreting the presented oscillatory modulation of firing rates in hippocampus. Only showing collapsed cross-regional modulation of spiking limits the ability to adjudicate if meaningful single unit activity was captured (often a challenge for these human recordings).

-Related to the above, the spiking data show much weaker support for signs of phase coding. Why would a population measure, like gamma, present much clearer phase coding evidence than specific neurons (the actual code)?

Minor comments:

-For clarity, in the introduction (line 70), when the microelectrode cohort is mentioned, the anatomical site of recordings should also be mentioned, similar to how cohort 1 is described (i.e. amygdala and/or hippocampus, ipsilateral?).

-Behavioral results (line 92), report performance for ‘known’ items, but it’s not clear what the reported values reflect (mean number of items?), as the total number of items is yet to be mentioned, it’s hard to gauge the level of performance (e.g. ratio or % values).

-Why are spectrograms shown limited to higher-frequency ranges only? Also, spectrogram ‘power’ color bars require units.

-Why are different model tests used for Figure 1 (E; amygdala) and (H; hippocampus)?

-Conditions in Figure 1 (i) are misaligned to labels.

-Figure 2 (e-g) should be clearly labeled as coming for an example subject. Also, why are PSD plots show with such a narrow x-axis base? Limits the purpose of displaying this data and over emphasizes spectral shape.

Reviewer #2 (Remarks to the Author):

This paper investigates the oscillations in the amygdala and hippocampus during the encoding of emotional and neutral stimuli. The data comes from human subjects with intracranial depth electrodes, making it especially compelling. Subjects are run on a standard memory task that has

been used in numerous other studies, so the results are relevant to a large literature. While a wide variety of analytical techniques are employed, there are only a few key findings. First, high gamma activity in the amygdala is enhanced during emotional stimuli that are later recollected compared with those that are forgotten or neutral. On the other hand, mid-gamma activity in the hippocampus is strongest for recollected stimuli irrespective of their valence. Second, theta band activity in the amygdala preferentially entrains theta and gamma activity in the hippocampus during the encoding of emotional stimuli. Third, emotional and neutral stimuli drive differences in the entrainment of hippocampal gamma to amygdala theta. These top-line results are intriguing, but the underlying analyses that the authors use to arrive at them are sometimes confusing, round-about, or indirect. Indeed, the paper would benefit from a revision of some of its analyses (and exclusion of others). Below I will outline the problematic sections and offer alternatives that, in this reviewer's opinion, are more straightforward.

1. Line 192 to 195. It is unclear what is demonstrated by Fig. 2e. The authors claim that the cross-correlation in Fig. 2e shows that hippocampal gamma is preferentially phase locked to amygdala gamma for 'all emotional stimuli.' I am not sure how this figure shows that. The PTA method here is that the cross correlation (CC) is calculated between gamma burst peaks in the hippocampus and raw LFP signals in the amygdala. Then, the cross-correlation is calculated between these two cross-correlation functions and the PSD of it indicates the strength of entrainment. While the figure certainly exhibits phase locking between amygdala theta and hippocampal gamma, it does not say anything about whether it is stronger for emotional or neutral stimuli, let alone whether it applies to 'all emotional stimuli.' Wouldn't just switching the CCs between the neutral and emotional conditions yield a similarly strong PSD? The direct test of the difference between neutral and emotional conditions should be to calculate the CC PSD for each stimulus and then compare the difference in their power between the emotional and neutral stimuli. For their claim to work, the distributions should not overlap, with all emotional stimuli yielding a higher PSD.
2. Lines 201 to 202. How does the PTA analysis show a theta phase difference between the eR and eKF conditions? Is it just because the CC is negative at 0-lag (suggesting the CCs are 180 degrees out of phase)? Wouldn't it be simpler to just calculate the difference in preferred amygdala theta phase for hippocampal gamma between the two conditions? The significance of this phase difference could be calculated with a permutation test (permuted across eR and eKF conditions).
3. Lines 209 to 212. The PACOi index is used to measure the difference in preferred phase between two conditions, but it seems like an oblique way to do so. I see how PACOi would be sensitive to the phase difference between conditions, but why not just measure the phase difference directly? You have a distribution of preferred phases for the eR and eKF conditions, so can't you apply the method I suggest in item #2. In general, I suggest you remove all PACOi analyses given that they only measure phase differences indirectly.
4. Lines 212 to 219. Here you measure what I suggested in item #3, but statistical significance is not provided. A permutation test would be appropriate.
5. Lines 235 to 237. How are PACOi results calculated 'per phase bin'? This needs to be explained better in the text.
6. Lines 290 to 294. This analysis seeks to demonstrate a directionality in gamma interactions between amygdala and hippocampus by calculating the CC between gamma burst peaks in the hippocampus and gamma power in the amygdala for two conditions and then taking their difference. Why not just take the difference in the peak lags for the two CCs? In addition, the only significant difference in Fig 4c was at 50 ms where the difference in the CC envelopes was reaching a minimal value and the SE had shrunk consequently. This segment of the CC is particularly sensitive to edge effects of the CC calculation since the epoched traces were only 60ms long.
7. The meaning of fig 4d is unclear. I have no idea how to interpret this.
8. Statistical validation of results are applied inconsistently. At the very least all major findings should be demonstrated at the subject population level, or within subject and the percentage of significant subjects provided.

Minor issues:

1. Line 601. Why demean before high pass filtering? Is it just to minimize the transient at the beginning of the trace when the filter receives zero padded values?

2. Page 62, line 123. PPC values can actually go below zero. See Vinck et al. 2010.
3. The discussion seems a bit skimpy. Perhaps the authors could further explore how their results relate to the physiology of theta phase dependent encoding/retrieval in CA1.
4. Lines 252 to 253. Isn't this expected given that the significant increases in gamma power in amygdala and hippocampus occupied different frequency ranges?
5. Lines 89. A citation showing that it is 'commonly observed' emotional stimuli have a higher remember false alarm rate would be helpful.
6. Given that the recollection and false recollection of emotional stimuli are both higher than neutral stimuli, would that suggest that emotional stimuli are remembered with less accuracy?

Reviewer #3 (Remarks to the Author):

The paper titled "Aversive memory formation in humans is determined by an amygdala-hippocampus phase code" sought to examine the directional connectivity of human amygdala-hippocampal interactions during aversive memory encoding. They found that emotional stimuli produce a unilateral influence from the amygdala to the hippocampus through theta oscillations and that memory for emotional stimuli depends on alignment of the hippocampus' gamma activity and neuronal firing to the amygdala's theta phase during encoding. Crucially, this interaction produces a transient lagged coherence that predicts of subsequent memory for emotional stimuli. Overall, they suggest that the precise theta phase interplay between the amygdala and hippocampus during emotional memory encoding might be indicative of a more general oscillatory mechanism through which the amygdala communicates emotional information throughout the brain.

This work represents a well-reasoned and critical next step in understanding emotion and memory interactions in the human brain. The authors utilize thoughtful and creative analyses to uncover the unique nature of amygdala hippocampal interactions during encoding of emotional images. Using a well-established aversive memory paradigm built on a rich literature in experimental animals and humans, the authors have produced truly novel data and analyses that have important implications for our understanding and developing treatments of memory and psychiatric disorders. I strongly support the publication of this article in Nature Communications, however a few minor issues should be addressed prior to publication. I've listed my suggestions for revisions by section and line below. Feel free to ask the editor for me to clarify any of these points if needed. Well done!

Comments

Introduction

1. Excellent introduction. No comments.

Results

1. Figure 1: Some statistical characterization of the temporal extent of the recollected emotional images vs the recollected neutral items for the hippocampus would be appreciated. The subsequent memory effect for gamma activity in the hippocampus seems to start more immediately and last longer for the recollected emotional vs neutral images.
2. It would be helpful to show a supplemental table that lists the number of trials in each emotion x memory condition.
3. Line 57: It seems like you're trying to tie the same time windows between amygdala gamma power in the amygdala (fig 1) and the changes in lagged granger causal theta activity. I think this could be made even more explicit by referring to the specific subpanel of figure 1 in addition to figure 2.
4. Figure 2 D: Please clarify if this is also in response to "viewing" emotional and neutral stimuli.
5. Figure 3 B right panel: Are these the approximate microwire locations in the hippocampus? Would be helpful to specify if so. It would also be helpful to specify the specific subregion of the hippocampus if possible. A subregion parcellation via ASHS would allow for some estimation of the hippocampal subfield. This could also just be estimated by comparison of each electrode's location to a known atlas. I'd suggest the Mai or Duvernoy atlases.

6. I appreciate the thoughtfulness put into connecting these analyses. Given the strong timing relationships between each of the phase, modulation, and transient coherence analyses, I think would be very helpful to include a schematic diagram that illustrates the relationship between each of these oscillatory features. This could be an idealized illustration of these oscillatory features. This could also be used as a graphical abstract of the studies novel findings.

Discussion

1. Line 107: Please also cite Bass et al., 2012, 2014, 2015 as further illustrations that theta-gamma stimulation to the amygdala is effective in rodents for organizing hippocampal responses and enhancing memory.
2. For citation 31, the direct amygdala stimulation occurred immediately following the emotionally neutral stimuli, rather than simultaneous to the stimuli.
3. Line 112: Given the clear role of recollection for differences in emotional memory findings in this study, please more clearly specify that these effects are tied to formation of strongly recollected emotional memories relative to familiar or forgotten stimuli. This is currently done parenthetically, but it's not immediately clear what is meant.
4. Line 122: This work also might have implications for therapeutic approaches of amygdala stimulation for memory disorders, in addition psychiatric disorders. I would recommend also mentioning this.
5. While I appreciate the brevity and conciseness of the discussion, if there is space, I would suggest adding more discussion. The schematic illustration of the papers related timing findings could be thoroughly discussed. I also think it would be helpful to further discuss the implications of this work as it relates to modulating or interfering with amygdala or hippocampal activity via direct brain stimulation. Finally, there could be more discussion of the limitations and future directions of this work.

Methods

1. Thank you for the thorough methods section and supplementary information on patients. Well done!

REVIEWER COMMENTS

We wish to thank all 3 expert Reviewers for their thoughtful and constructive comments, which have undoubtedly improved the quality of our manuscript. Here, we provide a point-by-point response to all comments. Our responses are in blue, with text that was in the original submission in blue italics, and new text in red.

Reviewer #1 (Remarks to the Author):

Costa et al. investigated electrophysiological responses and interactions between the amygdala and hippocampus during aversive and neutral stimuli, via human intracranial recordings. A larger literature, across species, has implicated these structures in memory formation and its emotional modulation. In their study, the authors employ a standard study/test recognition paradigm, using aversive (emotional) and neutral images. During these conditions, simultaneous invasive recordings from the amygdala and hippocampus were performed in two cohorts. In the second cohort, single unit activity was also recorded from the hippocampus. The authors report that both amygdala and hippocampus display increased gamma activity to successfully remembered aversive stimuli. In addition, spectral-granger analysis is reported to show directional influence of theta range activity between amygdala->hippocampus for aversive stimuli – but not theta coherence. Building on this influence, the authors finally report that hippocampal gamma and spiking activity is modulated by amygdala theta phase for aversive memory behavior – forming the basis of the title claim for evidence of phase coding. While several of the authors claims are well supported, the later observations of phase coding are more marginally supported by the data, particularly single unit findings. Further controls and caveats related to these observations should be provided.

We thank the Reviewer for the careful reading of our manuscript and insightful comments.

Main comments:

-As a general note, the authors should ensure that results reporting is always clearly conveying if the data being analyzed is from the encoding or recognition phase of the task throughout the manuscript. The authors do this in many places, but it might often occur several sentences later. Correcting this will benefit the reader.

R1.1. Action: Main text has been modified and we refer the Reviewer to the new Figure 5.

We apologise if this was unclear. We now state in the main text (line 106) “**All data reported here pertain to encoding-related responses.**”. As mentioned at line 107-110: “*we employed a subsequent memory approach, categorizing trials according to whether patients later remembered (R) the stimulus presented at encoding or not (known/forgotten items, KF) to operationalize successful encoding*”. Furthermore, in the new summary schematic figure of our reported findings, requested by Reviewer 3, we make it visually clear that our results pertain to encoding responses.

[REDACTED]

Fig. 5. Summary and time course of amygdala hippocampal activity and coupling during emotional memory formation. All data reported pertain to encoding-related responses (grey box). Peri-stimulus time is indicated by the downward arrow (left), with process onsets and durations (solid color bar) observed in the amygdala (pink), in the hippocampus (blue) or implying a dynamic between the two structures (pink and blue). Effects at encoding are divided (from left to right) based on whether we observed a significant effect for aversive scenes (emotion column), for subsequently remembered scenes (memory column), or are significantly discriminative for emotional vs. neutral memory formation (emotional memory column). At earliest latency, amygdala gamma activity increases more for subsequently remembered vs. not remembered emotional vs. neutral pictures. Next, amygdala-hippocampal PAC and Granger causal effects are associated with viewing aversive scenes as compared to neutral ones. The pink horizontal arrow represents the directionality of Granger causal influence from the amygdala to the hippocampus. Gamma activity in the hippocampus increased for stimuli that are later recollected independently of their valence. Hippocampal gamma and single-unit responses aligned to different amygdala theta phases during subsequently remembered vs. not-remembered (known/forgotten) emotional stimuli. The vertical rectangle represents the amygdala theta phase to which hippocampal gamma activity and neuronal firing couple for aversive scenes later remembered. If gamma activity in the hippocampus occurred during a different period of the theta cycle (outside the rectangle), aversive scenes were not subsequently remembered. Thick dashed line represents the transient lagged coherence between gamma activity in the amygdala and the hippocampus that is observed only for aversive scenes later remembered. The latency of around 37 ms is consistent with the phase difference between the two conditions around 25-45 ms. These results together point to an amygdala-hippocampus phase code for aversive memory formation.

-Related to this point, analyses comparing encoding and retrieval findings is limited (or if performed is hard to track). As the paradigm involves recognition of previously presented and new images, this provides an important point of comparison for responses to the same image, before and after learning. Given the focus on memory formation, and claims of evidence for a specific coding mechanism, comparing key findings at encoding and recognition of the same stimuli is critical. Does the same phase encoding occur during recognition? In addition to the points on surrogate data noted below, the recognition data provide a useful control dataset for showing the robustness of encoding data.

R1.2. Action: Main text has been modified to include future directions

The Reviewer raises an important point. It will be very interesting to determine how the amygdala and hippocampus interact during retrieval of emotional information, whether similar or completely novel oscillatory effects are observed, and whether there is reinstatement of patterns of activity displayed at encoding. However, the original submitted manuscript already included a total of 25 figures and 6 tables required to present a detailed, mechanistic account of emotional memory formation, to which we now add 5 new figures, and one new table in this revision. Doing the same with recognition data will, essentially double the current content. This would effectively be a second paper and, in our view, be beyond the major revision requested by the editor.

In the discussion we included the following lines as a future direction (see line 450-454):

Future studies may explore whether a similar phase offset is present at retrieval or whether a different phase relationship may exist between amygdala and hippocampus. As encoding and

retrieval has been shown to rely on complementary processes, the latter may be hippocampus-centered and thus may not rely on amygdala modulation^{12,50}.

We included this new reference in the reference list

50 Roozendaal, B., & McGaugh, J. L. (2011). Memory modulation. *Behavioral neuroscience*, 125(6), 797.

and refer to this paper already present in the original submission

12 Strange, B. A. & Dolan, R. J. β -Adrenergic modulation of emotional memory-evoked human amygdala and hippocampal responses. *Proceedings of the National Academy of Sciences*, doi:10.1073/pnas.0404282101 (2004).

-Trial numbers differ for the emotional (40) and neutral (80) stimuli. However, it's often not clear how this sample difference, or other data divisions (R vs. K trial#), are being accounted for when comparing electrophysiological data for these conditions. Spectral and certainly cross-frequency analysis will be influenced by such sample size differences (trial number).

R1.3. Action: we refer the Reviewer to Supplementary Table 6 and method session in the initial submission where this issue was already addressed.

We thank the Reviewer raising this point. Given that emotional items typically receive “old” responses at recognition more often than neutral items, the number of presented neutral items was larger in order to try to balance the number of subsequently remembered items. Supplementary Table 6 in the initial submission provides the number of analyzed trials per patient and conditions after artifact rejection. The method section in the original version of the manuscript addresses this issue. We apologize for the confusion, given the length of the Supplemental material it may not have been easy to find this information in the section “Phase to amplitude coupling: Modulation index (MI) and PACOI” (in the revised version of the manuscript at lines 663-670): *“As unequal trial numbers can confound phase estimates, we controlled for the unequal trial number as follows. For each patient, contact and experimental condition, the lowest number of trials was determined (N_{min}). From the other condition, a subset of N_{min} observations was randomly drawn from the observations constituting that particular experimental condition and the index of interest was computed. This procedure was re-computed 20 times by randomly selecting the subset of trials to the rest of the experimental conditions with higher number of trials. The controlled subsampled index was the average taken over the subsampled estimates”*.

-Related to this point, it's not always clear how multiple electrodes/units from individual subjects (nested data) are being handled. As this raise issues of independence in samples, multi-level model approaches or data collapsing must typically be employed.

R1.4. Action: we ran a hierarchical linear model analysis and include this result in new supplementary note 1.

We apologize for the lack of clarity regarding the statistics. All analyses collapsed over contacts that were anatomically limited to the amygdala or the hippocampus as previously adopted in (Mendez et al. 2016). The repeated measure is amygdala electrodes ($n=17$, 9 unilateral and 4

bilateral), hippocampus electrodes ($n=9$, 7 unilateral and 1 bilateral), or amygdala-hippocampus electrode pairs ($n=9$, 7 unilateral and 1 bilateral), respectively. In the present version of the manuscript, we now state this explicitly in the statistical analysis session (lines 589-594).

Statistical analysis. We analyzed electrophysiological responses from 17 amygdalae from 13 patients (five had left, 4 right, and 4 bilateral medial temporal electrodes in the amygdala) and 9 hippocampi from 8 patients (1 patient had bilateral hippocampal electrodes). All analysis collapsed over contacts (bipolar channels) that were anatomically limited to the amygdala or the hippocampus. Thus, statistical inference was based on a fixed effect approach as previously adopted⁵².

To supplement the fixed effects approach in our original submission, we have followed the suggestion of the Reviewer, and run a hierarchical linear model analysis. For this purpose, we used a random intercept model plus addition of Level 1 predictors following the formula $Y_{ij} = \gamma_{00} + \mu_{0j} + \gamma_{\text{sessions}} + e_{ij}$. The dependent variable is the mean gamma power extracted from the significant clusters obtained from time-frequency analyses in the amygdala and hippocampus, respectively. Level 1 predictors pertain to patients. We used a different dummy code if the recording pertains to different patients (for hippocampus and connectivity analysis this was the case for seven over eight patients), whereas we used the same index twice if the recording pertains to the same patient when bilateral electrodes are considered (this was the case for one over a total of eight patients). We found that amygdala and hippocampal responses were unrelated to recorded patient (amygdala interaction effect (eR - eKF) - (nR - nKF), $F_{(1, 14)}=2.4$, $P=0.14$ Fig. 1f); hippocampus main effect of successful encoding (R-KF), $F_{(1, 7)}=0.27$, $P=0.61$, Fig. 1i). These findings are in line with a fixed effects assumption, where the dependent variable is presumably constant across the units of observation. We include these results as Supplementary note 1, as follow:

*“In the amygdala (Fig. 1 d-f), successful encoding of aversive, but not of neutral scenes, was associated with fast gamma activity (97-125 Hz) starting 0.31 s after stimulus onset (emotion by subsequent memory interaction, summed t -value=1012.36, $P=0.01$, Fig. 1e). Subsequently remembered aversive scenes induced higher gamma power changes than subsequently known/forgotten aversive items (eR vs. eKF $t_{15}= 2.12$, $P=0.050$, $d=0.53$, post-hoc t -tests on mean power changes across significant time-frequency clusters, Fig. 1f, **Supplementary note 1**). Memory-related responses to neutral stimuli did not show a significant difference (nR vs. nKF, $t_{15}= -1.90$, $P=0.075$, $d=-0.47$).”*

*By contrast to the amygdala spectral power changes, hippocampal-induced responses (50-75 Hz from 0.54 s) were associated with subsequent recollection (R vs. KF) of both aversive and neutral pictures (Fig. 1 g-i; main effect of successful encoding, summed t -value=776.82, $P=0.004$, **Supplementary note 1**);.”*

Supplementary note 1 can be found in the last page of the Supplementary Material and we copy it below for Reviewer’s convenience:

Supplementary note 1

In the present study statistical inference was based on a fixed-effects approach. The repeated measure is amygdala ($n=17$, 9 unilateral and 4 bilateral electrodes) or hippocampus ($n=9$, 7 unilateral electrodes and 1 bilateral electrode), and amygdala-hippocampus pairs ($n=9$, 7

unilateral electrodes and 1 bilateral electrode). To complement the fixed effects approach we ran a hierarchical linear model analysis.

For this purpose, we used a random intercept model plus addition of Level 1 predictors following the formula $Y_{ij} = \gamma_{00} + \mu_{0j} + \gamma_{\text{sessions}} + e_{ij}$. The dependent variable is the mean gamma power extracted from the significant clusters obtained from time-frequency analyses in the amygdala and hippocampus, respectively. Level 1 predictors pertain to patients that correspond to different recordings or to the same patient when bilateral electrodes are considered. We found that amygdala and hippocampal responses were unrelated to recorded patient (mean gamma difference in the amygdala between (eR - eKF) and (nR - nKF), $F_{(1, 14)} = 2.4$, $P = 0.14$ Fig. 1f); mean gamma difference in the hippocampus (R-KF), $F_{(1, 7)} = 0.27$; $P = 0.61$, Fig. 1i). These findings are in line with a fixed effects assumption, where the dependent variable is presumably constant across the units of observation. Note that only one patient with bilateral recordings contributed to the connectivity statistics where amygdala-hippocampus pairs are used. Seizure onset was localized to the left precuneus and posterior cingulate in this patient, and not to the medial temporal lobe.

-Gamma effects are reported for several results, but the frequency range greatly differs. There are five different gamma ranges reported (97-125 Hz); (50-75 Hz); (50 -67 Hz); (50 -75 Hz); (60 -120 Hz); (80 -120 Hz). The authors should provide some interpretation of this variability and note it to the reader. Given the often low sensitivity of macro-depth electrodes in hippocampus/amygdala, these differences are most likely reflecting recording SNR, rather than unique functional signatures. These variable findings need to be reconciled with the specific analysis of a single range for gamma correlation later in the manuscript.

R1.5. Action: main text has been changed.

We thank the Reviewer for raising this point and agree with his/her assertion regarding SNR. Through the manuscript we now discuss gamma more clearly as a broadband activity and clarify that it is likely not a pure oscillation. The gamma band we found in our analysis is likely to be partly leaked from spiking activity. Since hippocampal spiking activity yields similar results to gamma activity in the context of locking to amygdala theta phase, this interpretation is plausible. Following the Reviewer's suggestion, we included the following paragraph at line 300-309.

“The frequency ranges of gamma activity differed between effects observed in the amygdala (80-120 Hz) and in the hippocampus (50-75 Hz). This variability most likely reflects differences in signal to noise ratio (SNR) suggesting that we are measuring a broadband gamma activity rather than an oscillatory activity. Moreover, cluster-based permutation tests³³ do not provide statistical inference for the exact latency and frequency of the effects³⁴. We selected time windows around the peaks of broadband gamma activity (60–120 Hz)^{27,28} - as frequency range overlapping between the two structures - ~~around the~~ in the hippocampal recordings. In a second step, we cut epochs around identified peaks (± 0.03 s) ~~gamma peaks for both hippocampus and amygdala~~ time-courses. Lastly, cross-correlations between ~~responses, and shifted these windows~~ were computed ~~the cross-correlogram~~ for successful and unsuccessful encoding of aversive scenes (Fig. 4a).”

Regarding the 60-120 Hz range we used for gamma correlation, this is the broadband gamma activity with frequency overlapping between the two structures. This cutoff frequency is also

supported by previous publications (Fedele et al. 2020; Kucewicz et al 2018) which are now quoted in the manuscript related with this aspect.

- 27 Fedele, T. et al. *The relation between neuronal firing, local field potentials and hemodynamic activity in the human amygdala in response to aversive dynamic visual stimuli. NeuroImage 213, doi:10.1016/j.neuroimage.2020.116705 (2020).*
- 28 Kucewicz, M. T. et al. *Electrical stimulation modulates high γ activity and human memory performance. Eneuro 5 (2018).*

-Appropriate surrogate data are critical for establishing chance levels with several of the analyses performed. It does appear the authors have employed such methods (i.e. through permutation testing), but the methods would benefit from more explicit details about specific permutation (what variable is being randomized) to ensure the inference of interest is being tested. In addition, the permuted/chance levels of metrics should be included in plots where relevant. As the authors know, small samples for many of the measured employed will always provide non-zero metrics. For example, the broadband gamma correlation analysis should include a randomization control, that is plotted with the observed data shown – providing better ground for the reliability of the effect shown.

R1.6. Action: permutation distributions have been added to Figure 3 c, h and Figure 4 d.

The Reviewer makes very helpful suggestions here. For statistical testing in all time frequency analyses, Granger causality analyses, and phase amplitude coupling analyses, we applied cluster-based permutation tests (Maris & Oostenveld, 2007) using the Monte Carlo method. We included more details in the methods session of the revised manuscript:

At lines 610-618:

“To correct the family wise error rate in the context of multiple comparisons across time and frequency dimensions, we applied a cluster-based permutation test⁴⁴ to the ensuing t-values obtained at each time-frequency bin to determine significant interactions or main effects in the time-frequency domain. We performed a 2-tailed t-test using a maximum summation cluster statistic Monte Carlo method. At each permutation step, selecting the a-priori time of interest in our data (from 0–1.5 s), clusters were formed by temporal and frequency adjacency with a cluster threshold of $\alpha=0.05$ and a paired t-test was calculated for high (35–150 Hz) and low frequencies (0–34 Hz) separately using a threshold of $\alpha=0.025$. Permutation steps were repeated 10000 times.”

For the PACOi analysis, the permutation testing description was already present in the original submission in the Methods sub-section titled *Phase-Amplitude Coupling Opposition index (PACOi)* “Statistical significance was computed by randomly shuffling ($n=1000$) the trial condition assignment (aversive remembered vs. aversive known/forgotten)⁶⁹” and now we include the histogram of the permutation values of the Watson Williams test in Figure 3c-h.

Fig. 3. Emotional memory-dependent amygdala phase opposition of hippocampal gamma and single neuron activity. **a**, Hippocampus gamma amplitude locks to different phase bins of amygdala theta depending on emotional memory outcome. Histogram of amygdala theta phases at which hippocampus broadband gamma amplitude occurs for eR (red) and eKF (black) trials as bar (left) and circular (right) plots. Inverted triangles represent the angle of preference in radians (eR:0.16 radians; eKF:1.83) after phase realignment. **b**, Circular plot, shaded areas in the circular plot represent PACOi results ($n=9$ electrodes) per phase bin ($n=20$). The analytic phase signal was divided in $n=20$ equal bins ($\phi_n = [-\pi, \pi]$) and the mean analytic amplitude was taken over those specific bins (see Materials and Methods at Phase to amplitude modulation index and Phase-Amplitude Coupling Opposition index (PACOi) sections) for each condition; max is the maximum value of phase-amplitude coupling for eR:1.19 and eKF:1.04. **c**, Histogram displays the permuted values for the Watson Williams test, red line represents the empirical statistic ($F_{ww}=187.49$, $P=0.00009$). **d**, Summary of electrode contact localization in the left and right amygdala (pink) and hippocampus (light blue) for patients included in the SFC analysis (Cohort 2: 6 patients and a total of 7 electrodes). The first contact is just lateral to the putative micro-wire location. **e**,

Example for a single subject showing a post-operative CT image, thresholded to visualize electrode contacts, co-registered with the corresponding pre-operative MRI scan in native space and superimposed onto the amygdala and hippocampus subfields. The sagittal view shows that contacts are in the amygdala and anterior hippocampus. The first coronal view shows hippocampal subfields and the white and black arrow point to the putative microelectrode location. The second coronal view shows amygdala nuclei and the two white and black arrows indicate the first two macroelectrodes locations. Subfields/nuclei are color coded. A schematic of the electrode implanted (1.3 mm diameter, 8 contacts of 1.6 mm length, and spacing between contact centers 5 mm; Ad-Tech, Racine, WI) is provided showing microwires protruding from the tip and macrocontacts (the first three contacts are numbered). **d-f**, As for **a**, but showing amygdala theta phases at which hippocampal spikes occur (average spike count over $n=31$ neurons. Inverted triangles represent the angle of preference in radians ($eR:0.18$; $eKF:1.40$) after phase realignment. **g**, Shaded areas in the circular plot represent spikes per the 20 phase bins for each condition. Maximum spike count for each condition is indicated ($eR:112$; $eKF:128$) as well as the total spike counts across neurons ($eR:1651$, $eKF:2030$). Upper subpanel: single-neuron waveform (mean \pm std) for one example neuron. **h**, Histogram displays the permuted values for the Watson Williams test, red line represents the empirical statistic ($F_{ww}=40.91$, $P=0.00009$). ~~Upper subpanel: single-neuron waveform (mean \pm std) for one example neuron.~~

Details about surrogate analysis used in the broadband gamma correlation analysis were already presented in the Methods session of the original submission (lines 780-783): “To test the strength of the correlation we performed a surrogate analysis in which we permuted the patients delay and peak envelope association and computed the Spearman rho after each shuffling ($n=10,000$). The observed rho was then compared to the permutation distribution. We now include in Fig. 4d the distribution of rho values from the permutation test. Red line represents the empirical statistic which exceeds the 95th percentile (black line on the right). The revised Fig. 4d is shown below.

d, Correlation between successful vs. unsuccessful emotional memory as measured as hippocampus-amygdala transient broadband gamma activity peak lag (x-axis) and hippocampus gamma delay as a function of amygdala theta cycle (y-axis); The x-axis represents the time lag measured by the peak of the cross-correlation (60-120 Hz) power envelopes between hippocampus and amygdala. The y-axis is the PACOi frequency-specific phase opposition result translated into a time lag. See Methods section for description of the transformation of PACOi frequency specific phase opposition into a time delay). The colored dots (red, blue, purple) represent 3 different patients for whom individual data is shown above and right (with corresponding frame and font color). In these 3 plots, left: time corresponds to the hippocampus-amygdala transient broadband gamma activity cross-correlation peak lag for eR vs. eKF trials (colored inverted triangle); right: hippocampus gamma delay as a function of amygdala theta cycle (the delay is the difference between black and red triangles,). Note the different sign between Patient s15 (green) and Patient s16 (purple) in the two correlated measures.

-For analysis of cross-frequency modulation between regions, a critical factor is the reliability of the specific frequency couplings. It would therefore be more important for the main figures to show comodulograms (currently in the supplement) and/or theta phase triggered spectrograms (not just triggered averages, for a given subject). Alternatively, a spectrogram of the triggered averages, rather than PSD, would better speak to specific coupling (see comments below on PSD).

R1.7. Action: Figure 2 has been changed.

We appreciate the Reviewer's suggestion and Figure 2 has been changed accordingly. In panel e, we now include the between-region phase to amplitude coupling comodulogram showing a significant difference in modulation strength (Modulation Index) between emotional and neutral items. In panel f, we include the statistical result for the emotion by successful encoding

effect. The absence of an interaction indicates that phase amplitude coupling between amygdala theta and hippocampal gamma is unrelated to memory. We kept the peak triggered average (PTA) results and clearly labeled them as coming from an example patient and now display PSD with a bigger x-axis. PTA analysis in panel **i** importantly shows that the magnitude of gamma amplitude is concentrated at different amygdala theta phase bins for subsequently remembered *vs.* known/forgotten emotional stimuli. Indeed, negative (or positive) values indicate opposing theta phase locking between conditions. We included further details in the figure legend provided below.

“Fig. 2. Emotion-dependent amygdala-hippocampus theta-gamma coupling. **a**, Granger causal influence of amygdala on hippocampal oscillations for aversive *vs.* neutral scenes (main effect of emotion). **b**, Mean Granger values for the significant cluster is plotted for each amygdala-hippocampus electrode pair (8 patients and a total of 9 electrode pairs). Horizontal/vertical bars represent mean \pm s.e.m. **c**, As for **a**, but in the direction hippocampus to amygdala. **d**, Amygdala theta phase (3–12 Hz) to hippocampus gamma amplitude (50–75 Hz) coupling is greater in response to **viewing** emotional *vs.* neutral stimuli. Data points represent each pair of electrode contacts located in amygdala and hippocampus. **e**, **Inter-**

regional phase-to-amplitude coupling comodulograms (calculated using the Modulation Index method), with amygdala phase on the *x*-axis and hippocampus gamma amplitude on the *y*-axis, showing group level statistics for the main effect of emotion (summed *t*-value=104.95, *P*=9.99 x 10⁻⁵). **f**, as for **e**, but showing the absence of a significant emotion by successful encoding effect (eR-eKF vs. nR-nKF). **g**, *Inter-regional cross-correlation between emotional and neutral (dark blue/dark red) peak triggered average (PTA) (left) and the corresponding power spectral density (PSD) of the cross-correlogram (right) for Patient 16. A theta component is evident in the PSD and fitted sine wave (thick black line). h*, Same representative patient, within-hippocampus cross-correlation between emotional and neutral (blue/red) conditions. PTAs were computed as in **e** but averaging the raw local field potentials from the hippocampal recordings. **i**, Same representative patient, PTA for aversive remember vs. know/forgotten (red/black) trials using the peaks of hippocampus gamma activity to average amygdala raw recordings. At zero lag, cross-correlogram negative values indicate that hippocampus gamma bursts during eR and eKF trials are locked to opposing amygdala theta phase bins. By contrast, cross-correlogram values equal to 1 would indicate an alignment between signals of the two conditions.”

We also changed the main text as follows (see lines 196-217):

“The degree to which single neuron spiking phase-locks to human hippocampal theta oscillations is a predictor of memory strength^{25, 26}. Given that spiking has been shown to correlate with gamma power^{27, 28}, we hypothesized that a stronger modulation of hippocampal gamma band activity by amygdala theta oscillations would occur during encoding of aversive scenes leading to remembering. We thus tested the phase-amplitude coupling (PAC) associated with subsequently remembered vs. non-remembered emotional items by calculating the modulation index (MI) (Supplementary Extended Data Figs. 11-14)²⁹. Amygdala theta phase coupling with hippocampal gamma amplitude was *on average* stronger for viewing emotional compared to neutral scenes ($F_{(1, 8)}=6.73$, $P=0.031$, $\eta^2=0.457$, Fig. 2d-e), similar to previous observations during passive viewing of emotional (fearful) faces vs. neutral landscapes¹⁶. However, PAC was unrelated to memory formation (i.e., MI values did not show a significant emotion by memory interaction, Fig. 2f).

~~This result was further corroborated using~~ Using a complementary approach we which isolated hippocampal gamma bursts in each condition during successful and unsuccessful encoding of aversive scenes and used them to trigger averages in amygdala raw field potentials. The cross-correlogram (CC) obtained from the peak-triggered averages (PTA) resulting from emotional vs. neutral stimuli confirmed that hippocampal gamma peaks locked to ongoing amygdala theta oscillations following all emotional stimuli and did not show phase difference between emotional and neutral stimuli (Fig. 2ge, for one representative patient, and Supplementary Extended Data Fig. 15 for remaining patients), in alignment with the theta-gamma PAC results derived via the MI approach (Fig. 2d,e). The same relationship was observed between hippocampal theta and gamma (Fig. 2hf, Supplementary Extended Data Fig. 17). At zero lag, cross-correlogram values equal to 1 indicate the same amygdala phase for hippocampal gamma bursts across conditions. By contrast, negative values at zero lag indicate gamma bursts locking to opposing theta phase bins.

The amygdala theta phase to which hippocampal gamma activity and neuronal firing couple determines subsequent remembering of aversive events.

Comparing peak-triggered averages for successful (eR) vs. unsuccessful encoded (eKF) aversive scenes revealed a theta phase difference between the two conditions (Fig. 2i), an effect observed across all patients (Supplementary Fig. 17)."

-PACOi. While these LFP results suggests some phase consistent offset – it does not support the authors statement that: “This result established a phase code for amygdala theta-to-hippocampus gamma coupling in determining aversive memory formation in humans”.

R1.8. Action: we clarify the phase code interpretation and update the main text.

This is an important point and we welcome the opportunity to further clarify the phase-code interpretation, which follows John Lisman’s (2005) definition regarding place cell coding in the hippocampus: “a corollary of theta phase coding is that ensembles representing nearby places should fire with a fixed average phase difference, irrespective of the particular position of the rat (so long as the rat is in both place fields). The specific prediction is that two cells coding for slightly different positions should show a cross-correlation with a peak offset equal to a significant fraction of theta.”

We updated the main text (from lines 237-242) and included a few sentences to explain how our results fit with a phase coding definition.

“This result established a phase code for amygdala theta-to-hippocampus gamma coupling in determining aversive memory formation in humans. We found that theta rhythm provides a time reference that allows assignment of specific emotional memory encoding contents indexed by the phase difference at which gamma activity locks to theta. These results are in line with a theta/gamma model as a general brain coding scheme³².”

32 Lisman, J. (2005). The theta/gamma discrete phase code occurring during the hippocampal phase precession may be a more general brain coding scheme. Hippocampus, 15(7), 913-922.

-As shown for LFP data, basic task event-related raster plots of spiking data should be shown. This provides basic context for interpreting the presented oscillatory modulation of firing rates in hippocampus. Only showing collapsed cross-regional modulation of spiking limits the ability to adjudicate if meaningful single unit activity was captured (often a challenge for these human recordings).

R1.9. Action: Two new figures have been added to the Supplementary material.

We thank the Reviewer for the suggestion. We now include the raster plot for hippocampal spikes for each unit ($n=31$) for both eR and eKF conditions separately, and the firing rate over all units for each condition across peri-stimulus time as Supplementary Fig. 22. We further include an example of one trial for one patient showing amygdala phase and hippocampal spikes from -0.5 to 1.5 seconds for one eR and one eKF trial as Supplementary Fig. 23.

“Supplementary Fig. 22: Event-related raster plot showing hippocampal spikes over peri-stimulus time (x -axis) for eR (left column) and eKF trials (right column) for each unit ($n=31$). Units are sorted in descending order as a function of spike number across trials for eR condition. Dotted vertical line represent the stimulus onset. Black vertical lines represent the post stimulus time interval (from 0.41 to 1.1 s) on which PACOi and SPC analysis were performed. (Bottom) Histogram showing the mean firing rate for all units over time for eR (left column) and eKF (right column) conditions.”

“Supplementary Fig. 23: One example trial for eR and one example trial for eKF (patient Z5) showing the occurrence of hippocampal spikes from one unit in relation with the ipsilateral amygdala theta phase.”

-Related to the above, the spiking data show much weaker support for signs of phase coding. Why would a population measure, like gamma, present much clearer phase coding evidence than specific neurons (the actual code)?

R1.10. Action: the main text has been updated.

We thank the Reviewer for raising this. Throughout the manuscript we discuss broadband gamma activity as a surrogate of spiking activity. However, the number of neurons recorded in this study was not very high, which may explain why phase coding was clearer using broadband gamma than single unit results. We add the following to the main text to make the Reviewer’s point more explicit:

Line 243-244,

*“In analyses of data from Cohort 1, we considered **broad band** gamma activity as an index of spiking activity^{27,28}”*

Lines 247-254

*“Overall, we observed that hippocampal spikes (from $n=31$ neurons) during successful vs. unsuccessful encoding of aversive scenes concentrated at different theta bins with a consistent phase difference of ~ 1.22 radians (two sample Kolmogorov-Smirnov test, $P=0.00072$), corresponding to ~ 26 ms time lag. At the single-unit level, the firing distribution of 6 neurons differed significantly (circular Kuiper test, Supplementary Fig. 21) between the two conditions, more than expected by chance (binomial test versus 5% chance, $P < 0.01$, Fig. 3f-h). **Note that***

capturing single unit activity is often a challenge in the context of human recordings. Thus, results may be limited by the low number of units captured during recording.”

Minor comments:

-For clarity, in the introduction (line 70), when the microelectrode cohort is mentioned, the anatomical site of recordings should also be mentioned, similar to how cohort 1 is described (i.e. amygdala and/or hippocampus, ipsilateral?).

R1.m1. We added the following details at lines 72-73:

“Cohort 2 comprised patients with microelectrode recordings implanted in the amygdala and in the ipsilateral hippocampus (Fig. 3d), permitting analysis of single neuron activity”.

-Behavioral results (line 92), report performance for ‘known’ items, but it’s not clear what the reported values reflect (mean number of items?), as the total number of items is yet to be mentioned, it’s hard to gauge the level of performance (e.g. ratio or % values).

R1.m2. Action: Supplementary Table 3 has been included and the main text has been updated.

We have now made this point clearer.

For each participant we calculated the performance for known responses for emotional and neutral items as follows:

- number of emotional known responses divided by the total number of emotional stimuli seen at encoding (40) multiplied by 100.
- number of neutral known responses divided by the total number of neutral stimuli seen at encoding (80) multiplied by 100.

We now specify that values refer to percentage and include in the legend of Fig. 1b (line 127-129) the following information:

“For eR items, 100% indicates correct remember responses to all previously presented emotional pictures and zero false alarms to new emotional pictures.”

In the revised version of the manuscript, we now include the raw trial number for stimulus types as Supplementary Table 3.

Supplementary Table 3. Behavioral data from 18 patients (Cohorts 1 and 2). Raw trial number for emotionally aversive (e) and neutral (n) correctly remembered, remembered false alarms, correctly known and known false alarms (R, RFA, K, KFA, respectively) responses for all patients (n=18). Individual patient performance plotted in Fig. 1b is expressed in percentage calculated as number of items reported here divided by the total number of either emotional (40) or neutral (80) stimuli seen at encoding multiplied by 100.

Patient	Re	ReFas	eK	eKFas	nR	nRFas	nK	nKFas
02	7	1	21	15	10	0	36	14
04	20	3	5	7	40	13	19	22
06	10	1	13	13	11	3	25	9
13	27	21	1	3	47	16	9	10
15	19	10	7	7	19	9	16	19
16	19	12	8	9	23	11	21	22
21	24	11	6	9	48	9	12	14
25	16	3	15	13	25	6	29	25
27	17	13	14	8	32	14	9	13
32	25	10	8	15	38	14	19	26
33	21	10	12	8	20	1	12	8
34	21	15	11	17	4	3	21	15
Z1	24	24	6	5	22	16	8	16
Z2	4	2	0	0	9	4	1	0
Z5	15	7	13	13	24	6	18	16
Z6	12	1	5	8	33	7	16	14
Z8	4	4	18	21	12	3	23	11
Z10	9	10	1	1	3	2	2	1
Mean	16.3 (1.69)	8.77 (1.59)	9.11 (1.39)	9.55 (1.31)	23.33 (3.29)	7.61 (1.25)	16.44 (2.12)	14.16 (1.69)

-Why are spectrograms shown limited to higher-frequency ranges only? Also, spectrogram ‘power’ color bars require units.

R1.m3. Action: we refer Reviewer to Supplementary Figure 4. Figure 1 has been changed.

We found no significant condition effects in the lower frequencies for any of the comparisons performed in the amygdala and hippocampus. We mention this in the original submission, in the Results section (please see revised version of the manuscript at lines 144-145): “Regarding lower oscillatory frequencies (1–34 Hz), significant condition differences were not evident in either structure (Supplementary Fig.-45)”.

As reported in the Methods section, lines 575-577,

Spectral analysis. Time-resolved spectral decomposition was computed for each trial using 7 Slepian multi-tapers for high frequencies (> 35 Hz) and a single Hanning taper for low frequencies (≤35 Hz).

We thus plot high and low frequency separately. The Reviewer can find the low-frequency plots reported below which was included in the original submission of the manuscript. Regarding the missing unit, we apologise for omitting this important information. We inserted the label ‘relative power’, since values represent the power change relative to the mean taken

over the baseline period (*i.e.*, a value of 0.1 means 10% increased power), as can be seen in the revised version of Figure 1 reported below. We further changed all supplementary figures accordingly (please see the revised version of the supplementary material).

[REDACTED]

Fig. 1. Fast amygdala gamma activity tracks enhanced emotional memory formation, whereas hippocampal gamma activity increases during emotional and neutral memory formation. **Aa**, Task design. Examples of aversive and neutral scenes presented during the experiment. **bB**, Recognition performance for emotionally aversive (e) and neutral (n) correctly remembered, remembered false alarms, correctly known and known false alarms (R, RFA, K, KFA, respectively) responses for all patients (n=18). Individual patient performance is plotted (horizontal/vertical lines indicate the mean \pm s.e.m). For eR items, 100% indicates correct remember responses to all previously presented emotional pictures and zero false alarms to new emotional pictures. **c**, Summary of electrode contact locations for cohort 1 in the left and right amygdala (pink) and hippocampus (light blue) for all patients included in the analysis (n=13 patients; 17 amygdala (A) electrodes; n=8 patients; 9 electrodes in both structures (A-H)). **d**, Time-frequency plots of amygdala gamma power change for aversive and neutral subsequently remembered, and not remembered (known/forgotten) trials. **e**, Time frequency resolved test statistics for the emotion by successful encoding effect. Black outline indicates the significant cluster. **f**, Mean amygdala gamma power, relative to baseline, in the significant cluster for each amygdala electrode. **g**, Time-frequency plots of hippocampus power change in the gamma range for aversive and neutral subsequently remembered KF trials. **h**, Time frequency-resolved test statistics showing the main effect of successful encoding (R vs. KF). **i**, Mean hippocampus gamma power, relative to baseline, in the significant cluster.

Supplementary Fig. 5. Stimulus-induced low frequency oscillatory power during encoding in amygdala and hippocampus (Cohort 1). Low frequency oscillatory relative power increases, primarily in the delta and theta ranges, are observed in all conditions in both **a**, Amygdala and **b**, Hippocampus. In neither structure did we observe an emotion by subsequent memory interaction, or main effect. eR: subsequently remembered and eKF know/forgotten aversive pictures; nR subsequently neutral remembered and nKF known/forgotten neutral trials. The colorbar indicates power change relative to baseline.

-Why are different model tests used for Figure 1 (E; amygdala) and (H; hippocampus)?

R1.m4. Action: we refer the Reviewer to the methods section of the original submission where this issue was addressed.

We show only significant effects. In the amygdala we observed a significant emotion by successful encoding effect while in the hippocampus we observed a main effect of memory (successful encoding). We described this in the original manuscript in the Methods section (please see lines 601-609 in the revised version of the manuscript): “We focused our analyses

on encoding responses to subsequently remembered vs. known/forgotten items. In both amygdala and hippocampus, we first tested for the interaction of emotion by successful encoding (aversive remember (eR) – aversive known/forgotten (eKF)) vs. (neutral remember (nR) – neutral known/forgotten (nKF)). Power interactions were computed using 8 bipolar channels obtained from 7 patients. One patient was excluded (Patient Z1) because only 2 trials in the neutral remembered condition were obtained. In absence of a significant interaction, we tested for the main effect of emotion (aversive – neutral), and the main effect of successful encoding (remember – known/forgotten).”

-Conditions in Figure 1 (i) are misaligned to labels.

R1.m5. Corrected with thanks. We changed the figure accordingly, please see **R1.m3**.

-Figure 2 (e-g) should be clearly labeled as coming for an example subject. Also, why are PSD plots show with such a narrow x-axis base? Limits the purpose of displaying this data and over emphasizes spectral shape.

R1.m6. Action: we changed Supplementary Figure 15, 16, 17 and Figure 2 g-i.

We thank the Reviewer for this remark, and have changed the figure label accordingly. As requested by the Reviewer, we now show PSD plots with a larger x-axis as can be seen in the edited Supplementary Fig. 15, 16 and 17 (reported below) and in Figure 2 g-i. Please see the updated Figure 2 in **R1.7**

Extended Data Supplementary Fig. 14 15. Inter-regional cross-correlation between emotional vs. neutral peak triggered average (PTA; blue/red) for each patient's amygdala-hippocampus pair (Cohort 1). Each PTA was computed by first filtering hippocampus gamma

activity (50–75 Hz), then identifying gamma peaks (minimum separation 0.1 s) and finally averaging the raw traces from amygdala recordings (± 0.12 s) centered ($t=0$ s) around the hippocampus gamma peaks (see Materials and Methods). Each subplot represents a lateral amygdala and lateral hippocampus bipolar channel pair. Bottom insert for each PTA: power spectral density (PSD) taken over the entire cross-correlogram to find the main spectral component that dominates the PTA. The PSD peak was used to fit the optimal sine wave (black), in the least squares sense. Note that all amygdala traces show a theta component reflected in the PSD and in the fitted (black) sine wave. All reported main effects were computed including lateral bipolar channels from all patients with simultaneous hippocampus and amygdala recordings ($n=8$ patients, 9 amygdala-hippocampal electrode pairs) during 0.41–1.1 s post-stimulus onset.

Hippocampus γ 50-75Hz -Hippocampus

— Fitted sine function
 — Emotional vs Neutral

Extended–Data Supplementary Fig. 16 15. *Single contact within-hippocampus cross-correlation between emotional vs. neutral peak triggered average (PTA; dark blue/dark red) (Cohort 1). Each PTA was computed by filtering hippocampus gamma activity (50–75 Hz), then identifying gamma peaks (minimum separation 0.1 s) and finally averaging the raw traces from the same hippocampus recordings (± 0.12 s) centered ($t=0$ s) around the peaks (see Materials and Methods). Each subplot represents a lateral hippocampus bipolar channel. Bottom insert: power spectral density (PSD) taken over the entire cross-correlogram to find the main spectral component that dominates the PTA. The PSD peak was used to fit the optimal sine wave (black) in the least squares sense. Note that all hippocampal traces show a theta component reflected in the PSD and in the fitted (black) sine wave. All reported main effects were computed including lateral hippocampal bipolar channels from all patients with simultaneous hippocampus and amygdala recordings ($n=8$ patients, 9 hippocampal electrodes) during 0.41–1.1 s post-stimulus onset.*

Extended Data Supplementary Fig. 17 16. Inter-regional cross-correlation between emotionally remembered (eR) vs. emotionally known plus forgotten (eKF) peak triggered

average (PTA; red/black) (Cohort 1). Each PTA was computed by filtering hippocampus gamma activity (50–75 Hz), then identifying gamma peaks (minimum separation 0.1 s) and finally averaging the raw traces from amygdala recordings (± 0.12 s) centered ($t=0$ s) around the hippocampus gamma peaks (see Materials and Methods). Each subplot represents a lateral amygdala and lateral hippocampus bipolar channel pair. Bottom insert: power spectral density (PSD) taken over the entire cross-correlogram to find the main spectral component that dominates the PTA. The PSD peak was used to fit the optimal sine wave (black), in the least squares sense. Note all amygdala traces show a theta component reflected in the PSD and in the fitted (black) sine wave. All reported simple effects (eR vs. eKF) were computed including lateral bipolar channels from all patients with simultaneous hippocampus and amygdala recordings ($n=8$ patients; 9 amygdala-hippocampal electrode pairs) during 0.41–1.1 s post-stimulus onset.

Reviewer #2 (Remarks to the Author):

This paper investigates the oscillations in the amygdala and hippocampus during the encoding of emotional and neutral stimuli. The data comes from human subjects with intracranial depth electrodes, making it especially compelling. Subjects are run on a standard memory task that has been used in numerous other studies, so the results are relevant to a large literature.

We thank the Reviewer for their positive evaluation of our manuscript.

While a wide variety of analytical techniques are employed, there are only a few key findings. First, high gamma activity in the amygdala is enhanced during emotional stimuli that are later recollected compared with those that are forgotten or neutral. On the other hand, mid-gamma activity in the hippocampus is strongest for recollected stimuli irrespective of their valence. Second, theta band activity in the amygdala preferentially entrains theta and gamma activity in the hippocampus during the encoding of emotional stimuli. Third, emotional and neutral stimuli drive differences in the entrainment of hippocampal gamma to amygdala theta. These top-line results are intriguing, but the underlying analyses that the authors use to arrive at them are sometimes confusing, round-about, or indirect. Indeed, the paper would benefit from a revision of some of its analyses (and exclusion of others). Below I will outline the problematic sections and offer alternatives that, in this reviewer’s opinion, are more straightforward.

We thank the Reviewer for the insightful comments. We have revised the manuscript to more explicitly explain the rationale behind the methodology used in this study.

1. Line 192 to 195. It is unclear what is demonstrated by Fig. 2e. The authors claim that the cross-correlation in Fig. 2e shows that hippocampal gamma is preferentially phase locked to amygdala gamma for ‘all emotional stimuli.’ I am not sure how this figure shows that. The PTA method here is that the cross correlation (CC) is calculated between gamma burst peaks in the hippocampus and raw LFP signals in the amygdala. Then, the cross-correlation is calculated between these two cross-correlation functions and the PSD of it indicates the strength of entrainment. While the figure certainly exhibits phase locking between amygdala theta and hippocampal gamma, it does not say anything about whether it is stronger for emotional or neutral stimuli, let alone whether it applies to ‘all emotional stimuli.’ Wouldn’t just switching the CCs between the neutral and emotional conditions yield a similarly strong PSD? The direct test of the difference between neutral and

emotional conditions should be to calculate the CC PSD for each stimulus and then compare the difference in their power between the emotional and neutral stimuli. For their claim to work, the distributions should not overlap, with all emotional stimuli yielding a higher PSD.

R2.1. Action: Methods section about PTA analysis has been updated and Figure 2 has been changed.

We thank the Reviewer for raising this point. The Reviewer is right regarding the logic of the proposed analysis. However, since PAC-correlation metrics are scale-invariant, there's a serious limitation about how to relate PAC intensity to its slope (see Tort et al 2010 Fig. 7B). One can obtain exactly the same slopes by multiplying the amplitude envelope with a linear transform: $\text{corr}(X, Y) = \text{corr}(X, C*Y+D)$, with X representing the phase of an oscillation, Y the amplitude envelope of another oscillation and C and D being different constants. Furthermore, PAC-correlation metrics are more sensitive to noise relative to Tort's MI (see Tort et al 2010 Fig. 9). The reason we used CC was to show how different patients showed different lags in the coupling. This feature is unique for CC measures and MI does not measure time lags. In sum, we used MI to measure PAC strength and we complemented the MI strength analysis with the CC plots to display the time lag of the coupling.

Please also note that Figure 2 has changed due to requests from the Reviewers. The degree of phase locking (PAC) between conditions is calculated using the Modulation Index method. We now include the between-region phase to amplitude coupling comodulogram showing a significant difference in modulation strength between emotional and neutral items as panel e. The revised Figure 2 is shown below.

“Fig. 2. Emotion-dependent amygdala-hippocampus theta-gamma coupling. a, Granger causal influence of amygdala on hippocampal oscillations for aversive vs. neutral scenes (main effect of emotion). **b,** Mean Granger values for the significant cluster is plotted for each amygdala-hippocampus electrode pair (8 patients and a total of 9 electrode pairs). Horizontal/vertical bars represent mean \pm s.e.m. **c,** As for **a,** but in the direction hippocampus to amygdala. **d,** Amygdala theta phase (3–12 Hz) to hippocampus gamma amplitude (50–75 Hz) coupling is greater in response to **viewing** emotional vs. neutral stimuli. Data points represent each pair of electrode contacts located in amygdala and hippocampus. **e,** **Inter-regional phase-to-amplitude coupling comodulograms** (calculated using the Modulation Index method), with amygdala phase on the x-axis and hippocampus gamma amplitude on the y-axis, showing group level statistics for the main effect of emotion (summed t -value=104.95, $P=9.99 \times 10^{-5}$). **f,** as for **e,** but showing the absence of a significant emotion by successful encoding effect (eR-eKF vs. nR-nKF). **g,** **Inter-regional cross-correlation between emotional and neutral (dark blue/dark red) peak triggered average (PTA) (top) (left) and the corresponding power spectral density (PSD) of the cross-correlogram (bottom) (right) for Patient 16.** A theta component is evident in the PSD and fitted sine wave (thick black line). **hf,** Same representative patient, within-hippocampus cross-correlation between emotional and neutral (blue/red) conditions. PTAs were computed as in **e** but averaging the raw local field potentials from the

hippocampal recordings. **ig**, Same representative patient, PTA for aversive remember vs. know/forgotten (red/black) trials using the peaks of hippocampus gamma activity to average amygdala raw recordings. At zero lag, cross-correlogram negative values indicate that hippocampus gamma bursts during eR and eKF trials are locked to opposing amygdala theta phase bins. By contrast, cross-correlogram values equal to 1 would indicate an alignment between signals of the two conditions.”

We changed the main text as follows at lines 196-206

The degree to which single neuron spiking phase-locks to human hippocampal theta oscillations is a predictor of memory strength^{25, 26}. Given that spiking has been shown to correlate with gamma power^{27, 28}, we hypothesized that a stronger modulation of hippocampal gamma band activity by amygdala theta oscillations would occur during encoding of aversive scenes leading to remembering. We thus tested the phase-amplitude coupling (PAC) associated with subsequently remembered vs. non-remembered emotional items by calculating the modulation index (MI) (~~Extended Data~~ **Supplementary Figs. 10-13** **11-14**)²⁹. Amygdala theta phase coupling with hippocampal gamma amplitude was, **on average**, stronger for **viewing** emotional compared to neutral scenes ($F_{(1, 8)}=6.73$, $P=0.031$, $\eta^2=0.457$, Fig. 2d-e), similar to previous observations during passive viewing of emotional (fearful) faces vs. neutral landscapes¹⁶. However, PAC was unrelated to memory formation (i.e., MI values did not show a significant emotion by memory interaction, **Fig. 2f**).

We apologise for the lack of clarity in the methods section about the peak-triggered average analysis. We would like to stress that the result obtained from cross correlations between PTAs (hippocampus gamma - amygdala raw) comparing aversive remember vs. known/forgotten trials was the first evidence that hippocampal gamma bursts during successful vs. unsuccessful encoding of aversive scenes locked to different amygdala theta phases. This result motivated us to develop the PACOi method to formally quantify the strength of phase opposition between these two conditions.

Regarding the Reviewer’s last point, it is correct that switching CCs between emotional and neutral stimuli would obviously give the same PSD. In the revised manuscript, we have now clarified that the inference of stronger phase locking for emotional compared to neutral trials is based upon the Modulation Index results.

We improved the text description as follows:

Line 737-752

~~“Inter-regional peak-triggered averages (PTAs).~~ **Inter-region peak-triggered averages (PTA)**. Cross-correlations between the averaged PTAs of two conditions served to illustrate the phase-offset between the PTA of condition A and the PTA of condition B (Fig. 2.g-i). ~~We computed PTA as a complementary way to quantify the intra and inter regional PAC.~~ To compute PTAs, ~~First, we band-pass filtered the hippocampal gamma activity between 50-75 Hz from a particular brain region (amygdala/hippocampus) using a two-pass finite impulse response (FIR) filter with a filter order of 3 cycles in the lowest frequency bound. Peaks were detected (0.1 s minimum inter-peak distance) in this filtered signal and time intervals of ± 0.12 s around these peaks were used to average the raw traces (z-scored) over the same region (hippocampus gamma - hippocampus raw PTA, Supplementary Fig. 16) or between regions, taking amygdala raw trace (hippocampal gamma - amygdala raw PTA, Supplementary Fig. 15-17) different brain regions.~~ Once averaged, the PTAs were de-trended.

~~Cross-correlations between the averaged PTAs of two conditions served to compare PTAs from two conditions (Fig. 2 e-g and f Extended Data Fig. 14-16). Power spectral density (PSD) of the cross-correlated PTAs was computed to find the main spectral component that dominates the PTA (PSD peak; f_{peak}). For example, a cross-correlation between the averaged PTAs of two conditions with a theta sine wave component with a peak offset at time lag zero would be in line with a phase-opposition resulting from gamma activity peaks aligned at different theta phase bins in condition A and B. Best sinusoidal fit was computed over the cross-correlated PTA with frequency, phase, amplitude and offset as free parameters. The frequency parameter was constrained selecting the frequency range of $\pm f_{peak}/4$.~~

2. Lines 201 to 202. How does the PTA analysis show a theta phase difference between the eR and eKF conditions? Is it just because the CC is negative at 0-lag (suggesting the CCs are 180 degrees out of phase)?

R2.2. Action: main text has been updated.

Indeed, at zero lag, negative CC values indicate that hippocampus gamma bursts during eR and eKF trials are locked to opposing theta phase bins. We included this information in the legend of Figure 2i (lines 191-195).

“ig, Same representative patient, PTA for aversive remember vs. know/forgotten (red/black) trials using the peaks of hippocampus gamma activity to average amygdala raw recordings. At zero lag, negative CC values indicate that hippocampus gamma bursts during eR and eKF trials are locked to opposing theta phase bins. By contrast, CC values equal to 1 would indicate an alignment between signals of the two conditions.”

Wouldn't it be simpler to just calculate the difference in preferred amygdala theta phase for hippocampus gamma between the two conditions? The significance of this phase difference could be calculated with a permutation test (permuted across eR and eKF conditions).

Action: we refer Reviewer to Supplementary Figure 18 in the original submission where an equivalent analysis was already performed.

In the original submission we performed an equivalent analysis as the one proposed by the Reviewer. In the PACOi analysis we found patient-specific broadband gamma activity sub-bands (similar to the one specified by the cross-correlation).

Supplementary Fig. 1819. Phase-Amplitude Coupling Opposition index (PACOi) for aversive remembered and aversive known/forgotten trials for individual amygdala-hippocampal electrode pairs (Cohort 1). For each pair, left: comodulogram showing the amygdala phase (x-axes) to hippocampus amplitude (y-axes) phase opposition between emotional remembered (eR) vs. emotional know + forgotten (eKF) trials. Colorbar indicates the PACOi values (see Materials and Methods, fig. S13) masked by (uncorrected) statistical contrast between the eR vs. eKF group of trials. For each amygdala-hippocampal electrode pair, right: averaged hippocampus amplitude to amygdala phase histogram resulting from the significant (masked comodulogram) PACOi. Red represents eR and black eKF. PACOi analyses were performed over the 0.41–1.1 s post-stimulus time interval.

We assessed the delay-specific theta-gamma phase differences employing the formula already reported in the Method section “Relationship between within vs. cross-frequency hippocampus-amygdala coupling”:

$$\text{delay} = \frac{\left(\text{angle} \left(z_{eR}^{f_x, f_y} \right) - \text{angle} \left(z_{eKF}^{f_x, f_y} \right) \right)}{(2\pi)} * \left(\frac{1}{f_{xc}} \right)$$

Looking at the y-axis of Figure 4d (in the original submission and reported below) where this result is reported, the Reviewer will see that some patients showed a positive lag and others a negative lag, all but one, are different from zero showing that the phase difference between

conditions is present in most of our measurements. Please see the revised version of Fig 4d. Further details can be found in our response to point R2.7

d, Correlation between successful vs. unsuccessful emotional memory as measured as hippocampus-amygdala transient broadband gamma activity peak lag (x-axis) and hippocampus gamma delay as a function of amygdala theta cycle (y-axis); The x-axis represents the time lag measured by the peak of the cross-correlation (60-120 Hz) power envelopes between hippocampus and amygdala. The y-axis is the PACOi frequency-specific phase opposition result translated into a time lag. See Methods section for description of the transformation of PACOi frequency specific phase opposition into a time delay). The colored dots (red, blue, purple) represent 3 different patients for whom individual data is shown above and right (with corresponding frame and font color). In these 3 plots, left: time corresponds to the hippocampus-amygdala transient broadband gamma activity cross-correlation peak lag for eR vs. eKF trials (colored inverted triangle); right: hippocampus gamma delay as a function of amygdala theta cycle (the delay is the difference between black and red triangles,). Note the different sign between Patient s15 (green) and Patient s16 (purple) in the two correlated measures.

Regarding the Rewiever's last point, PACOi generates its own surrogate distributions through a permutation test, which was reported in the original version of the manuscript and is now described in line 712: "Statistical significance was computed by randomly shuffling ($n=1000$) the trial condition assignment (*aversive remembered vs. aversive known/forgotten*)⁶⁹"

3. Lines 209 to 212. The PACOi index is used to measure the difference in preferred phase between two conditions, but it seems like an oblique way to do so. I see how PACOi would be sensitive to the phase difference between conditions, but why not just measure the phase difference directly? You have a distribution of preferred phases for the eR and eKF conditions, so can't you apply the method I suggest in item #2. In general, I suggest you remove all PACOi analyses given that they only measure phase differences indirectly.

R2.3. Action: we explain the rationale behind the use of PACOi and update the main text.

We thank the Reviewer for this remark which allows us to better explain the rationale behind the use of the PACOi method. First of all, we would like to clarify that PACOi is based on the phase opposition sum (POS) described in detail by VanRullen (VanRullen, R. How to evaluate phase differences between trial groups in ongoing electrophysiological signals. *Frontiers in neuroscience* **10**, 426-426 (2016)). The assumption underlying the test is that, if there is a consistent phase difference between two experimental conditions, the magnitude of the PPC computed over all trials (serving as a baseline) will be lower relative to the sum of the PPCs measured separately for each trial group (i.e. eR and eKF). The measure employs a permutation test on where the sum of PPCs (test statistic) computed from eR and eKF separately (within-condition contrast) is compared with the sum of PPCs computed from a pool of trials with the experimental labels permuted N times (between-condition contrast). If PPC within condition is higher than PPC between-condition, then it proves that eR and eKF don't have the same phase.

Although PACOi measures the phase difference indirectly, it solves two important problems that need to be considered in the context of phase estimation:

1) The magnitude of the phase concentration by itself is not informative.

PACOi measures whether this phase opposition (*i.e.* phase difference) is higher than a null model where trials are permuted (see VanRullen 2016 for a thorough explanation). Measuring the phase difference “directly” does not guarantee a statistically meaningful estimate per se. Analogous to the Reviewer’s point 2, in the original version of the manuscript we measured the phase delay between the two conditions as described in the section “*Relationship between within vs. cross-frequency hippocampus-amygdala coupling*” starting at line 767 in the revised version of the manuscript).

2) Relative vs absolute phase difference.

The advantage of PACOi is that it can measure a phase difference independently of where it occurs along the 0-2 π interval. Since it is based on differences in the sum of pair-wise phase consistencies (PPCs) of two conditions relative to the sum of PPCs of these two conditions with the trials permuted (null model), it is insensitive to the angle where the phase difference occurs. This is analogous to classical coherence analysis, where consistent phase differences between channel-pairs across trials are clipped between 0-1 independently of their location in the Angard diagram.

We now discuss in the method section both advantages and limitation of the PACOi approach from line 724-735:

“Although PACOi measures phase difference indirectly (see VanRullen 2016 for an in-depth discussion about phase opposition measures in general), it provides substantial advantages for dealing with two important problems that need to be considered in the context of phase estimation: 1) the magnitude of the phase concentration by itself is not informative. 2) Relative vs. absolute phase difference. PACOi measures whether the phase opposition (*i.e.*, phase difference) is higher than a null model where trials are permuted⁶⁹. Since it is based on differences in the sum of PPCs of two conditions relative to the sum of PPCs of these two conditions with the trials permuted (null model), the PACOi measure is insensitive to the angle where the phase difference occurs. In other words, the advantage of using PACOi method is that PACOi can measure phase difference independently of where it occurs along the 0-2 π interval. This approach is analogous to classical coherence analysis, where consistent phase differences between channel-pairs across trials are clipped between 0-1 independently of their location in the Angard diagram.”

- 69 VanRullen, R. How to evaluate phase differences between trial groups in ongoing electrophysiological signals. *Frontiers in neuroscience* 10, 426-426 (2016).

4. Lines 212 to 219. Here you measure what I suggested in item #3, but statistical significance is not provided. A permutation test would be appropriate.

R2.4. Action: permutation test has been run and results have been included in the main text.

We have now included a permutation test, as suggested by the Reviewer (line 232-236).

“In line with the peak-triggered average results, hippocampal gamma amplitude during successful vs. unsuccessful encoding of aversive scenes concentrated at different theta bins with a consistent phase difference of ~1.67 radians, corresponding to approximately 30-45 ms time difference (permutation test using Watson-Williams test, $F_{ww}=187.49$, $P=0.00009$, Fig. 3a-c)”

5. Lines 235 to 237. How are PACOi results calculated ‘per phase bin’? This needs to be explained better in the text.

R2.5. Action: the text has been updated with further methodological details.

Thanks for bringing up this issue and apologies for not providing a clearer explanation. PACOi’s phase-binning is based on the phase binning used in the PAC analysis. We did not compute a single PACOi value per phase bin (in case the Reviewer refers to this possibility). We computed PAC by dividing the analytic phase signal into 20 equal bins and taking the mean of the analytic amplitude over these specific phase-bins.

We revised the legend of Figure 3 (line 262-265) – the updated figure can be found below - and now make explicit that:

“The analytic phase signal was divided in $n=20$ equal bins ($\phi_n=[-\pi \pi]$) and the mean analytic amplitude was taken over those specific bins (see Materials and Methods at Phase to amplitude modulation index and Phase-Amplitude Coupling Opposition index (PACOi) sections) for each condition;”

This operation constitutes the “phase-amplitude histogram”, typical for state-of-the-art PAC metrics (Tort et al 2010 *J Neurophysiol*). The phase-amplitude histogram is computed for the two experimental conditions to be contrasted. At this point, we transform the phase-amplitude histogram into a complex number, as done in Canolty et al. (2006) with the following formula:

$$z_k = 1/N \sum_{b=1}^N (a_b e^{i\phi_b}),$$

with k being a single trial of a particular experimental condition, N the total number of phase bins, and a_b and ϕ_b representing the amplitude and the angle of a specific bin b . Therefore, z_k is a complex number per single trial and experimental condition. Supplementary Fig. 18 in the revised manuscript -already present in the original submitted manuscript - describes the analysis steps (the figure can be found below and in the revised version of the Supplementary material). We now better explain in the figure caption that “ z value is a weighted average that indicates the phase bin of the lower-frequency oscillation at which the amplitude of the high-frequency oscillation is strongest.

Supplementary Fig. 18. Phase-Amplitude Coupling Opposition index (PACOi). PACOi measures the phase opposition between two Phase-Amplitude Coupling (PAC) estimates obtained from two experimental conditions (left and right columns). The rationale of the metric is to test whether the high frequency amplitude of condition 1 vs. 2 locks at different phase bins. If PAC concentrates at different phase bins, the sum of the pairwise-phase consistency (PPC) of each experimental condition will exceed the PPC obtained from all trials together. Here we represent a simulation in which for each experimental condition and trial, the time series are filtered into low (left column; black traces) and high frequencies (left column: red traces) and the analytic phase and amplitude are computed taking the Hilbert transform respectively (for illustrative purposes we only show a few seconds of the time series). Second, the analytic phase of the low frequency is binned ($b=20$) and the high frequency amplitude is averaged within each phase bin, forming a histogram (left column; red histogram). Third, for each trial, this histogram is transformed into a complex number (z_k ; inserted formula) by multiplying the high frequency amplitude (histogram y-axis: ab) by the low frequency phase (histogram x-axis: $\exp(i\phi_b)$; where ϕ_b denotes the average phase of each bin). This z value is a weighted average that indicates the phase **bin** of the lower-frequency oscillation at which the amplitude of the high-frequency oscillation is strongest. The z_k complex values were taken for each experimental trial and were normalized to unit length. The right column has the same conventions as the left but for condition 2. Middle column: The phase opposition is based on the sum of the PPC of each condition relative to the PPC calculated taking together the trials of the two conditions ($PACOi = PPC_{eR} + PPC_{eKF} - 2 * PPC_{eR,eKF}$). The PPC is defined as

$$PPC = \frac{(\tilde{Z} * \text{conj}(\tilde{Z}) - N)}{(N * (N - 1))}$$

and

$$\tilde{Z} = \sum_{k=1}^N Z_k$$

being the sum of single trial complex z values (N =total number of trials). PPC expected value runs from 0 (uniform phase distribution) and 1 (maximum pairwise phase consistency). Negative value may appear due to unbiasedness of the PPC (see Vinck 2010 for details)¹. Statistical significance was established by permuting ($n=1000$ times) the trials of each experimental conditions for a number of pair of frequencies (5–20 Hz in 1 Hz steps and 40–120 Hz in 5 Hz steps). For each patient and bipolar channel, we selected the frequency phase-amplitude pairs that showed a significant PACOi (uncorrected).

We included this reference in the revised version of the Supplementary Material

- 1 Vinck, M., van Wingerden, M., Womelsdorf, T., Fries, P. & Pennartz, C. M. A. The pairwise phase consistency: A bias-free measure of rhythmic neuronal synchronization. *NeuroImage* **51**, 112-122, doi:10.1016/j.neuroimage.2010.01.073 (2010).

For the Reviewer's convenience, we copy the relevant method sections and revised Figure 3 below:

Phase to amplitude modulation index. We calculated phase-to-amplitude coupling (PAC) using the Modulation index (MI) as previously defined^{29,67}. This was performed over the time period where we found a subsequent memory by emotion interaction within amygdala gamma broadband activity (0.41–1.1 s; Fig. 1h). First, for each patient and bipolar channel of interest, single trials were bandpass filtered around two sets of frequencies to obtain instantaneous phase (f_p ; from 5–20 Hz in steps of 1 Hz) and amplitude (f_a ; from 40–120 Hz in steps of 5 Hz). The analytic phase (ϕ_{f_p}) was obtained by taking the angle of the Hilbert transform of the bandpass filtered data around f_p with a bandwidth of ± 2 Hz. For the same trial, the analytic amplitude (a_{f_a}) was similarly obtained but by taking the magnitude of the Hilbert transform of the bandpass filtered data around f_a with a bandwidth of ± 15 Hz. We used FIR filters with the filter order being set to three cycles of the lower bandwidth bound. Before filtering, single trials were z -scored. Second, for a given frequency pair, we constructed the amplitude-phase histograms as follows. The analytic phase signal was divided in $n=20$ equal bins ($\phi_n \in [-\pi, \pi]$) and the mean analytic amplitude was taken over those specific bins. Third, for each frequency pair, the MI is computed as the Kullback-Leibler divergence⁶⁸ between the amplitude-phase histogram pooled from all corresponding trials and compared to the uniform distribution.

Phase-Amplitude Coupling Opposition index (PACOi). The MI measures the modulation of the amplitude of high frequencies by the phase of low frequencies independently of the preferred phase angle at which the high frequency amplitude occurs. We derived a metric, the PACOi, to exploit potential differences in the preferred phase angle that two PAC distributions can produce. For example, the maximum at which the amplitude of an oscillation (i.e., gamma) concentrates in a given phase bin (i.e., theta peak) in each experimental condition can be different (or not) from another condition (i.e., theta trough). To formalize this quantification, we tested whether the pair-wise phase consistency (PPC) of each trial type exceeded the overall PPC taken over the two groups of trials together⁶⁹. Supplementary Fig. 18 provides, schematically, the rationale and quantification of the measure. The calculation of PACOi starts by first transforming the amplitude-phase histograms (see Modulation index above) to the complex domain as follows:

$$z_k = 1/N \sum_{b=1}^N (a_b e^{i\phi_b}),$$

with k being a single trial of a particular experimental condition, N the total number of phase bins, and a_b and ϕ_b representing the amplitude and the angle of a specific bin b . Therefore, we obtain a complex number per single trial and experimental condition. Once in the complex domain, these indices are normalized to unit length ($z/abs(z)$) and the PPC is computed for each condition separately (PPC_1, PPC_2), and then by combining the trials of the two conditions (PPC_{all} ; Supplementary Fig. 18). This procedure was carried out, with the rationale that if the gamma amplitude in eR and eKF concentrates systematically at different bins of theta, then the sum of the PPC obtained from each experimental condition separately should be larger than the PPC obtained from all trials pooled. Statistical significance was computed by randomly shuffling ($n=1000$) the trial condition assignment (aversive remembered vs. aversive known/forgotten)⁶⁹. It is a common observation that the angular bin to which the phase opposition points varies between test subjects^{69,70}. Each patient's phase distributions were therefore realigned (Fig. 3a) such that, for each patient, the amygdala theta phase at which the hippocampus broadband gamma coupled was set to a phase angle of zero under the eR condition. The exact number of phase shifts was obtained by computing the angle at which the vector strength was pointing at ($angle(\langle z_{eR} \rangle)$), being $\langle \rangle$ the average over complex single trials). Once the angle was obtained (i.e., $\pi/2$) this angle was subtracted from both the eR and eKF conditions. Thus, eR theta-gamma PAC histogram is centered around zero. However, eKF angle preference can fall either $[-\pi, 0]$ or $[0, \pi]$, which is a nontrivial property confirming our hypothesis that eR and eKF are associated with opposite phase angles. By convention, we mirror flipped the eKF histogram distribution when the vector strength angle falls with the $[-\pi, 0]$ interval.

Although PACOi measures phase difference indirectly (see VanRullen 2016 for an in-depth discussion about phase opposition measures in general), it provides substantial advantages for dealing with two important problems that need to be considered in the context of phase estimation: 1) the magnitude of the phase concentration by itself is not informative. 2) Relative vs. absolute phase difference. PACOi measures whether the phase opposition (i.e., phase difference) is higher than a null model where trials are permuted⁶⁹. Since it is based on differences in the sum of PPCs of two conditions relative to the sum of PPCs of these two conditions with the trials permuted (null model), the PACOi measure is insensitive to the angle where the phase difference occurs. In other words, the advantage of using PACOi method is that PACOi can measure phase difference independently of where it occurs along the 0-2pi interval. This approach is analogous to classical coherence analysis, where consistent phase differences between channel-pairs across trials are clipped between 0-1 independently of their location in the Angard diagram.

Fig. 3. Emotional memory-dependent amygdala phase opposition of hippocampal gamma and single neuron activity. **a**, Hippocampus gamma amplitude locks to different phase bins of amygdala theta depending on emotional memory outcome. Histogram of amygdala theta phases at which hippocampus broadband gamma amplitude occurs for eR (red) and eKF (black) trials as bar (left) and circular (right) plots. Inverted triangles represent the angle of preference in radians (eR:0.16 radians; eKF:1.83) after phase realignment. **b**, Circular plot, shaded areas in the circular plot represent PACOi results ($n=9$ electrodes) per phase bin ($n=20$). The analytic phase signal was divided in $n=20$ equal bins ($\phi_n=[-\pi \pi]$) and the mean analytic amplitude was taken over those specific bins (see Materials and Methods at Phase to amplitude modulation index and Phase-Amplitude Coupling Opposition index (PACOi) sections) for each condition; max is the maximum value of phase-amplitude coupling for eR:1.19 and eKF:1.04. **c**, Histogram displays the permuted values for the Watson Williams test, red line represents the empirical statistic ($F_{ww}=187.49$, $P=0.00009$). **d**, Summary of electrode contact localization in the left and right amygdala (pink) and hippocampus (light blue) for patients included in the SFC analysis (Cohort 2: 6 patients and a

total of 7 electrodes). The first contact is just lateral to the putative micro-wire location. e, Example for a single subject showing a post-operative CT image, thresholded to visualize electrode contacts, co-registered with the corresponding pre-operative MRI scan in native space and superimposed onto the amygdala and hippocampus subfields. The sagittal view shows that contacts are in the amygdala and anterior hippocampus. The first coronal view shows hippocampal subfields and the white and black arrow point to the putative microelectrode location. The second coronal view shows amygdala nuclei and the two white and black arrows indicate the first two macroelectrodes locations. Subfields/nuclei are color coded. A schematic of the electrode implanted (1.3 mm diameter, 8 contacts of 1.6 mm length, and spacing between contact centers 5 mm; Ad-Tech, Racine, WI) is provided showing microwires protruding from the tip and macrocontacts (the first three contacts are numbered). f, As for a, but showing amygdala theta phases at which hippocampal spikes occur (average spike count over $n=31$ neurons. Inverted triangles represent the angle of preference in radians ($eR:0.18$; $eKF:1.40$) after phase realignment. g, Shaded areas in the circular plot represent spikes per the 20 phase bins for each condition. Maximum spike count for each condition is indicated ($eR:112$; $eKF:128$) as well as the total spike counts across neurons ($eR:1651$, $eKF:2030$). Upper subpanel: single-neuron waveform ($\text{mean} \pm \text{std}$) for one example neuron. h, Histogram displays the permuted values for the Watson Williams test, red line represents the empirical statistic ($F_{ww}=40.91$, $P=0.00009$). Upper subpanel: single-neuron waveform ($\text{mean} \pm \text{std}$) for one example neuron.

6. Lines 290 to 294. This analysis seeks to demonstrate a directionality in gamma interactions between amygdala and hippocampus by calculating the CC between gamma burst peaks in the hippocampus and gamma power in the amygdala for two conditions and then taking their difference. Why not just take the difference in the peak lags for the two CCs?

R2.6a. Action: we updated Figure 4b and the legend with further details.

The Reviewer is correct and this is indeed one of the outputs we obtained. Figure 4b now illustrates the subtraction operation:

The right figure (thick line with red & black colors) is the result of the R_{ha} envelope (emotional subsequent remember CC between hippocampal gamma peaks (h) and amygdala gamma peaks (a) minus the KF_{ha} envelope (emotional know/forgotten CC between hippocampal gamma peaks (h) and amygdala gamma peaks (a)). The legend of figure 4 has been updated as follows (line 355-358):

“The amplitude envelopes from all peaks and single trial cross-correlations (thin lines, left) were averaged (thick lines, left). The difference between the envelope R_{ha} (emotional subsequent remember CC between hippocampal gamma peaks (h) and amygdala gamma peaks

(a)) minus envelope KF_{ha} (emotional know/forgotten CC between hippocampal gamma peaks (h) and amygdala gamma peaks (a)) was computed ~~contrasted~~ (dashed red and black thick line, right).”

In addition, the only significant difference in Fig 4c was at 50 ms where the difference in the CC envelopes was reaching a minimal value and the SE had shrunk consequently. This segment of the CC is particular sensitive to edge effects of the CC calculation since the epoched traces were only 60ms long.

R2.6b. Action: Figure 4 has been changed and the main text has been updated.

The Reviewer is again correct and what we aimed to measure was the magnitude of the difference of the amplitude of the cross-correlation envelopes between the eR vs eKF conditions. For the Reviewer’s convenience, we copy the updated Figure 4 below. Following the Reviewer’s remark, we included a note about the risk of edge effects as follows (see line 314-327):

“Using the amplitude envelope as a measure of synchrony strength between the two structures, we ~~indeed~~ found stronger amygdala to hippocampus transient gamma synchronization for subsequently remembered compared to not remembered aversive scenes (lag 0.044–0.056 s, summed t -value=29.73, $P=0.011$, Fig. 4c, Supplementary Fig. 20; controlling the number of trials between conditions, lag 0.044–0.06 s, summed t -value=31.00, $P=0.011$). The time period where the difference appears may be prone to edge artifacts, although we note that there is no significant difference between conditions at the corresponding negative lag (indexing stronger hippocampus to amygdala transient gamma synchronization for subsequently remembered compared to not remembered aversive scenes). At shorter lags, there is more variability in amygdala to hippocampus transient gamma synchronization for subsequently remembered compared to not remembered aversive scenes (Supplementary Fig. 19), but this variability is, in turn, related to the patient-specific amygdala theta-hippocampal gamma phase lag between eR vs. eKF trials, as described next.” Three individual patient amygdala-hippocampal gamma cross-correlations for eR vs. eKF trials are shown in Fig. 4d (inserts).

Fig. 4. Broadband gamma (60–120 Hz) transient connectivity between amygdala and hippocampus is paced by amygdala theta-hippocampal gamma phase opposition. *a, b*, Schematic depiction of the analysis. *a*, Broadband gamma activity was bandpass filtered (60–120 Hz) and peaks in hippocampal recordings identified (top; two example trials from one patient). Short epochs around identified peaks were cut **out** (± 0.03 s; thin horizontal lines under gamma traces). For each epoch, the amplitude envelope of the cross-correlation between hippocampus (blue) and amygdala (green) epochs was computed for eR (below, red envelope) and eKF (black) trials. *b*, The amplitude envelopes from all peaks and single trial cross-correlations (thin lines) were averaged (thick lines, left). The difference between the envelope R_{ha} (emotional subsequent remember CC between hippocampal gamma peaks (h) and amygdala gamma peaks (a)) minus envelope KF_{ha} (emotional know/forgotten CC between hippocampal gamma peaks (h) and amygdala gamma peaks (a)) was computed (dashed red and black thick line, right). The negative and positive x-axis values of the envelope cross-correlogram indicate that hippocampus gamma leads amygdala gamma (light blue) and the reverse directionality (grey), respectively. *c*, Following the analysis in A and B in all patients in Cohort 1, amplitude envelope cross-correlation shows amygdala leading hippocampus broadband gamma activity. The shaded contours represent \pm s.e.m (color indicates directionality of transient coupling). Red bar depicts significant lag window for all trials; light red bar above after controlling the number of trials between conditions. *d*, Correlation between successful vs. unsuccessful emotional memory as measured as hippocampus-amygdala transient broadband gamma activity peak lag (x-axis) and hippocampus gamma delay as a function of amygdala theta cycle (y-axis). The x-axis represents the time lag measured by the peak of the cross-correlation (60-120 Hz) power envelopes between hippocampus and amygdala. The y-axis is the PACOi frequency-specific phase opposition result translated into a time lag. See Methods section for description of the transformation of PACOi frequency specific phase opposition into a time delay. The colored dots (red, blue, purple) represent 3 different patients for whom individual data is shown above and right (with corresponding frame and font color). In these 3 plots, left: time corresponds to the hippocampus-amygdala transient broadband gamma activity cross-correlation peak lag for eR vs. eKF trials (colored inverted triangle); right: hippocampus gamma delay as a function of amygdala theta cycle (the delay is the difference between black and red triangles). Note the different sign between Patient 15 (green) and Patient 16 (purple) in the two correlated measures.

7. The meaning of fig 4d is unclear. I have no idea how to interpret this.

R2.7. Actions: Figure 4d has been updated, permutation test has been run and the permutation distribution has been added in the figure. We propose a schematic representation of the results which can go into the Supplementals material or main text at the Reviewer's discretion.

We apologise for the confusion produced by this figure and we have now thoroughly updated Fig. 4d. The new panels better illustrate the relation between hippocampus-amygdala transient broadband gamma activity cross-correlation peak lag for eR vs. eKF trials and the hippocampus gamma delay as a function of amygdala theta cycle. We now included individual data from three different patients as insets. We also made several improvements in the legend (line 364-371) and main text (lines 327-344) that we hope will clarify the meaning of this figure.

d, Correlation between successful vs. unsuccessful emotional memory as measured as hippocampus-amygdala transient broadband gamma activity peak lag (x-axis) and hippocampus gamma delay as a function of amygdala theta cycle (y-axis); The x-axis represents the time lag measured by the peak of the cross-correlation (60-120 Hz) power envelopes between hippocampus and amygdala. The y-axis is the PACOi frequency-specific phase opposition result translated into a time lag. See Methods section for description of the transformation of PACOi frequency specific phase opposition into a time delay). The colored dots (red, blue, purple) represent 3 different patients for whom individual data is shown above and right (with corresponding frame and font color). In these 3 plots, left: time corresponds to the hippocampus-amygdala transient broadband gamma activity cross-correlation peak lag for eR vs. eKF trials (colored inverted triangle); right: hippocampus gamma delay as a function of amygdala theta cycle (the delay is the difference between black and red triangles,). Note the different sign between Patient s15 (green) and Patient s16 (purple) in the two correlated measures.

Please note that, during the revision process, we found a bug in our code that affected two patients of the hippocampus-amygdala transient broadband gamma activity peak lag (x-axis) estimation. Our code looked for the positive maximum peak lag while it should have used the absolute peak lag (both positive and negative). This change affected data from two patients and the overall correlation result was only minimally affected: original Spearman's $\rho=0.81$, $P=0.011$; correct Spearman's $\rho=0.78$, $P=0.013$).

We furthermore ran a permutation analysis and update the main text as follow (lines 337-343):

In line with the Granger causality results showing theta amygdala to hippocampus directionality effect, we found that amygdala gamma bursts lead hippocampal gamma bursts (Spearman $\rho=0.81$, $P=0.011$; Spearman's $\rho=0.78$, $P=0.013$; $P_{\text{permuted}}=0.0078$, Fig. 4d), with a latency of 37.2 ± 5.9 ms (mean \pm s.e.m.). The correlation suggests a monotonic relation between tri-partite activity (hippocampus gamma, amygdala theta, amygdala-hippocampus

broadband 60-120 Hz): the PACOi theta-gamma time delay is linearly related to the lag obtained in the amplitude envelope cross-correlation in the 60-120 Hz range.

As explained in the point above, the x -axis represents the time lag measured by the peak of the cross-correlation (60-120 Hz) power envelopes between hippocampus and amygdala. The y -axis translates the PACOi frequency specific phase opposition into a time lag obtained by the frequency-pairs specific for each patient as follows:

$$delay = \frac{\left(angle \left(z_{eR}^{f_x, f_y} \right) - angle \left(z_{eKF}^{f_x, f_y} \right) \right)}{(2\pi)} * \left(\frac{1}{f_{xc}} \right)$$

This formula measures, for each patient, the PACOi angle difference (as in Fig. 3a) between the eR and eKF conditions weighted by the frequency centroid resulting from the significant PACOi low frequencies. The correlation suggests a monotonic relation between the tri-partite activity (hippocampus gamma, amygdala theta, and amygdala-hippocampus broadband 60-120 Hz cross correlation peak). The PACOi theta-gamma time delay is monotonically related to the lag obtained in the amplitude envelope cross-correlation in the 60-120 Hz range. We propose this figure as a schematic representation of the results which can go into the Supplementals or main text at the Reviewer's discretion.

The following information is now included in the main text (lines 327-344):

“Finally, to verify the role of ongoing amygdala theta in the modulation of the transient gamma synchronization between amygdala and hippocampus, we tested whether the transient connectivity indexed by the amplitude envelope and PACOi results were related. PACOi phase differences were converted to time delays (Fig. 4d, y -axis) using the following formula:

$$delay = \left(angle \left(z_{eR}^{f_x, f_y} \right) - angle \left(z_{eKF}^{f_x, f_y} \right) \right) / (2\pi) * (1/f_{xc})$$

This formula measures, for each patient, the PACOi angle difference (as in Fig. 3a) between the eR and eKF conditions weighted by the frequency centroid resulting from the significant PACOi low frequencies.

These estimates were correlated with the peak of the envelope of the cross-correlation computed as the difference between subsequently remembered vs. not remembered aversive trials (Fig. 4d x-axis). In line with the Granger causality results showing theta amygdala to hippocampus directionality effect, we found that amygdala gamma bursts lead hippocampal gamma bursts (Spearman's $\rho=0.78$, $P=0.013$; Fig. 4d), with a latency of 37.2 ± 5.9 ms (mean \pm s.e.m.). The correlation suggests a monotonic relation between tri-partite activity (hippocampus gamma, amygdala theta, amygdala-hippocampus broadband 60-120 Hz): the PACOi theta-gamma time delay is linearly related to the lag obtained in the amplitude envelope cross-correlation in the 60-120 Hz range. This significant correlation indicates a role of amygdala theta in synchronizing amygdala and hippocampal high gamma activity.”

The completed updated Figure 4 can be found here below

Fig. 4. Broadband gamma (60–120 Hz) transient connectivity between amygdala and hippocampus is paced by amygdala theta-hippocampal gamma phase opposition. a, b, Schematic depiction of the analysis. a, Broadband gamma activity was bandpass filtered (60–120 Hz) and peaks in hippocampal recordings identified (top; two example trials from one patient). Short epochs around identified peaks were cut out ($\pm 0.03s$; thin horizontal lines under gamma traces). For each epoch, the amplitude envelope of the cross-correlation between hippocampus (blue) and amygdala (green) epochs was computed for eR (below, red envelope) and eKF (black) trials. b, The amplitude envelopes from all peaks and single trial cross-correlations (thin lines) were averaged (thick lines, left) and the difference between the envelope R_{ha} (emotional subsequent remember CC between hippocampal gamma peaks (h) and amygdala gamma peaks (a)) minus envelope KF_{ha} (emotional know/forgotten CC between hippocampal gamma peaks (h) and amygdala gamma peaks (a)) was computed (dashed red and black thick line, right). The negative and positive x-axis values of the envelope cross-correlogram indicate that hippocampus gamma leads amygdala gamma (light blue) and the reverse directionality (grey), respectively. c, Following the analysis in A and B in all patients in Cohort 1, amplitude envelope cross-correlation shows amygdala leading hippocampus broadband gamma activity. The shaded contours represent \pm s.e.m (color indicates directionality of transient coupling). Red bar depicts significant lag window for all trials; light red bar above after controlling the number of trials between conditions. d, Correlation between successful vs. unsuccessful emotional memory as measured as hippocampus-amygdala transient broadband gamma activity peak lag (x-axis) and hippocampus gamma delay as a function of amygdala theta cycle (y-axis). The x-axis represents the time lag measured by the peak of the cross-correlation (60-120 Hz) power envelopes between hippocampus and amygdala. The y-axis is the PACOi frequency-specific phase opposition result translated into a time lag. See Methods section for description of the transformation of PACOi frequency specific phase opposition into a time delay. The colored dots (red, blue, purple) represent 3 different patients for whom individual data is shown above and right (with corresponding frame and font color). In these 3 plots, left: time corresponds to the hippocampus-amygdala transient broadband gamma activity cross-correlation peak lag for eR vs. eKF trials (colored inverted triangle); right: hippocampus gamma delay as a function of amygdala theta cycle (the delay is the difference between black and red triangles). Note the different sign between Patient 15 (green) and Patient 16 (purple) in the two correlated measures.

8. Statistical validation of of results are applied inconsistently. At the very least all major findings should be demonstrated at the subject population level, or within subject and the percentage of significant subjects provided.

R2.8. Action: we provide further justifications for the statistic used and we have updated Supplementary Fig. 19.

We thank the Reviewer for raising this important point. We now make it clear in the main text that we adopt a fixed-effect statistical model throughout. For all main findings we used permutation methods at the population level. For statistical testing in all time frequency, Granger causal and phase amplitude coupling analyses, we applied a cluster-based permutation test (Maris & Oostenveld, 2007) using the Monte Carlo method. In the PACOi and broadband gamma correlation analysis we estimated surrogate statistics at the population level. In designing the memory paradigm, task difficulty and task duration must be planned taking into account the clinical and cognitive status of patients performing this task. The latter can be quite variable, which in some cases will lead to a limited number of trials for some conditions per subject. Thus, we at no stage devised an analysis strategy at the single subject level.

Nevertheless, in the original submission we do – in the spirit of transparent reporting – show single subject data for all main analyses. Consistent patterns can be observed in all of these. We used scatter dot plots for time frequency, granger and PAC analysis. For PACOi analysis we show patient-specific broadband gamma activity sub-bands. Colorbar indicates the PACOi masked by uncorrected statistical contrast between the eR vs. eKF group of trials (Supplementary Fig. 19).

Supplementary Fig. 18 19. Phase-Amplitude Coupling Opposition index (PACOi) for aversive remembered and aversive known/forgotten trials for individual amygdala-hippocampal electrode pairs (Cohort 1). For each pair, left: comodulogram showing the amygdala phase (x-axes) to hippocampus amplitude (y-axes) phase opposition between emotional remembered (eR) vs. emotional know + forgotten (eKF) trials. Colorbar indicates the PACOi values (see Materials and Methods, fig. S13) masked by the (uncorrected) statistical contrast between the eR vs. eKF group of trials. For each amygdala-hippocampal electrode pair, right: averaged hippocampus amplitude to amygdala phase histogram resulting from the significant (masked comodulogram) PACOi realigned. Red represents eR and black eKF. PACOi analyses were performed over the 0.41–1.1 s post-stimulus time interval.

Finally, Supplementary Fig. 19 in the original submission showed broadband gamma (60–120 Hz) transient connectivity analysis between hippocampus and amygdala assessed by the amplitude envelope of cross-correlation in Cohort 1. We updated this figure adding red triangles representing the peak lag of the cross-correlation used in Fig. 4d for a better understanding of the results.

Supplementary Fig. 20 19. Broadband gamma (60–120 Hz) transient connectivity analysis between hippocampus and amygdala assessed by the amplitude envelope of cross-correlation (Cohort 1). Red/black curves represent the average amplitude envelope cross-correlation (Env. xcorr.) contrast between eR vs. eKF). The input signals to compute the cross-correlation were the hippocampus and amygdala broadband gamma activities taken over the 0.41–1.1 s post-stimulus time interval. Red triangles represent the peak lag of the cross-correlation used in Fig. 4d. Shaded grey area represents the s.e.m. taken over the epochs, a.u.: arbitrary units.

Minor issues:

1. Line 601. Why demean before high pass filtering? Is it just to minimize the transient at the beginning of the trace when the filter receives zero padded values?

R2.m1 Action: we included new references to support the method used.

The demeaning before filtering reduces the ringing artifacts since there is no “DC jump” in the signal. Demeaning is performed before zero-padding. This is a well-established practice (see de Cheveigne et al 2019 and Widmann et al 2015). We insert the following in the main text (lines 759-760):

“Before bandpass filtering, single trials were demeaned in order to reduce ringing artifacts 71,72”

We included the following reference in the main text:

71 de Cheveigné, A., & Nelken, I. (2019). Filters: when, why, and how (not) to use them. *Neuron*, 102(2), 280-293.

72 Widmann, A., Schröger, E., & Maess, B. (2015). Digital filter design for electrophysiological data—a practical approach. *Journal of neuroscience methods*, 250, 34-46.

2. Page 62, line 123. PPC values can actually go below zero. See Vinck et al. 2010.

R2.m2. Action: we included the reference suggested by the Reviewer.

We thank the Reviewer for noticing this fact, which we are aware of. As described in Vinck et al 2010, the negative bias is a consequence of the unbiasedness of the PPC. We address the negative bias in the permutation approach since this bias will be present in the permutation distribution too. We include the following detail in the legend of Supplementary Fig. 18 in the Supplementary material and refer to Vinck et al 2010.

PPC expected value runs from 0 (random uniform phase distribution) and 1 (maximum pairwise phase consistency). Negative value may appear due to the unbiasedness of the PPC (see ⁷⁷ for details)

77 Vinck, M., van Wingerden, M., Womelsdorf, T., Fries, P., & Pennartz, C. M. (2010). The pairwise phase consistency: a bias-free measure of rhythmic neuronal synchronization. *Neuroimage*, 51(1), 112-122.

3. The discussion seems a bit skimpy. Perhaps the authors could further explore how their results relate to the physiology of theta phase dependent encoding/retrieval in CA1.

R2.m3. Action: we extended the discussion and included a summary figure of the results.

Many thanks for the suggestion. We now extended the discussion as suggested here and by Reviewer 3 (380-458).

“Our data show that in response to an aversive visual stimulus, the amygdala influences ongoing hippocampal theta oscillations, which in turn organize the amplitude of local gamma activity and neuronal firing (Figure 5). High gamma activity in the amygdala was enhanced at 310 ms after stimulus presentation with a greater amplitude for emotional aversive scenes that were later remembered (relative to those receiving known responses or forgotten ones). Subsequently, aversive scenes triggered unidirectional transmission of theta oscillations from the amygdala to the hippocampus compared to neutral ones (from 430 to 770 ms). At around 500 ms gamma power increased in the hippocampus for stimuli that were later recalled irrespective of their valence. Considering a time window from 400 to 1000 ms, the alignment of hippocampal broadband gamma activity (60-120 Hz) and neuronal firing to the amygdala’s theta phase differed between subsequently remembered vs. not-remembered emotional stimuli. We observed a consistent phase difference between the two conditions of ~1.67 radians, corresponding to approximately 30-45 ms. Crucially, this time difference also led to a transient lagged coherence (latency of around 37 ms) between gamma activity in the two structures that predicts subsequent memory for emotional stimuli (Fig. 5). The BLA has strong connections to the hippocampus and electrical stimulation of the BLA improved performance in memory tasks ^{36,37}Theta-gamma-phase-amplitude-coupling-in-the-human-hippocampus-has-been-shown-to-be-a-general-mechanism-for-memory-encoding^{28,29}.—In rodents, theta-modulated gamma stimulation applied to BLA is an efficient protocol to enhance hippocampal CA1 gamma

responses^{40,41}, and directly stimulating the human amygdala at theta-modulated gamma frequency immediately following the presentation of emotionally neutral stimuli leads to memory enhancement for simultaneously presented emotionally neutral stimuli⁴². Theta-gamma phase-amplitude coupling in the human hippocampus has been shown to be a general mechanism for memory encoding^{38,39}. Here, we show that emotional memory enhancement does not simply depend on theta-gamma phase-amplitude coupling. Confirming theoretical positions that memory formation is phase-dependent in humans^{23,24}, we showed that the formation of memories for emotional stimuli (allowing later recollected emotional memories relative to familiar or forgotten ones) depends on the amygdala theta phase to which the hippocampal gamma and related neuronal firing couples.

We highlight the importance of theta modulation of gamma activity in encoding of emotional memories, and lend support to direct stimulation protocols employing theta-burst stimulation aimed at improving memory⁴²⁻⁴⁴. Theta-burst stimulation is the delivery of several stimulation pulses at high frequency (*i.e.*, gamma frequency) that rhythmically alternate (in the theta range) with periods of no stimulation. By contrast, deep brain stimulation of the amygdala applied continuously at high frequency (*e.g.*, 160 Hz), currently being trialed for PTSD⁴⁵, may not permit the amygdala-hippocampal phase code mechanism described here to take place in response to emotional events, thereby limiting the formation of novel emotional memories.

Hippocampal long term potentiation (LTP), a form of synaptic plasticity thought to be involved in learning and memory^{30,46}, is optimally induced in CA1 when electrical stimulation occurs at the peak of the theta oscillation⁴⁷. This has led to a suggestion that inputs arriving during different specific theta phases will generate different synaptic modifications, which in turn will influence the likelihood that these inputs are encoded into long term memory^{31,47,48}. Our findings support this suggestion, by showing that the precise theta phase at which hippocampal gamma peaks and spikes occurred determined whether aversive scenes were later remembered, or not. Critically, the phase difference between subsequently remembered *vs.* forgotten or familiar emotional stimuli associated with this process translates to the time period required for amygdala and hippocampal gamma bursts to reach transient time-lagged coherence (~25-45 ms). It is possible that this lag is related to the time required for noradrenergic input, upon which emotional memory formation is critically dependent^{12,13}, to reach the medial temporal lobe, or that amygdala theta phase-dependent effects in the hippocampus are linked to the optimal conditions required for "emotion tagging" of memory⁴⁹ to occur. Future studies may explore whether a similar phase offset is present at retrieval or whether a different phase relationship may exist between amygdala and hippocampus. As encoding and retrieval has been shown to rely on complementary processes, the latter may be hippocampus-centered and thus may not rely on amygdala modulation^{12,50}. Moreover, these findings could represent a general mechanism through which amygdala oscillations influence other brain areas to enable emotion-induced modulation of further aspects of cognition, including perception, attention and decision-making^{14,51}, and inform therapeutic approaches of amygdala stimulation to memory dysfunctions and psychiatric disorders. “

4. Lines 252 to 253. Isn't this expected given that the significant increases in gamma power in amygdala and hippocampus occupied different frequency ranges?

R2.m4. We thank the Reviewer for this observation. The Reviewer asks whether coherence between amygdala and hippocampus in the gamma range was expected to not be significant since the increase in gamma power in the two structures occupied different frequency ranges. Coherence is based on phase and power relationship. Power can be correlated between oscillations of different frequencies, while phases cannot easily synchronize unless one

assumes n:m phase synchronization. Moreover, the significant effect in each structure come from cluster-based statistic. As we are now mentioning in the manuscript (lines 304-305) “cluster-based permutation tests³³ do not provide statistical inference for the exact latency and frequency of the effects³⁴”. Variability between effects may then most likely reflects differences in signal to noise ratio (SNR) suggesting that we are measuring a broadband gamma activity rather than an oscillatory activity.

5. Lines 89. A citation showing that it is ‘commonly observed’ emotional stimuli have a higher remember false alarm rate would be helpful.

R2.m5. Action: following Reviewer suggestion, we now quote Bessette-Symons 2017 (line 92)

19 Bessette-Symons, B. A. The robustness of false memory for emotional pictures. *Memory*, 2017.

6. Given that the recollection and false recollection of emotional stimuli are both higher than neutral stimuli, would that suggest that emotional stimuli are remembered with less accuracy?

R2.m6. Action: the main text has been changed to address this point.

This is an interesting point regarding the effect of emotion on memory, for which there is a long history but currently no clear consensus. Previous studies have explored whether emotional memories are remembered more accurately than neutral ones (e.g., Brown & Kulik, 1977; Loftus, 1993; Matlin & Stang, 1978). There is evidence that emotion heightens the feeling of remembering without necessarily implying a difference in accuracy between emotional and neutral items (Sharot 2004, Talarico 2003, Bessette-Symons 2017). For instance, Bessette-Symons et al. shows that emotional pictures enhance higher relatedness compared to neutral ones. Although relatedness might be difficult to control when using scenes and thus could be a possible confound, several pieces of evidence (Bessette-Symons 2017, Gallo et al. 2009) support that relatedness alone does not provide a sufficient explanation for all memory effects associated with emotional items compared to neutral ones. We now refer to this aspect in the main text at line 91-98:

“Across both patient cohorts, recollection performance for emotional stimuli (eR) was higher than all other response categories (Fig. 1b). As is commonly observed¹⁹, the remember false alarm (RFA) rate was higher for emotional (mean \pm s.e.m.: 21.94% \pm 3.98) compared to neutral stimuli (9.51% \pm 1.56; $t_{17}=3.76$, $P=0.002$, $d=0.878$). This difference may partially be linked to higher perceived relatedness for emotional items compared to neutral ones, which might increase a false sense of recollection due to generalization in memory^{20,21}. However, it has been shown that relatedness alone is not sufficient to account for all emotion-dependent differences in false recognition^{19,22}.”

We now refer to this study already quoted in the submitted manuscript

19 Bessette-Symons, B. A. The robustness of false memory for emotional pictures. *Memory* 26, 171-188 (2018).

And included the following reference in the revised version of the manuscript

- 20 Riberto, M., Paz, R., Pobric, G. & Talmi, D. The neural representations of emotional experiences are more similar than those of neutral experiences. *Journal of Neuroscience* **42**, 2772-2785 (2022).
- 21 Bierbrauer, A., Fellner, M.-C., Heinen, R., Wolf, O. T. & Axmacher, N. The memory trace of a stressful episode. *Current Biology* **31**, 5204-5213. e5208 (2021).
- 22 Gallo, D. A., Foster, K. T. & Johnson, E. L. Elevated false recollection of emotional pictures in young and older adults. *Psychology and aging* **24**, 981 (2009).

Reviewer #3 (Remarks to the Author):

The paper titled “Aversive memory formation in humans is determined by an amygdala-hippocampus phase code” sought to examine the directional connectivity of human amygdala-hippocampal interactions during aversive memory encoding. They found that emotional stimuli produce a unilateral influence from the amygdala to the hippocampus through theta oscillations and that memory for emotional stimuli depends on alignment of the hippocampus’ gamma activity and neuronal firing to the amygdala’s theta phase during encoding. Crucially, this interaction produces a transient lagged coherence that predicts of subsequent memory for emotional stimuli. Overall, they suggest that the precise theta phase interplay between the amygdala and hippocampus during emotional memory encoding might be indicative of a more general oscillatory mechanism through which the amygdala communicates emotional information throughout the brain.

This work represents a well-reasoned and critical next step in understanding emotion and memory interactions in the human brain. The authors utilize thoughtful and creative analyses to uncover the unique nature of amygdala hippocampal interactions during encoding of emotional images. Using a well-established aversive memory paradigm built on a rich literature in experimental animals and humans, the authors have produced truly novel data and analyses that have important implications for our understanding and developing treatments of memory and psychiatric disorders. I strongly support the publication of this article in Nature Communications, however a few minor issues should be addressed prior to publication. I’ve listed my suggestions for revisions by section and line below. Feel free to ask the editor for me to clarify any of these points if needed. Well done!

We are extremely grateful to the Reviewer for their positive and encouraging feedback.

Comments

Introduction

1. Excellent introduction. No comments.

Thank you.

Results

1. Figure 1: Some statistical characterization of the temporal extent of the recollected emotional images vs the recollected neutral items for the hippocampus would be appreciated. The subsequent memory effect for gamma activity in the hippocampus seems to start more immediately and last longer for the recollected emotional vs neutral images.

R3.1. Action: we regrettably disclose that this analysis is not possible with our spectral analysis.

We appreciate the point raised by the Reviewer. However, within the context of the cluster-based permutation test, it is very difficult to make a strong statement regarding the extent of the cluster. First, the spectral analysis has limitations with regard to the time and frequency resolution (i.e. uncertainty principle: the fundamental limit to the accuracy with which two physical quantities, energy of an oscillation and time, can be predicted). Second, in the context of the cluster-based permutation test, the null hypothesis is about the exchangeability of the data, not about the parameter of the data. This issue appears recurrently in the literature (recently addressed in <https://onlinelibrary.wiley.com/doi/10.1111/psyp.13335> and originally stated in Maris and Oostenveld 2007). As explained [here](https://www.fieldtriptoolbox.org/faq/how_not_to_interpret_results_from_a_cluster-based_permutation_test/) ([https://www.fieldtriptoolbox.org/faq/how not to interpret results from a cluster-based permutation test/](https://www.fieldtriptoolbox.org/faq/how_not_to_interpret_results_from_a_cluster-based_permutation_test/)), the exact latency of the cluster may be influenced by the following factors:

- The signal to noise ratio in the data
- The length of the sliding time window (when calculating frequency data)
- The chosen time bins or regions of interest on which to perform the statistical test
- The threshold chosen to select samples to belong to a cluster (choosing a stringent threshold will lead to a focal effect, while a liberal threshold will produce a widespread effect).
- Coverage of the sEEG electrodes.

A potential solution would be to define an *a priori* window. To avoid circularity, we could only do this by splitting the data in N folds but we would drastically decrease our SNR.

We now make specific reference to these limitations in the main text at lines 300-305

The frequency ranges of gamma activity differed between effects observed in the amygdala (80-120 Hz) and in the hippocampus (50-75 Hz). This variability most likely reflects differences in signal to noise ratio (SNR) suggesting that we are measuring a broadband gamma activity rather than an oscillatory activity. Moreover, cluster-based permutation tests³³ do not provide statistical inference for the exact latency and frequency of the effects³⁴ We selected time windows of broadband gamma activity (60–120 Hz)^{27,28}- as frequency range overlap between the two structures - in the hippocampal recordings.

We refer to these papers already present in the reference list of the original submission:

- 27 Fedele, T. et al. *The relation between neuronal firing, local field potentials and hemodynamic activity in the human amygdala in response to aversive dynamic visual stimuli.* *NeuroImage* 213, doi:10.1016/j.neuroimage.2020.116705 (2020).
- 28 Kucewicz, M. T. et al. *Electrical stimulation modulates high γ activity and human memory performance.* *Eneuro* 5 (2018).
- 33 Maris, E. & Oostenveld, R. *Nonparametric statistical testing of EEG- and MEG-data.* *Journal of Neuroscience Methods* 164, 177-190, doi:10.1016/j.jneumeth.2007.03.024 (2007).

We added this new reference in the revised version of the manuscript:

- 34 Sassenhagen, J. & Draschkow, D. *Cluster-based permutation tests of MEG/EEG data do not establish significance of effect latency or location.* *Psychophysiology* 56, e13335 (2019).

2. It would be helpful to show a supplemental table that lists the number of trials in each emotion x memory condition.

R3.2. Action: we refer the Reviewer to Supplementary Table 6

Supplementary Table 6 (already present in the original submission) provides the number of analyzed trials per patient and conditions following exclusions based on both time- and frequency-domain artifact rejection, and random selection used for the connectivity analysis. We appreciate that there was a lot of information in the supplementary material, so it is not surprising that this was missed!

3. Line 57: It seems like you're trying to tie the same time windows between amygdala gamma power in the amygdala (fig 1) and the changes in lagged granger causal theta activity. I think this could be made even more explicit by referring to the specific subpanel of figure 1 in addition to figure 2.

R3.3. Done with thanks. We now refer to Fig. 1e and Supplementary Fig. 4.

4. Figure 2 D: Please clarify if this is also in response to “viewing” emotional and neutral stimuli.

R3.4. Action: the main text has been changed.

Yes, thanks. This effect is specific for viewing emotional relative to neutral scenes. We changed the text accordingly.

Line 202-205:

*“Amygdala theta phase coupling with hippocampal gamma amplitude was **on average** stronger for **viewing** emotional compared to neutral scenes ($F_{(1, 8)}=6.73$, $P=0.031$, $\eta^2=0.457$, Fig. 2d-e), similar to previous observations during passive viewing of emotional (fearful) faces vs. neutral landscapes¹⁶.”*

5. Figure 3 B right panel: Are these the approximate microwire locations in the hippocampus? Would be helpful to specify if so. It would also be helpful to specify the specific subregion of the hippocampus if possible. A subregion parcellation via ASHS would allow for some estimation of the hippocampal subfield. This could also just be estimated by comparison of each electrode's location to a known atlas. I'd suggest the Mai or Duvernoy atlases.

R3.5. Action: we segmented amygdala and hippocampal subfields.

We are aware that some research groups perform subfield determination in this context. However, we consider this to be quite ambitious given the inaccuracies in post-operative CT to preoperative MRI coregistration, particularly given factors such as brain shifts produced by air introduced during the surgery. We have thus until now only mentioned medial versus lateral localization of the electrodes. On the reviewer's recommendation we now specify in the legend of Fig. 3d that the first contact is at the putative microelectrode location.

“Right side: **d** Summary of electrode contact localization in the left and right amygdala (pink) and hippocampus (light blue) for patients included in the SFC analysis (Cohort 2: 6 patients and a total of 7 electrodes). **The first contact is just lateral to the putative micro-wire location.**”

Furthermore, our revised manuscript now includes hippocampal subfield and amygdala subnuclei segmentation, performed using FreeSurfer V7.2.0. Please note that we are aware of the pros and cons of ASHS vs. Freesurfer, but currently have a collaborative grant with the Freesurfer MTL segmentation group, and have therefore elected to use this software. Detailed electrode localization in amygdala and hippocampus are now provided for one example patient in the revised manuscript as Fig. 3e, with the rest of the patients displayed in Supplementary Fig.2

Fig. 3. Emotional memory-dependent amygdala phase opposition of hippocampal gamma and single neuron activity. a, Hippocampus gamma amplitude locks to different phase bins of amygdala theta depending on emotional memory outcome. Histogram of amygdala theta

*phases at which hippocampus broadband gamma amplitude occurs for eR (red) and eKF (black) trials as bar (left) and circular (right) plots. Inverted triangles represent the angle of preference in radians (eR:0.16 radians; eKF:1.83) after phase realignment. **b**, Circular plot, shaded areas in the circular plot represent PACOi results (n=9 electrodes) per phase bin (n=20). The analytic phase signal was divided in n=20 equal bins ($\phi_n = [-\pi, \pi]$) and the mean analytic amplitude was taken over those specific bins (see Materials and Methods at Phase to amplitude modulation index and Phase-Amplitude Coupling Opposition index (PACOi) sections) for each condition; max is the maximum value of phase-amplitude coupling for eR:1.19 and eKF:1.04. **c**, Histogram displays the permuted values for the Watson Williams test, red line represents the empirical statistic ($F_{ww}=187.49$, $P=0.00009$). ~~**b** Right side: **d**, Summary of electrode contact localization in the left and right amygdala (pink) and hippocampus (light blue) for patients included in the SFC analysis (Cohort 2: 6 patients and a total of 7 electrodes). The first contact is just lateral to the putative micro-wire location. **e**, Example for a single subject showing a post-operative CT image, thresholded to visualize electrode contacts, co-registered with the corresponding pre-operative MRI scan in native space and superimposed onto the amygdala and hippocampus subfields. The sagittal view shows that contacts are in the amygdala and anterior hippocampus. The first coronal view shows hippocampal subfields and the white and black arrow point to the putative microelectrode location. The second coronal view shows amygdala nuclei and the two white and black arrows indicate the first two macroelectrodes locations. Subfields/nuclei are color coded. A schematic of the electrode implanted (1.3 mm diameter, 8 contacts of 1.6 mm length, and spacing between contact centers 5 mm; Ad-Tech, Racine, WI) is provided showing microwires protruding from the tip and macrocontacts (the first three contacts are numbered). **d**, **f**, As for **a**, but showing amygdala theta phases at which hippocampal spikes occur (average spike count over n=31 neurons. Inverted triangles represent the angle of preference in radians (eR:0.18; eKF:1.40) after phase realignment. **g**, Shaded areas in the circular plot represent spikes per the 20 phase bins for each condition. Maximum spike count for each condition is indicated (eR:112; eKF:128) as well as the total spike counts across neurons (eR:1651, eKF:2030). Upper subpanel: single-neuron waveform (mean±std) for one example neuron. **h**, Histogram displays the permuted values for the Watson Williams test, red line represents the empirical statistic ($F_{ww}=40.91$, $P=0.00009$). ~~Upper subpanel: single-neuron waveform (mean±std) for one example neuron.~~~~*

Supplementary Fig. 2: Electrode contact localization for patients (Cohort 2) with macro and micro electrodes localized in the amygdala (pink outline) and hippocampal (light blue outline) subfields. Arrows in the figure representing hippocampal subfields point to the putative microelectrodes location which extends from the first contact as can be seen in the schematic example of the electrode. Arrows in the figure representing the amygdala subfields point to the contacts from where the LFP was extracted for the spike field coherence analysis. For all patients, post-operative CT images from each patient were co-registered with their corresponding pre-operative MRI scans in native space using lead-DBS v2.5 and superimposed to display amygdala and hippocampal contacts (CTs were thresholded so as to only show electrode contacts). Amygdala and hippocampal subfields were obtained using FreeSurfer V.7.2.0. For each subject and brain structure the legend reports the amygdala and hippocampal subfields where contacts are localized. All amygdala and hippocampal subfield colour codes are shown below, and a schematic example of the implanted electrode type (1.3 mm diameter, 8 contacts of 1.6 mm length, and spacing between contact centers 5 mm; Ad-Tech, Racine, WI) is provided.

6. I appreciate the thoughtfulness put into connecting these analyses. Given the strong timing relationships between each of the phase, modulation, and transient coherence analyses, I think would be very helpful to include a schematic diagram that illustrates the relationship between each of these oscillatory features. This could be an idealized illustration of these oscillatory features. This could also be used as a graphical abstract of the studies novel findings.

R3.6. Action: we include Figure 5 to illustrate the relationship between results.

We thank the Reviewer for the suggestion and now include the following figure (Fig. 5) as a schematic diagram of the results. Please note that, as indicated by the associate editor, Nature Communications does not support graphical abstracts.

[REDACTED]

Fig. 5. Summary and time course of amygdala hippocampal activity and coupling during emotional memory formation. All data reported pertain to encoding-related responses (grey box). Peri-stimulus time is indicated by the downward arrow (left), with process onsets and durations (solid color bar) observed in the amygdala (pink), in the hippocampus (blue) or implying a dynamic between the two structures (pink and blue). Effects at encoding are divided (from left to right) based on whether we observed a significant effect for aversive scenes (emotion column), for subsequently remembered scenes (memory column), or are significantly discriminative for emotional vs. neutral memory formation (emotional memory column). At earliest latency, amygdala gamma activity increases more for subsequently remembered vs. not remembered emotional vs. neutral pictures. Next, amygdala-hippocampal PAC and Granger causal effects are associated with viewing aversive scenes as compared to neutral ones. The pink horizontal arrow represents the directionality of Granger causal influence from the amygdala to the hippocampus. Gamma activity in the hippocampus increased for stimuli that are later recollected independently of their valence. Hippocampal gamma and single-unit

responses aligned to different amygdala theta phases during subsequently remembered vs. not-remembered (known/forgotten) emotional stimuli. The vertical rectangle represents the amygdala theta phase to which hippocampal gamma activity and neuronal firing couple for aversive scenes later remembered. If gamma activity in the hippocampus occurred during a different period of the theta cycle (outside the rectangle), aversive scenes were not subsequently remembered. Thick dashed line represents the transient lagged coherence between gamma activity in the amygdala and the hippocampus that is observed only for aversive scenes later remembered. The latency of around 37 ms is consistent with the phase difference between the two conditions around 25-45 ms. These results together point to an amygdala-hippocampus phase code for aversive memory formation.

Discussion

1. Line 107: Please also cite Bass et al., 2012, 2014, 2015 as further illustrations that theta-gamma stimulation to the amygdala is effective in rodents for organizing hippocampal responses and enhancing memory.

R3.7. Done with thanks. We included the following references in the discussion.

- 36 Bass, D. I., Partain, K. N. & Manns, J. R. Event-specific enhancement of memory via brief electrical stimulation to the basolateral complex of the amygdala in rats. *Behavioral neuroscience* **126**, 204 (2012).
- 37 Bass, D. I., Nizam, Z. G., Partain, K. N., Wang, A. & Manns, J. R. Amygdala-mediated enhancement of memory for specific events depends on the hippocampus. *Neurobiology of learning and memory* **107**, 37-41 (2014).
- 41 Bass, D. I. & Manns, J. R. Memory-Enhancing Amygdala Stimulation Elicits Gamma Synchrony in the Hippocampus. *Behavioral Neuroscience* **129**, 244-256, doi:10.1037/bne0000052 (2015).

2. For citation 31, the direct amygdala stimulation occurred immediately following the emotionally neutral stimuli, rather than simultaneous to the stimuli.

R3.8. Apologies for this slip. We modified the discussion accordingly, see line 419-422

In rodents, theta-modulated gamma stimulation applied to BLA is an efficient protocol to enhance hippocampal CA1 gamma responses^{40,41}, and directly stimulating the human amygdala at theta-modulated gamma frequency immediately following the presentation of emotionally neutral stimuli leads to memory enhancement ~~for simultaneously presented emotionally neutral stimuli~~⁴².

3. Line 112: Given the clear role of recollection for differences in emotional memory findings in this study, please more clearly specify that these effects are tied to formation of strongly recollected emotional memories relative to familiar or forgotten stimuli. This is currently done parenthetically, but it's not immediately clear what is meant.

R3.9. Done with thanks. When effects are discriminative for subsequent remembering of aversive scenes, we now clarify that this effect is relative to known/forgotten trials.

Lines 382-384.

“High gamma activity in the amygdala was enhanced at 310 ms after stimulus presentation with a greater amplitude for emotional aversive scenes that were later remembered (relative to those receiving known responses or forgotten ones).”

4. Line 122: This work also might have implications for therapeutic approaches of amygdala stimulation for memory disorders, in addition psychiatric disorders. I would recommend also mentioning this.

R3.10. Done with thanks. We changed the text accordingly, see lines 454-458.

“Moreover, these findings could represent a general mechanism through which amygdala oscillations influence other brain areas to enable emotion-induced modulation of further aspects of cognition, including perception, attention and decision-making^{14,51}, and inform therapeutic approaches of amygdala stimulation to memory dysfunctions and psychiatric disorders. “

5. While I appreciate the brevity and conciseness of the discussion, if there is space, I would suggest adding more discussion. The schematic illustration of the papers related timing findings could be thoroughly discussed. I also think it would be helpful to further discuss the implications of this work as it relates to modulating or interfering with amygdala or hippocampal activity via direct brain stimulation. Finally, there could be more discussion of the limitations and future directions of this work.

R3.11. Action: The discussion has been extended.

Done with thanks. We have extended the discussion as suggested (lines 380-458), as well as including the suggestions of reviewer 2 (**R2.m3**).

“Our data show that in response to an aversive visual stimulus, the amygdala influences ongoing hippocampal theta oscillations, which in turn organize the amplitude of local gamma activity and neuronal firing (Figure 5). High gamma activity in the amygdala was enhanced at 310 ms after stimulus presentation with a greater amplitude for emotional aversive scenes that were later remembered (relative to those receiving known responses or forgotten ones). Subsequently, aversive scenes triggered unidirectional transmission of theta oscillations from the amygdala to the hippocampus compared to neutral ones (from 430 to 770 ms). At around 500 ms, gamma power increased in the hippocampus for stimuli that were later recalled irrespective of their valence. Considering a time window from 400 to 1000 ms, the alignment of hippocampal broadband gamma activity (60-120 Hz) and neuronal firing to the amygdala’s theta phase differed between subsequently remembered vs. not-remembered emotional stimuli. We observed a consistent phase difference between the two conditions of ~1.67 radians, corresponding to approximately 30-45 ms. Crucially, this time difference coincides with the latency of a transient lagged coherence (~37 ms) between gamma activity in the two structures that predicts subsequent memory for emotional stimuli (Fig. 5).

The BLA has strong connections to the hippocampus and electrical stimulation of the rat BLA improved performance in memory tasks^{36,37}. Theta-gamma phase-amplitude coupling in the human hippocampus has been shown to be a general mechanism for memory encoding^{28,29}. In rodents, theta-modulated gamma stimulation applied to BLA is an efficient protocol to enhance hippocampal CA1 gamma responses^{40,41}, and directly stimulating the human amygdala at theta-modulated gamma frequency immediately following the presentation of emotionally neutral stimuli leads to memory enhancement for simultaneously presented emotionally neutral

stimuli⁴¹. Theta-gamma phase-amplitude coupling in the human hippocampus has been shown to be a general mechanism for memory encoding^{38,39}. Here, we showed that emotional memory enhancement does not simply depend on theta-gamma phase-amplitude coupling. Confirming theoretical positions that memory formation is phase-dependent in humans^{23,24}, we showed that the formation of memories for emotional stimuli (~~allowing later recollected emotional memories relative to familiar or forgotten ones~~) depends on the amygdala theta phase to which hippocampal gamma and related neuronal firing couples.

We highlight the importance of theta modulation of gamma activity in encoding of emotional memories, and lend support to direct stimulation protocols employing theta-burst stimulation aimed at improving memory⁴²⁻⁴⁴. Theta-burst stimulation is the delivery of several stimulation pulses at high frequency (*i.e.*, gamma frequency) that rhythmically alternate (in the theta range) with periods of no stimulation. By contrast, deep brain stimulation of the amygdala applied continuously at high frequency (*e.g.*, 160 Hz), currently being trialed for PTSD⁴⁵, may not permit the amygdala-hippocampal phase code mechanism described here to take place in response to emotional events, thereby limiting the formation of novel emotional memories.

Hippocampal long term potentiation (LTP), a form of synaptic plasticity thought to be involved in learning and memory^{30,46}, is optimally induced in CA1 when electrical stimulation occurs at the peak of the theta oscillation⁴⁷. This has led to a suggestion that inputs arriving during different specific theta phases will generate different synaptic modifications, which in turn will influence the likelihood that these inputs are encoded into long term memory^{31,47,48}. Our findings support this suggestion, by showing that the precise theta phase at which hippocampal gamma peaks and spikes occurred determined whether aversive scenes were later remembered, or not. Critically, the phase difference between subsequently remembered *vs.* forgotten or familiar emotional stimuli ~~associated with this process~~ translates to the time period required for amygdala and hippocampal gamma bursts to reach transient time-lagged coherence (~25-45 ms). It is possible that this lag is related to the time required for noradrenergic input, upon which emotional memory formation is critically dependent^{12,13}, to reach the medial temporal lobe, or that amygdala theta phase-dependent effects in the hippocampus are linked to the optimal conditions required for "emotion tagging" of memory⁴⁹ to occur. Future studies may explore whether a similar phase offset is present at retrieval or whether a different phase relationship may exist between amygdala and hippocampus. As encoding and retrieval have been shown to rely on complementary processes, the latter may be hippocampus-centered and thus may not rely on amygdala modulation^{12,50}. Moreover, these findings could represent a general mechanism through which amygdala oscillations influence other brain areas to enable emotion-induced modulation of further aspects of cognition, including perception, attention and decision-making^{14,33}, and inform therapeutic approaches of amygdala stimulation to memory dysfunctions and psychiatric disorders.”

Regarding limitations of our work, these are now mentioned in different places in the manuscript:

Lines 300-305

The frequency ranges of gamma activity differed between effects observed in the amygdala (80-120 Hz) and in the hippocampus (50-75 Hz). This variability most likely reflects differences in signal to noise ratio (SNR) suggesting that we are measuring a broadband gamma activity rather than an oscillatory activity. Moreover, cluster-based permutation tests³³ do not provide statistical inference for the exact latency and location of the effects³⁴. We selected time windows of broadband gamma activity (60–120 Hz)^{27,28} - as frequency range overlap between the two structures in the hippocampal recordings.

We refer to these papers already present in the reference list of the original submission:

- 27 Fedele, T. et al. *The relation between neuronal firing, local field potentials and hemodynamic activity in the human amygdala in response to aversive dynamic visual stimuli. NeuroImage 213*, doi:10.1016/j.neuroimage.2020.116705 (2020).
- 28 Kucewicz, M. T. et al. *Electrical stimulation modulates high γ activity and human memory performance. Eneuro 5* (2018).
- 33 Maris, E. & Oostenveld, R. *Nonparametric statistical testing of EEG- and MEG-data. Journal of Neuroscience Methods 164*, 177-190, doi:10.1016/j.jneumeth.2007.03.024 (2007).

We added this new reference in the revised version of the manuscript:

- 34 Sassenhagen, J. & Draschkow, D. *Cluster-based permutation tests of MEG/EEG data do not establish significance of effect latency or location. Psychophysiology 56*, e13335 (2019).

Lines 724-735

“Although PACOi measures phase difference indirectly (see VanRullen 2016 for an in-depth discussion about phase opposition measures in general), it provides substantial advantages for dealing with two important problems that need to be considered in the context of phase estimation: 1) the magnitude of the phase concentration by itself is not informative. 2) Relative vs. absolute phase difference. PACOi measures whether the phase opposition (*i.e.*, phase difference) is higher than a null model where trials are permuted⁶⁹. Since it is based on differences in the sum of PPCs of two conditions relative to the sum of PPCs of these two conditions with the trials permuted (null model), the PACOi measure is insensitive to the angle where the phase difference occurs. In other words, the advantage of using PACOi method is that PACOi can measure phase difference independently of where it occurs along the 0-2 π interval. This approach is analogous to classical coherence analysis, where consistent phase differences between channel-pairs across trials are clipped between 0-1 independently of their location in the Angard diagram.”

- 69 VanRullen, R. *How to evaluate phase differences between trial groups in ongoing electrophysiological signals. Frontiers in neuroscience 10*, 426-426 (2016).

Methods

1. Thank you for the thorough methods section and supplementary information on patients. Well done!

Many thanks again.

REVIEWER COMMENTS

Reviewer #1 (Remarks to the Author):

The authors have provided a considered and detailed response to reviewer concerns, I have no further comment.

Reviewer #2 (Remarks to the Author):

The authors are to be commended for thoroughly addressing all my concerns. Their manuscript has been substantially improved. My only suggestion is that they include the explanatory schematic for Fig 4D in the main text and not as a supplementary figure.

Reviewer #3 (Remarks to the Author):

All of my comments have been thoroughly addressed. Well done!

Cory Inman
Assistant Professor
University of Utah

REVIEWER COMMENTS

We would like to once again thank all three expert Reviewers for their constructive contribution to the peer review of this work which has improved the quality of our manuscript. We are delighted that our responses were satisfactory and no further comments need to be addressed.

Reviewer #1 (Remarks to the Author):

The authors have provided a considered and detailed response to reviewer concerns, I have no further comment.

We thank the Reviewer for the positive feedback and we are delighted that our responses were satisfactory.

Reviewer #2 (Remarks to the Author):

The authors are to be commended for thoroughly addressing all my concerns. Their manuscript has been substantially improved. My only suggestion is that they include the explanatory schematic for Fig 4D in the main text and not as a supplementary figure.

We thank the Reviewer for the positive feedback. We split Fig. 4 in two figures and included the schematic depiction of the correlation analysis as part of Fig. 5 that can be found below.

Fig. 5. Broadband gamma (60–120 Hz) transient connectivity between amygdala and hippocampus is paced by amygdala theta-hippocampal gamma phase opposition. a, schematic depiction of the correlation analysis displayed in **b**. Left: example trial of broadband amygdala (pink) and hippocampal (blue) gamma activity. Cross correlation between hippocampal gamma peaks (blue arrow) and amygdala gamma peaks (pink arrow) for eR and eKF conditions was computed. Right: as example, we show hippocampal gamma activities (blue) occurring at two different amygdala theta phase (pink sine wave) for eR (red arrow) and eKF (black) conditions. Individual data for the cross correlation of amygdala hippocampus broadband activity peak lag were correlated with the PACOi frequency-specific phase opposition result translated into a time lag using the formula displayed in the figure. **b,** Correlation between successful vs. unsuccessful emotional memory as measured as hippocampus-amygdala transient broadband gamma activity peak lag (x-axis) and hippocampus gamma delay as a function of amygdala theta cycle (y-axis). The colored dots (red, blue, purple) represent 3 different patients for whom individual data is shown above and right (with corresponding frame and font color). In these 3 plots, left: time corresponds to the hippocampus-amygdala transient broadband gamma activity cross-correlation peak lag for eR vs. eKF trials (colored inverted triangle). The shaded contours represent \pm s.e.m.; right: hippocampus gamma delay as a function

of amygdala theta cycle (the delay is the difference between black and red triangles). Note the different sign between Patient 15 (green) and Patient 16 (purple) in the two correlated measures. Source Data are provided as a Source Data file.

Reviewer #3 (Remarks to the Author):

All of my comments have been thoroughly addressed. Well done!

Cory Inman

Assistant Professor

University of Utah

We are grateful to Cory Inman for his positive and encouraging feedback. Many thanks for your comments and suggestions!